

# Insights into the size-resolved dust emission from field measurements in the Moroccan Sahara

Cristina González-Flórez[1], Martina Klose[2], Andrés Alastuey[3], Sylvain Dupont[4], Jerónimo Escribano[1], Vicken Etyemezian[5], Adolfo Gonzalez-Romero[1,3], Yue Huang[6], Konrad Kandler[7], George Nikolich[5], Agnesh Panta[7], Xavier Querol[3], Cristina Reche[3], Jesús Yus-Díez[3,8], and Carlos Pérez García-Pando[1,9]

[1]Barcelona Supercomputing Center (BSC), Barcelona, Spain
[2]Karlsruhe Institute of Technology (KIT), Institute of Meteorology and Climate Research - Department Troposphere Research (IMK-TRO), Karlsruhe, Germany
[3]Institute of Environmental Assessment and Water Research (IDAEA-CSIC), Barcelona, Spain
[4]INRAE, Bordeaux Sciences Agro, ISPA, Villenave d'Ornon, France
[5]Desert Research Institute, Las Vegas, US
[6]NASA Goddard Institute for Space Studies, New York, US
[7]Institute of Applied Geosciences, Technical University of Darmstadt, 64287 Darmstadt, Germany
[8]Grup de Meteorologia, Departament de Física Aplicada, Universitat de Barcelona, C/Martí i Franquès, 1, 08028, Barcelona, Spain
[9]ICREA, Catalan Institution for Research and Advanced Studies, Barcelona, Spain

**Correspondence:** C. González-Flórez, cristina.gonzalez@bsc.es

**Abstract.** Mineral dust effects upon climate are strongly affected by its particle size distribution (PSD). In particular, the emitted dust PSD partly controls the dust lifetime and its global distribution. Despite the extensive research performed on this topic over the last decades, there are still substantial gaps in our understanding of the emitted PSD along with its potential variability and associated causes. In this study, we provide insights into the saltation and size-resolved dust emission process

based on measurements obtained during a comprehensive wind erosion and dust emission field campaign that took place in the Moroccan Sahara in September 2019 in the context of the FRontiers in dust minerAloGical coMposition and its Effects upoN climaTe (FRAGMENT) project. The measurement site located in a remote ephemeral lake, consisting of a smooth hard-crusted paved sediment surface surrounded by small sand dunes, is characterized by strong and frequent saltation and dust emission conditions, and relatively low sandblasting efficiencies. Our study, which thoroughly analyses the number and mass PSDs

of both the concentration and diffusive flux (the latter typically assumed to be equivalent to the emitted dust PSD), detects statistically significant dependencies upon friction velocity ($u_*$), wind direction and type of event (regular events vs haboob events). We discuss the potential underlying causes of such variability, including the effect of dry deposition, an enhanced fragmentation of aggregates, and the impact of the haboob gust front. We clearly identify and quantify the major role played by dry deposition in shaping the diffusive flux PSD variations, modulated by the wind direction-dependent fetch length of

our measurement location and $u_*$. Our estimates show the importance of dry deposition relative to emission, representing up to $\sim$40 % for super-coarse particles ($> 10\,\mu m$) and up to $\sim$20 % for particles as small as $\sim$5 μm in diameter. While we attribute the enhancement (reduction) in submicron (supermicron) particles with $u_*$ to the effect of dry deposition, an enhanced fragmentation of aggregates with $u_*$ could still play a complementary yet arguably smaller role. We additionally find clear





differences in the PSDs associated to haboob events in comparison with the regular events, i.e., a higher (lower) proportion

of supermicron (submicron) particles for equivalent or higher $u_*$ values, and more vigorous dry deposition and variability in the coarse and super-coarse dust mass fractions. We hypothesize that these differences are due to 1) a smaller horizontal (spatial) extent of the haboob events (which is equivalent to the effect of a smaller fetch), 2) the effect of the moving haboob gust front, where $u_*$ and dust emission are maximized, along with its changing proximity to the measurement site (which is equivalent to a variable fetch), and/or 3) the increased resistance of soil aggregates to fragmentation associated to the observed

increases in relative humidity along the haboob outflow. We finally compare the obtained PSDs with both the PSDs predicted by the original and a recently updated version Brittle Fragmentation Theory (BFT), the latter accounting for super-coarse dust emission. For the comparison with the updated BFT we transform our optical diameters into geometric diameter PSDs, assuming dust particles are tri-axial ellipsoids with an index of refraction consistent with measured optical properties during the campaign. We obtain a substantially lower (higher) proportion of submicron (supermicron) particles in the diffusive flux

PSDs in comparison with the original BFT PSDs. Also, our PSDs show a higher proportion of particles above $\sim 2\,\mu m$ and a higher mass fraction of super-coarse particles, despite large effect of dry deposition upon this fraction. All in all, our results indicate that dry deposition needs to be adequately considered to estimate the emitted PSD, even in studies limited to the fine and coarse size ranges ($< 10\,\mu m$), and particularly in measurement locations with long fetches.

## 1 Introduction

Mineral dust emitted by wind erosion from arid and semi-arid regions dominates the global aerosol mass load (Textor et al., 2006) and plays a key role in the Earth System by perturbing the energy, water, iron, phosphorous and carbon cycles (Okin et al., 2004; Bristow et al., 2010; Shao et al., 2011b; Knippertz and Stuut, 2014; Jickells and Moore, 2015). The effects of dust aerosol are controlled by its amount and physico-chemical properties, i.e. particle size distribution (PSD), mineralogy, shape, and mixing state (Tegen and Lacis, 1996; Karanasiou et al., 2012; Huang et al., 2014; Miller et al., 2014).

Despite the progress achieved over the last decades, the size-resolved dust emission flux and its spatio-temporal variability remain as key uncertainties in the description of the dust life cycle in atmospheric and Earth System models (Kok, 2011a; Evan et al., 2014; Adebiyi and Kok, 2020; Klose et al., 2021). Dust emission is complex: the most efficient release of dust particles is through saltation (Gillette, 1977; Gomes et al., 1990; Shao et al., 1993; Shao, 2008), which is – as dust emission itself – modulated by soil properties (e.g. soil texture, mineralogical composition, presence and stability of aggregates), surface soil

conditions (e.g. moisture, vegetation cover, crust, roughness) and land use (e.g. agriculture, grazing) (Tegen et al., 2002; Pierre et al., 2012; Perlwitz et al., 2015a, b; Klose et al., 2019). Current global quantitative knowledge of many of these factors is poor or nonexistent, which demands certain simplifications in model dust emission schemes.

The emitted dust PSD and its variability has attracted much attention over the last years (Alfaro et al., 1997; Fratini et al., 2007; Sow et al., 2009; Shao et al., 2011a; Kok, 2011a, b; Ishizuka et al., 2014; Khalfallah et al., 2020; Shao et al., 2020;

Dupont, 2022). Constraining the PSD at emission is crucial as the residence time of dust particles in the atmosphere is strongly influenced by their size, with coarser particles falling out more quickly due to gravitational settling (Ryder et al., 2013).





The majority of dust particles are likely released through saltation bombardment, in which soil aggregates are fragmented by impacts from larger saltating grains, and aggregate disintegration, in which saltating aggregates are fragmented upon striking the soil surface (Shao et al., 1993; Shao, 2001; Alfaro et al., 1997). In the particle size range up to $\sim 10\,\mu m$ in diameter,

some theoretical frameworks predict enhanced aggregate disintegration (or fragmentation) with increasing wind speed during saltation and thus a higher proportion of emitted fine particles, along with dependencies of the PSD on soil properties (Shao et al., 1993; Alfaro et al., 1997; Shao, 2001). In contrast, the emitted PSD is posited to be relatively independent of wind speed and soil properties in another theoretical framework (Kok, 2011a), based on Brittle Fragmentation Theory (BFT). The scarcity of data and the observational uncertainties further hamper robust conclusions about the potential variability of the emitted

PSD. It has been argued that observed variations in the emitted PSD may be largely within the systematic errors among the experimental datasets (Kok et al., 2017). There is even more uncertainty in the emission of particles larger than $10\,\mu m$, whose contribution to transport and climate is thought to be underestimated (Kok, 2011a; Ryder et al., 2019; Adebiyi and Kok, 2020), due to 1) the lack of field data, 2) the limitations related to the inlets of optical particle counters and other aerosol samplers used for reference measurements, 3) the lower amount of particles (which increases uncertainties), and 4) the potential effect

of dry deposition upon the calculated diffusive fluxes (Dupont et al., 2015; Fernandes et al., 2019; Adebiyi et al., 2022).

Most studies use the flux-gradient method to obtain the diffusive flux PSD. Because this approach assumes a constant flux layer, the net (emitted) dust flux at the surface equals the obtained diffusive dust flux a few meters above the surface if gravitational settling is neglected (Dupont et al., 2021). Since the gravitational settling term is assumed to be small for dust smaller than $\sim 10\,\mu m$ (Fratini et al., 2007), most studies have traditionally assumed that the diffusive flux PSD is equivalent to

the emitted dust PSD, with the exception of Shao et al. (2011a). The diffusive flux PSD is therefore used directly to constrain or evaluate dust emission schemes, or even to assess to what extent the emitted dust PSD may be affected by atmospheric forcing and soil properties. However, using modeling Dupont et al. (2015) and more recently Fernandes et al. (2019) have shown the potentially large effect of dry deposition (including losses by turbulent and Brownian motion, and inertial impaction) upon the diffusive flux PSD.

Given the incompleteness of measurements, and the apparent contradiction among theories, field observations and wind tunnel experiments, the European Research Council project entitled FRontiers in Dust Mineralogical Composition and its effects upon climate (FRAGMENT) has conducted field campaigns in distinct desert dust source regions to better understand the size-resolved dust emission for a range of meteorological and soil conditions. The goal of FRAGMENT is to better understand dust emission, its mineralogical composition and the effects of dust upon climate, by combining field measurements, labora-

tory analyses, remote and in situ spectroscopy, theory and modelling. In this contribution, we provide new insights into the size-resolved dust emission and its variability using measurements collected during the first FRAGMENT field campaign that took place in the Moroccan Sahara in September 2019, taking advantage of the large number of dust events of varying intensity captured during this one-month measurement period.

Section 2 describes the field measurement site and the experimental set-up, along with the methodology used for calculating

1) the dynamical parameters characterizing key properties of the near-surface boundary layer, 2) the saltation flux, 3) the diffusive dust flux and its uncertainties, and 4) the sandblasting efficiency. It also describes the dry deposition resistance-based





scheme used to further support our analysis of the variability in the dust PSDs. Section 3 first overviews the atmospheric conditions and dust events measured during the campaign and provides a broad characterization of the saltation and diffusive fluxes, along with the associated sandblasting efficiences. Then, a variety of aspects related to the dust PSD at emission and
its variability are thoroughly analyzed and discussed, including the identification and removal of the anthropogenic aerosol influence, the differences between the concentration and diffusive flux PSDs and their dependencies upon friction velocity ($u_*$) and wind direction, the PSD differences between two major types of events measured, the potential role of different mechanisms in the variability of the PSDs, and the comparison of our measured PSDs with BFT. Section 4 draws the main conclusions of the study and the perspectives for future work.

**2   Data and methods**

**2.1   The FRAGMENT dust field campaign in the Moroccan Sahara**

The first FRAGMENT field campaign took place in September 2019 in a small ephemeral lake, locally named "L'Bour", located in the Lower Drâa Valley of Morocco. L'Bour (29°49'30" N, 5°52'25" W) lies at the edge of the Saharan Desert, ∼15 km west of M'Hamid El Ghizlane, ∼70 km east of Lake Iriki, ∼50 km east of the Erg Chigaga dune field, ∼1.5 km north
of the dry Drâa river, ∼30 km north of the Moroccan-Algerian border, and ∼25 km south of the Jbel Hassan Brahim mountain range (840 m.a.s.l) (Figs. 1a, 1b and 1c). We chose the location and time period of the campaign based on the analysis of remote sensing data (Ginoux et al., 2010), in situ inspection and local advice, considering both scientific criteria and logistic aspects such as accessibility.

    L'Bour is approximately flat and devoid of vegetation or other obstacles within a radius of ∼1 km around our measurement
location. Small sand dune fields surround the lake, and during the campaign, dunes south of the site were accompanied by some vegetation/shrubs. The surface of L'Bour consists of a smooth hard crust (hereafter referred to as paved sediment) mostly resulting from drying and aeolian erosion of paleo-sediments (González-Romero et al., in prep.). In Appendix A we include a close-up of a small dune and the lake's paved sediment surface, along with their respective PSDs analyzed using dry dispersion (minimally dispersed) and wet dispersion (fully dispersed) techniques (González-Romero et al., in prep.). The
paved sediment PSDs exhibit two prominent modes peaking at ∼ 100 μm and ∼ 10 μm. The fully dispersed PSD of the paved sediment shows disaggregation of silt aggregates observed at sand sizes in the minimally dispersed PSD. The sand dune PSDs display a dominant mode ranging between ∼50 and ∼400 μm peaking at (∼ 150 μm) and contain only a small fraction of particles smaller than 50 μm. The fully dispersed PSD of the sand dune shows disaggregation of clay aggregates observed at silt sizes in the minimally dispersed PSD. The volume median diameter of the sand dune (and therefore of the saltators) for
minimally and fully dispersed techniques are 132.2 μm and 137.6 μm, respectively. According to the fully dispersed PSD, the texture of the surface paved sediment is loam (McKee, 1983). During the campaign, we did not observe any substantial change in the paved sediment. We observed some growth of vegetation in nearby areas, particularly to the south, after a flooding event that took place during the night of September 6th. The flooding, which did not affect our site, was caused by a convective storm





that produced heavy rain upstream of the Drâa river and whose cold pool outflow generated a strong "haboob" dust storm that
passed our site (see Sect.3.1).

L'Bour is surrounded by other dust sources in all directions, including dunes concentrated in small flat areas and other
ephemeral lakes such as Iriki and Erg Smar. Therefore the fetch length (i.e., the distance between the measurement location
and the upwind border of the source area (Dupont et al., 2021) is not limited to the dimensions of L'Bour. We estimate long
fetches of about 60 km and 10 km in the western and eastern predominant wind directions (see Fig. 2e or Appendix B),
respectively, which are approximately parallel to the Drâa river bed and perpendicular to the alignment of our instruments (Fig.
1d), as described in Sect. 2.2.

## 2.2    Field measurements

The site layout is shown in Figs. 1d and 1e. The alignment of the instruments was informed by prior analysis of nearby
automated weather stations, maintained by the IMPETUS and FENNEC projects (Schulz and Judex, 2008; Hobby et al.,
2013); the enerMENA initiative (Schüler et al., 2016), and ERA5 and ERA-Interim wind reanalysis, which suggested a south-
westerly predominant wind direction. To avoid shadowing between instruments as much as possible, instruments were aligned
roughly perpendicular to this predominant wind direction. Below we describe only the instruments and measurements used in
this contribution. Measurements performed during the campaign with other instruments displayed in Fig. 1d are discussed in
companion papers (e.g. Panta et al., in prep.; Yus-Díez et al., in prep.).

### 2.2.1    Meteorological measurements

In the center of the experimental site (Fig. 1d), we deployed a 10-m meteorological tower equipped with five 2-D sonic
anemometers (Campbell Scientific WINDSONIC4-L) at 0.4 m, 0.8 m, 2 m, 5 m and 10 m height and four aspirated shield
temperature sensors (Campbell Scientific 43502 fan-aspirated shield with 43347 RTD Temperature Probe) at 1 m, 2 m, 4 m,
and 8 m height to measure wind and temperature profiles, respectively (Fig. 1e). Wind measurements were recorded every 2 s
and temperature every 1 s. (We also placed two 3-D sonic anemometers measuring at 50 Hz at 1 m and 3 m height that are
not used in this paper.) All anemometers were oriented toward the north using a magnetic compass. A site-specific correction
for magnetic declination using the International Geomagnetic Reference Field (IGFR) model (1590-2024) was applied as a
post-processing, which translated into a counterclockwise adjustment of $\sim 1\,°$ to the measured wind direction respective to the
true north. In the vicinity of the tower, we installed a Young tipping bucket rain gauge (Campbell Scientific 52203 unheated
Rain Gauge) at 1 m height, a four-component net radiometer (Campbell Scientific NR01-L radiometer) measuring short-wave
and long-wave upwelling and downwelling radiative fluxes at 1.5 m, and a temperature and relative humidity probe (Campbell
Scientific HC2A-S3) at 0.5 m (Fig 1e). Pressure was recorded inside the data logger cabinet in a tripod near the tower.

The time series of the measurements described above were inspected in order to detect and remove invalid values. Most
of them corresponded to periods of testing at the beginning of the campaign or instrument cleaning, and were identified and
deleted manually. We averaged all meteorological variables over 15 minute intervals, consistent with the time averaging chosen





**Figure 1.** a) Location of study area in northern Africa. b) Zoom over Morocco and Algeria. c) Zoom over the Lower Drâa Valley. d) Experimental setup in "L'Bour" (Morocco). The diagonal black line is perpendicular to the approximate predominant wind direction estimated based on prior data analysis. Green circles highlight the instruments used for this paper: TOWER (meteorological tower equipped with five 2-D sonic anemometers and four aspirated shield temperature sensors), FIDAS (two Fidas optical particle counters at 1.8 and 3.5 m height, respectively), RAIN-GAUGE, RADIOMETER (four-component net radiometer), RH-T (temperature and relative humidity probe at 0.5 m), SANTRI-4 (Size-resolved saltation particle counter). Red circles indicate instruments not used in this contribution, but discussed in companion papers: FWI1, FWI2 and FWI3 (Free-Wing Impactors), FPS (Flat-Plate deposition sampler), LOW-VOL-PM10 and LOW-VOL-TSP (Low volume samplers), AETH/NEPH (multi-wavelength aethalometer and polar nephelometer), MWAC (Modified Wilson and Cook samplers), SMOIS (soil moisture sensors), TRIPOD (pressure and data loggers); e) Picture of the main instruments as deployed in the field.



to compute the dynamical parameters characterizing the near-surface boundary layer. This averaging time has been shown to account for all significant turbulent structures carrying momentum flux (Dupont et al., 2018).

### 2.2.2 Size-resolved dust concentration measurements

At a distance of $\sim 18\,\mathrm{m}$ from the tower, we placed two Fidas 200S (Palas GmbH) optical particle counters (OPCs) on a scaffold-
ing (Fig 1e) at $1.8\,\mathrm{m}$ (referred to as FidasL) and $3.5\,\mathrm{m}$ height (FidasU) from which we calculate the dust fluxes (see Sect. 2.3.2). We recorded 2-min average number concentrations of suspended dust in sixty-three diameter size bins of equal logarithmic width between 0.2 and $19.1\,\mu\mathrm{m}$ that were averaged over 15 minutes for analysis. Data from the first three bins were not used as they showed an unrealistic abrupt descent of the concentration (border measurement limitations). Therefore, we considered the Fidas to be efficient from the fourth bin (from $0.25\,\mu\mathrm{m}$). The sampling system of the Fidas operates with a volume flow of
$4.8\,\mathrm{l\,min^{-1}}$ and is equipped with a Sigma-2 sampling head (manufacturer Palas GmbH). The Sigma-2 sampler has been vali- dated by the Association of German Engineers (VDI-2119, 2013) and tested in various studies concluding that it is a reliable collector for coarse and super-coarse particles (Dietze et al., 2006; Tian et al., 2017; Waza et al., 2019; Rausch et al., 2022). The Sigma-2 head is expected to be largely insensitive to wind intensity (Waza et al., 2019) as it ensures a wind-sheltered, low-turbulence air volume inside the sampler (Tian et al., 2017). The inlet includes a drying line (Intelligent Aerosol Drying
System, IADS, Palas GmbH), connecting the sampling head to the control unit, whose temperature is regulated according to the ambient temperature and humidity, avoiding condensation effects. Moisture compensation is guaranteed through a dynamic adjustment of the IADS temperature up to a maximum heat capacity of $90\,\mathrm{W}$. Unlike most of the meteorological instruments that were connected to a battery that could be charged either by the power generator or a solar panel, the two Fidas depended exclusively on the generator. Therefore, there were some gaps in the time series associated to generator maintenance periods
and to some short power blackouts.

In Sect. 3.3 we analyze the 15-min concentration PSDs averaged over $u_*$ intervals. For each $u_*$ interval we also provide the standard error, which measures how far the calculated average is likely to be from the true average. Therefore, uncertainty is proportional to the standard deviation and inversely proportional to the square root of the number of measurements in each interval.

The two Fidas were calibrated in the field at the start of the campaign using monodisperse (non-absorbing) polystyrene latex spheres (PSLs). Therefore, the (default) optical diameters typically used to report the PSDs obtained with OPCs are diameters of PSLs that produce the same scattered light intensity as the measured dust particles. As in the majority of previous studies (e.g. Fratini et al., 2007; Sow et al., 2009; Shao et al., 2011a; Ishizuka et al., 2014; Dupont et al., 2021), we use optical diameters to analyze the PSDs and their variability throughout most of this contribution. We also compare these "optical diameter"
PSDs with the original Brittle Fragmentation Theory Kok (2011a), where the emitted dust PSD is derived by analogy to the fragmentation of brittle materials such as glass spheres constrained by PSD measurements unharmonized in terms of diameter type. Since dust is aspherical and light-absorbing, we additionally provide a synthesis of our results after transforming our optical diameters into dust geometric diameters assuming a more realistic shape and composition. The geometric or volume- equivalent diameter is the diameter type used in dust modeling and it refers to the diameter of a sphere with the same volume





as the aspherical particle. In this way, our results can also be compared with an updated version of BFT that accounts more realistically for super-coarse dust emission (Meng et al., 2022), and that was constrained with measured PSDs harmonized to dust geometric diameters assuming tri-axial ellipsoids (Huang et al., 2021).

We transform the default PSL diameters into dust geometric diameters following Huang et al. (2021), which involves calculating the theoretical scattered intensities of the PSLs and the aspherical dust. Then, the comparison of both scattered intensities
allows remapping the PSL into dust geometric diameters if both functions are monotonic with diameter. The calculation of the scattered intensity depends to first order on the wavelength of the light beam used in the OPC, the scattering angle range of the OPC's light sensor, and the shape and refractive index of the particles, which are specified and discussed below.

**Wavelength of the light beam and scattering angle:** The Fidas determines the number and size of particles using a polychromatic unpolarized LED light source. Each particle that moves through the measurement volume generates a scattered light
impulse that is detected at an angle of $90 \pm 5°$. Unfortunately, neither the characteristics of the polychromatic light beam of the Fidas, nor the spectral sensitivity of the sensor are provided by the manufacturer. However, the manufacturer provides a software that allows to convert the obtained PSDs with PSLs to PSDs of spherical particles assuming 16 different refractive indices. We used this information, the information on the scattering angle, and the Lorenz-Mie code used in Escribano et al. (2019) to infer a light spectrum that can best reproduce the software conversions between spherical aerosol types. Our optimization problem
was constrained to fit a sum of Gaussian spectra over the wavelength domain. The resulting single-Gaussian optimal spectrum has a center wavelength of 389 nm and a standard deviation of 77 nm. We have therefore used this spectrum to convert the optical PSL diameters to dust geometric diameters. The obtained spectrum is consistent with the apparent bluish LED light of the Fidas.

**Shape:** The sideward scattered intensity depends on particle shape. Since PSLs are spherical, we obtained their single-
scattering properties based on Lorenz-Mie theory. For dust, we assume dust particles are tri-axial ellipsoids, because extensive measurements have found that dust particles are three-dimensionally aspherical (Huang et al., 2021). To quantify dust asphericity, we used an aspect ratio (AR) of 1.46, which is the median AR of the more than 300.000 individual dust particles collected during our campaign and analyzed in the laboratory using Scanning Electron Microscopy (SEM) coupled with Energy-Dispersive X-ray Spectrometry (EDX) (Panta et al., in prep.). We did not perform measurements of the height-to-width
ratio (HWR), so we assume HWR= 0.45, which is the closest value to the global median of 0.4 obtained in Huang et al. (2021). We combined the AR and HWR with the database of shape-resolved single-scattering properties of ellipsoidal dust particles (Meng et al., 2010), after Huang et al. (2021).

**Refractive index:** Our preliminary analyses of the optical properties (Yus-Díez et al., in prep.) and mineralogical composition (González-Romero et al., in prep.) suggest imaginary parts of the refractive index between 0.0015 and 0.002, consistent
with chamber-based re-suspension estimates using Moroccan soil samples in Di Biagio et al. (2019). Here, we use a value of 0.0015 for the imaginary part, and we assume a value of 1.49 for the real part as obtained in Di Biagio et al. (2019) with their Moroccan samples.





In Appendix C, Fig. C1 we confront the obtained geometric diameters with the default optical diameters. Based on our transformation, the optical diameters overestimate the dust diameters between $\sim 0.5$ and $\sim 13\,\mu$m and underestimate them at

finer and coarser sizes due to the combined effects of dust refractive index and asphericity.

At the end of the campaign, the two Fidas were placed at the same height (1.8 m) for inter-calibration. Appendix D describes the corrections applied to FidasU in order to remove the systematic concentration differences between both OPCs.

### 2.2.3 Saltation flux measurements

Time and size-resolved saltation counts were measured with three SANTRI (Standalone AeoliaN Transport Real-time In-

strument) platforms (Etyemezian et al., 2017; Goossens et al., 2018). Two SANTRIs (SANTRI-4 and SANTRI-5 in Fig. 1d) consisted of duplicate optical gate devices (OGDs, Etyemezian et al., 2017) at 5 cm height, single OGDs at 15 and 30 cm heights and a cup anemometer and wind vane at $\sim 1.1$ m height, and measured at 1 s intervals. Saltation counts were recorded in 7 size bins, whose lower and upper diameter limits were calculated from the recorded sensor reference voltage levels. The two bins with, respectively, the smallest and largest diameters were excluded from further analysis due to a large noise level

for the former and an absent upper diameter limit for the latter. On average, the remaining size range extended roughly from 85 to 450 $\mu$m in diameter. A third SANTRI (SANTRI-3 in Fig. 1d) collected data from two OGDs at multi-kHz frequencies, but is not analyzed here. Due to technical issues with SANTRI-5, results presented here will focus on SANTRI-4 using the front one of the two bottom sensors.

## 2.3 Inferred quantities

### 2.3.1 Dynamical parameters characterizing the near-surface boundary layer

Monin-Obukhov similarity theory (Monin and Obukhov, 1954) allows describing the vertical profiles of some variables (e.g. wind speed or temperature) as a function of dimensionless groups. In aeolian erosion studies, $u_*$ is a key parameter that represents the surface wind shear stress. In this study, $u_*$ is calculated from the law of the wall approach, which assumes a logarithmic or pseudo logarithmic form (for non-neutral atmospheric stability conditions) of the mean wind velocity profile

within the surface layer (e.g. Stull, 1988; Arya, 2001; Foken and Napo, 2008; Shao, 2008)

$$\overline{U}(z) = \frac{u_*}{\kappa}\left[\ln\left(\frac{z}{z_0}\right) - \Psi_m\left(\frac{z}{L}\right) + \Psi_m\left(\frac{z_0}{L}\right)\right] \tag{1}$$

where $\overline{U}(z)$ denotes the mean horizontal wind speed at height $z$, $\kappa = 0.4$ is the von Karman constant, $L$ is the Obukhov length, $z_0$ is the aerodynamic roughness length and $\Psi_m = \int_{z_0/L}^{z/L}[1 - \Phi_m(\zeta)]\frac{d\zeta}{\zeta}$, where $\zeta = z/L$ and $\Phi_m$ is the similarity function for momentum.

Here, we use

$$\Psi_m = \begin{cases} -6\frac{z}{L} & \text{if } \zeta > 0 \text{ (Businger et al., 1971; Högström, 1988)} \\ -\ln\left(\frac{(\zeta_0^2+1)(\zeta_0+1)^2}{(\zeta^2+1)(\zeta+1)^2}\right) - 2[\tan^{-1}(\zeta) - \tan^{-1}(\zeta_0)] & \text{if } \zeta \leq 0 \text{ (Benoit, 1977)} \end{cases} \tag{2}$$





with $\zeta = (1 - 19.3z/L)^{1/4}$ and $\zeta_0 = (1 - 19.3z_0/L)^{1/4}$ (Högström, 1988)

The Obukhov length ($L$) can be derived as (Foken and Napo, 2008)

$$L = -\frac{\theta_r u_*^3}{\kappa g \overline{w'\theta'_0}} \tag{3}$$

where $\theta_r$ is a reference potential temperature, $g = 9.81\,\mathrm{m\,s^{-2}}$ is the gravitational acceleration and $\overline{w'\theta'}$ is the surface kine-
matic heat flux. Heat flux ($H = \rho_{air} c_p \overline{w'\theta'}$ with air density $\rho_{air}$ and specific heat capacity of air at constant pressure $c_p = 1004\,\mathrm{J\,kg^{-1}\,K^{-1}}$) can be also estimated from the bulk-aerodynamic formulation for the sensible-heat flux (e.g. Shao, 2008;
Klose et al., 2019)

$$H = \rho_{air} c_p \left(\frac{T_0 - T_r}{r_a}\right) \tag{4}$$

where $T_r$ is the temperature at reference height $z_r$, $T_0$ the soil surface temperature, $r_a = (C_h u_r)^{-1}$ the bulk aerodynamic
resistance between $z_0$ and $z_r$ with $u_r$ the wind at reference height and $C_h = \kappa^2/([\ln(\frac{z}{z_0}) - \Psi_m(\frac{z}{L})][\ln(\frac{z}{z_0}) - \Psi_h(\frac{z}{L})])$ (e.g.
Arya, 2001; Stull, 1988) the bulk heat transfer coefficient, where $\Psi_h = \int_{z_0/L}^{z/L} [1 - \Phi_h(\zeta)]\frac{d\zeta}{\zeta}$, being $\Phi_h(\zeta)$ the similarity function
for sensible heat. Here, we use

$$\Psi_h = \begin{cases} 0.05\ln\left(\frac{z}{z_0}\right) - 7.8\frac{z}{L} & \text{if } \zeta > 0 \text{ (Businger et al., 1971; Högström, 1988)} \\ 0.05\ln\left(\frac{z}{z_0}\right) - 1.9\ln\left[\frac{(\lambda_0+1)}{(\lambda+1)}\right] & \text{if } \zeta \leq 0 \text{ (Benoit, 1977)} \end{cases} \tag{5}$$

with $\lambda = (1 - 11.6z/L)^{1/2}$ and $\lambda_0 = (1 - 11.6z_0/L)^{1/2}$ (Högström, 1988).

Therefore, $\overline{w'\theta'}$, needed for calculating $L$, can be inferred from Eq. 4. We chose $2\,\mathrm{m}$ as the reference height $z_r$, because
at this height we had both temperature and wind measurements. $T_0$ was obtained from radiometer measurements of surface
longwave radiative flux and $\rho_{air}$ was determined from relative humidity and temperature measurements at $0.5\,\mathrm{m}$ height and
pressure at $1.5\,\mathrm{m}$ height, by making use of Tetens' formula (Tetens, 1930) and the ideal gas law (e.g. Stull, 1988).

Applying a linear regression based on Eq. 1 and neglecting $\Psi_m(z_0/L)$, we obtain

$$\overline{U}(z) = m[\ln(z) - \Psi_m] + n \tag{6}$$

where $m$ and $n$ are the slope and intercept of the linear regression. Thus, $u_* = m\kappa$ and $z_0 = \exp(-n/m)$. An iterative proce-
dure was performed to deduce $u_*$, $z_0$ and $L$ for every 15-minute period. This iterative procedure assumes neutral conditions
as a first guess, and then corrects for stability using the expressions shown before. As in previous studies, this procedure was
applied only when wind increased with height and for wind speeds at $2\,\mathrm{m}$ height larger than $\sim 1\,\mathrm{m\,s^{-1}}$ (Marticorena et al.,
2006; Khalfallah et al., 2020). In addition, results were only considered when the difference between the computed and mea-
sured wind profile was less than 10% and when the resulting dimensionless height $\zeta = z_r/L$ was in the range $(-10, 2)$. This
is the range for which Monin-Obukhov theory seems to be valid (Kramm et al., 2013). The relationship between $u_*$ and $z_0$ is
analyzed in Sect. 3.2.



### 2.3.2 Size-resolved flux-gradient dust flux

We estimate the near-surface vertical diffusive flux, $F$, using the flux-gradient method (Gillette et al., 1972). This approach, by analogy with Fick's law for molecular diffusion, assumes that the diffusive dust flux is proportional to the vertical gradient of the local mean dust concentration, $c$, where the dust eddy diffusion coefficient, $K_d$, is the constant of proportionality. Thermal stratification effects are accounted for following the Monin-Obukhov theory (Monin and Obukhov, 1954) through the similarity function for dust $\Phi_d$, that translates into an adjustment of $K_d$. This yields

$$F = -\frac{K_d}{\Phi_d}\frac{\partial c}{\partial z} \tag{7}$$

where $K_d = K_m/Sc_t$ with momentum eddy diffusion coefficient $K_m$ and turbulent Schmidt number $Sc_t$. Similar to Eq. 7, the momentum flux $\langle u'w' \rangle$ can be expressed proportionally to the vertical gradient of the horizontal wind speed, $u$ as

$$\langle u'w' \rangle = -\frac{K_m}{\Phi_m}\frac{\partial u}{\partial z} \tag{8}$$

Assuming that trajectory crossing effects are negligible, which is considered reasonable for particle diameters smaller than 10–20 µm (Csanady, 1963), $K_m$ and $K_d$ are equivalent and lead to $Sc_t = 1$ and $\Phi_m = \Phi_d$. If additionally, a constant momentum flux layer is assumed, then $\langle u'w' \rangle = -u_*^2$. Dividing Eqs. 7 and 8, taking into account these assumptions and substituting from Eq. 1 we obtain the widely-used expression proposed in Gillette et al. (1972)

$$F_n(D_i) = u_* \kappa \frac{c_l^n(D_i) - c_u^n(D_i)}{\ln\left(\frac{z_u}{z_l}\right) - \Psi_m\left(\frac{z_u}{L}\right) + \Psi_m\left(\frac{z_l}{L}\right)} \tag{9}$$

where $c_u^n(D_i)$ and $c_l^n(D_i)$ are the number concentrations of dust particles with diameter $D_i$ measured by the two Fidas at $z_u = 3.5\,\text{m}$ and $z_l = 1.8\,\text{m}$ in bin $i$. Note that the FidasU concentrations include the systematic corrections derived from the intercomparison of the two Fidas by the end of the campaign (See Appendix D).

Eq. 9 is applied to each of the sixty-three size intervals of the Fidas using 15-min average concentrations. Thus, the total number and mass diffusive fluxes are obtained by summing over all size bins. The mass flux in each bin is inferred from its respective number flux as

$$F_m(D_i) = F_n(D_i)\frac{1}{6}\rho_d\pi D_i^3 \tag{10}$$

where $D_i=\sqrt{d_{max} * d_{min}}$ is the mean logarithmic diameter in bin number $i$, $d_{max}$ and $d_{min}$ are the minimum and maximum particle diameters of bin $i$, $F_n(D_i)$ and $F_m(D_i)$ are the 15-min averaged number and mass diffusive fluxes with diameter $D_i$ and $\rho_d$ is the dust particle density, which we assume to be $2500\,\text{kg m}^3$ (Fratini et al., 2007; Reid et al., 2008; Kaaden et al., 2009; Sow et al., 2009; Kok et al., 2021). All diameters can be either the default optical or the obtained geometric ones.

All calculations are performed using the original size bins of the Fidas (63 bins ranging from 0.2 µm to 19.1 µm). However, such a high bin resolution leads to substantial noise in the coarse and super-coarse bins of the mass PSDs. Therefore, we integrated the 63-bin PSDs into 16 bins to represent both the mass concentration and mass diffusive flux PSDs. The size-resolved diffusive flux can exhibit positive and negative values, with the former representing an upward (net emission) flux and





the latter a downward (net deposition) flux. Well-developed erosion conditions are normally characterized by positive fluxes. For this reason, in this study flux PSDs containing any negative value in any of the integrated mass and number bins where $D_i > 0.42\,\mu\mathrm{m}$ (to avoid the anthropogenic aerosol influence, see Sect. 3.3.1) have been excluded.

The calculation of the uncertainty of each 15-min size-resolved diffusive flux is described in Appendix E. In Sect. 3.3 we analyze the 15-min diffusive flux PSDs averaged over $u_*$ intervals along with their uncertainties. The average total uncertainty for each $u_*$ interval is calculated as the square root of the quadratic sum of the standard error and the average diffusive flux uncertainty within each $u_*$ interval. The average diffusive flux uncertainty for each $u_*$ ($\sigma_{F(D_i)_{avg}}$) is calculated as:

$$\sigma_{F(D_i)_{avg}} = \sqrt{\sum \sigma^2_{F(D_i)_j}}/N \tag{11}$$

where $\sigma^2_{F(D_i)_j}$ is the uncertainty of each 15-min size-resolved diffusive flux in the $u_*$ interval, $N$ is the number of 15-min measurements in the $u_*$ interval, $i$ is the size bin and $j$ is the measurement index within each $u_*$ interval.

### 2.3.3 Saltation flux and sandblasting efficiency

The total streamwise saltation flux, $Q$ is defined as the vertical integral of the height-dependent streamwise saltation flux densities derived from the measured saltation counts. $Q$ was calculated as described in Klose et al. (2019) assuming an exponentially decreasing vertical profile of saltation flux density and using least-squares curve fitting for the three measurement heights. Profiles with coefficients of determination $R^2 < 0.5$ were excluded. Of the remaining profiles, more than 99% have $R^2 > 0.95$ and more than 98% have $R^2 > 0.99$. Sandblasting efficiency, $\alpha$, is defined as the ratio of total vertical (diffusive) dust flux to horizontal (saltation) flux in mass, $\alpha = F/Q$. When calculating $\alpha$ we excluded the vertical flux measurements in which either the net flux was negative or any of the 15 merged mass and number bins where $D_i > 0.42\,\mu\mathrm{m}$ (to avoid the anthropogenic aerosol influence, see in Sect. 3.3.1) was negative.

### 2.4 Estimation of the size-resolved dry deposition flux

Our focus is to understand the dust PSDs and their variability covering a wide range of dust sizes including well above $10\,\mu\mathrm{m}$. Therefore we cannot neglect the potential influence of dry deposition. In order to better understand the obtained concentration and flux-gradient dust flux PSDs, we estimate the dry deposition flux ($F_{dep}$) for each bin as:

$$F_{dep}(D_i) = v_{dep}(D_i)c_{int}(D_i) \tag{12}$$

where $v_{dep}$ is the dry deposition velocity, $c_{int}$ is the concentration at the intermediate height between the two Fidas, and $D_i$ the diameter of each bin $i$. The dry deposition velocity is typically parameterized using a resistance model that includes gravitational settling ($v_g$) and a series of resistors accounting for the aerodynamic ($R_a$) and surface ($R_s$) resistances that can be implemented in multiple forms. We used the same form as Fernandes et al. (2019) in their modeling study.

$$v_{dep}(D_i) = \frac{1}{R_a + R_s(D_i) + R_a R_s(D_i) v_g(D_i)} + v_g(D_i) \tag{13}$$





where $R_a = \ln(\frac{z_{int}}{z_0})/(\kappa u_*)$ represents the turbulent transfer close to the surface, $z_{int}$ is the intermediate height between
the two Fidas, and $z_0$ the aerodynamic roughness length as derived in Sect. 2.3.1. The surface or quasi-laminar resistance
$R_s = [u_*(S_c^{-2/3}+10^{-3/S_t})]^{-1}$ accounts for losses by Brownian motion, and inertial impaction; $S_c = \nu/D_g(D_i)$ is the Schmidt
number and $S_t = u_*^2 v_g(D_i)/(g\nu)$ the Stokes number, where $D_g(D_i) = \kappa T C_c/(3\pi\rho_{air}\nu D_i)$ is the Brownian diffusivity, $\kappa$
is the Boltzmann constant, $T$ is the air temperature at $1\,\mathrm{m}$ height, $C_c$ is the Cunningham slip correction factor and $\nu =$
$1.45 \cdot 10^{-5}\,\mathrm{m^2\,s^{-1}}$ is the air kinematic viscosity. The settling velocity $v_g(D_i)$ is calculated for each size bin as $v_g(D_i) =$
$C_c\sigma_{pa}gD_i^2/(18\nu)$ where $\sigma_{pa} = (\rho_d - \rho_{air})/\rho_{air}$ is the particle-to-air density ratio.

## 3  Results and discussion

### 3.1  Overview of the atmospheric conditions and dust events during the campaign

Times series of measured atmospheric conditions and near-surface dust concentrations are displayed in Fig. 2; $u_*$ and atmospheric stability, along with saltation and diffusive fluxes are displayed in Fig. 3. As expected, the diurnal cycles of temperature
and relative humidity are anti-correlated (Fig. 2b), and temperature inversions (Fig. 2a) along with atmospheric stability (Fig. 3b) are prevalent during nighttime. Temperature at $2\,\mathrm{m}$ ranges from slightly less than $20\,^\circ\mathrm{C}$ during the night to up to $\sim 40\,^\circ\mathrm{C}$ during the day, and surface relative humidity ranges from as low as $6\,\%$ during the day to up to $\sim 65\,\%$ during the night. There is a shift after September 14th, with substantial increases in temperature and decreases in relative humidity, with the exception of September 17th, when relative humidity appears to be temporarily high.

The diurnal cycles of surface wind (Fig. 2d) and $u_*$ (Fig. 3a) along with the associated cycles of saltation and diffusive fluxes (Figs. 3c, 3d and 3e) and dust concentration (Figs. 2f and 2g) are generally associated to the diurnal cycle of solar heating. In the early morning, as the surface starts to warm and releases turbulent sensible heat, the lower atmosphere becomes unstable. As the day evolves, momentum is mixed downward from the stronger winds aloft increasing wind speed and $u_*$, while stability progressively tends towards neutrality (Fig. 3b). Winds are generally channelled through the valley, broadly parallel to the Drâa river, alternating between two opposite and preferential wind directions, centered around $80\,^\circ$ and $240\,^\circ$ (Fig. 2e). (In Appendix B, Fig. B1 depicts the distribution of wind direction and $u_*$ during the campaign.) We refer to the dust events associated to these recurring diurnal cycles as "regular" events, for which maximum winds at $10\,\mathrm{m}$ can reach 15-min average values up to $\sim 11\,\mathrm{m\,s^{-1}}$ (Fig. 2d). From September 22th to 25th winds remain relatively calm, and after the 25th diurnal cycles are less marked and dust events are more intermittent and short-lived.

In addition to these regular events, we also captured two strong cold pool outflows (hereafter referred to as "haboob" events) in the evening of September 4th and in the afternoon of September 6th, both marked with a red "H" in Figs. 2 and 3. Cold pool outflows result from density currents created by latent heat exchange of evaporating rain in deep convective downdrafts. The arrival of sharply-defined dust walls, caused by the gust fronts at the leading edge of the outflow winds, were not only directly witnessed by the field campaign team, but can be also clearly detected in the measurements. In the supplemental material we provide a 1-minute frequency time-lapse video recorded from the Fidas location during September 6th, which clearly shows the arrival of the haboob in the afternoon. Both haboob events are characterized by the highest $10\,\mathrm{-m}$ winds recorded during



**Figure 2.** Time series (UTC) of 15-min average (a) temperature (°C) at 1, 2, 4 and 8 m, (b) relative humidity (%) and temperature (°C) at 0.5 m, (c) pressure (hPa) at 1.5 m, (d) mean wind speed (m s$^{-1}$) and (e) mean wind direction (°) at 0.4, 0.8, 2, 5 and 10 m, (f) FidasL (1.8 m) particle concentrations in number $c_l^n$ (# m$^{-3}$) and (f) in mass $c_l^m$ (µg m$^{-3}$). In (e) and (f) total concentrations are represented as lines (left y-axis) whereas size-resolved concentrations are shown as colour contours (right y-axis) in the original size bin resolution. Vertical grey lines in (a-d) highlight periods for which $u_*$ is above 0.15 m s$^{-1}$. Horizontal grey lines in (e) highlight wind directions for which $u_*$ is above 0.15 m s$^{-1}$ (Fig. 3).



the campaign (15-min averages of $\sim$11.5 and $\sim$14 m s$^{-1}$, respectively) and unusually fast changes in atmospheric conditions with values consistent with previous haboob studies (Miller et al., 2008): sudden increases in wind speed, decreases in 2-m temperature of $\sim$8–9 °C, increases in relative humidity of $\sim$24–32 % and a rise of $\sim$2 hPa in surface pressure (Fig. 2c).

During these events, precipitation was not detected by our rain gauge, but during the night of September 6th there was water flowing downriver, which caused flooding of large areas in the vicinity of our lake on the next day (not affecting the lake itself), suggesting that heavy showers occurred over the mountain range to the north of our location (Fig. 1c).

Dust concentration (Figs. 2f and 2g) exhibits peaks of varying intensity about every $\sim$1–2 days, consistent with the wind speed and $u_*$ patterns. Number and mass concentrations were $5 \cdot 10^7$ # m$^{-3}$ and 1243 µg m$^{-3}$ on average, respectively, and

there were 10 days when the 15-min dust mass concentration exceeded $10^4$ µg m$^{-3}$. As expected for dust, the number concentration was dominated by fine particles and the mass concentration by coarse and super-coarse dust. Dust concentration is generally correlated with saltation (Fig. 3c) and diffusive fluxes (Figs. 3d and 3e), with the notable exception of an event that extends over the evening of September 17th and the morning of the 18th. During this event, concentrations reached values that are among the highest recorded during the campaign (Figs. 2f and 2g), although winds are low (Fig. 2d), saltation is absent

(Fig. 3c), and diffusive fluxes are negative (note that negative fluxes are not represented in Fig. 3d and 3e), which implies that dust was transported from elsewhere and deposited, but not emitted from our site. Given that convective storms were spotted from a distance during that evening and the event is characterized by high relative humidity values (Fig. 2b), we hypothesize that those highly dust-loaded air masses that slowly and persistently reached our site were generated by precedent haboob activity upwind.

Also, during the campaign, we detected the presence of anthropogenic aerosols with diameters below $\sim$0.4 µm, whose influence is most visible when winds are weak and mass concentrations low (see Appendix F, Fig. F1), consistent with measured optical properties analyzed in a companion contribution (Yus-Díez et al., in prep.). This is particularly evident between September 8th and 10th, when low wind comes from the east (i.e. from M'Hamid). Such anthropogenic aerosol influence at the lower end of the measured PSD range is further evidenced and discussed in Sect. 3.3.1.

Saltation and diffusive fluxes are highly correlated and occur regularly throughout the campaign, peaking typically between noon and 18 UTC in accordance with maximum surface winds and $u_*$. Averaged over 15 minutes, saltation is typically detected when $u_*$ is $\sim 0.15$ m s$^{-1}$ or above, which happens nearly everyday. $u_*$ shows peaks of up to $\sim 0.4$ m s$^{-1}$ during regular events, and reaches up to $\sim 0.6$ m s$^{-1}$ during the haboob event that occurred on the afternoon of September 6th (Fig 3a). Wind erosion occurs mostly under unstable or close to neutral atmospheric conditions (Fig. 3b). For $u_*$ above $0.15$ m s$^{-1}$, the 15-min

average of total vertical diffusive flux in terms of number (mass) is on average $3.4 \cdot 10^6$ # m$^{-2}$ s$^{-1}$ (175 µg m$^{-2}$ s$^{-1}$), reaching a maximum value of $8.4 \cdot 10^7$ # m$^{-2}$ s$^{-1}$ (5116 µg m$^{-2}$ s$^{-1}$) on September 6th.

## 3.2 Characterization of saltation and sandblasting efficiency

Figs. 4a, 4b and 4c display the diffusive flux, saltation flux and sandblasting efficiency against $u_*$. We use coincident 15-min data between saltation and diffusive flux, and only when the diffusive flux is positive in all dust size bins above 0.4 µm (see

Sect. 3.3.1 for more details). The points corresponding to the haboobs on 4th and 6th September are depicted with squares and



**Figure 3.** Time series (UTC) of 15-min averaged (a) friction velocity $u_*$ ($\mathrm{m\,s^{-1}}$), (b) atmospheric stability represented by z/L, where z is the reference height $2\,\mathrm{m}$, (c) saltation flux ($\mathrm{g\,m^{-1}\,s^{-1}}$), (d) bulk and size-resolved diffusive flux in number ($\#\,\mathrm{m^{-2}\,s^{-1}}$) and (e) bulk and size-resolved diffusive flux in mass ($\mathrm{\mu g\,m^{-2}\,s^{-1}}$). Grey areas in (a)-(c) highlight times with $u_* > 0.15\,\mathrm{m\,s^{-1}}$. Data gaps in $u_*$, atmospheric stability, and diffusive fluxes result from limits in the applicability of the law of the wall method. The size resolved diffusive fluxes are shown in the integrated size bin resolution. Only the bulk and size-resolved diffusive fluxes that are positive are represented.





triangles, respectively. Regression curves of the form $a \cdot u_*^b$ are also represented for different ranges of $u_*$. The 95% confidence intervals of the parameters of each regression curve are shown in Appendix G, Table G1. The diffusive flux ranges mostly between $\sim 10^1$ and $\sim 10^3 \, \mu g \, m^{-2} \, s^{-1}$ and the power law exponent $b$ increases when small values of $u_*$ are not considered, being 3.35 for $u_* > 0.1 \, m \, s^{-1}$ and 4.04 for $u_* > 0.2 \, m \, s^{-1}$ (Fig. 4a). The obtained exponents are within the range shown in

Ishizuka et al. (2014) (their Fig. 5), where $b$ varies between approximately 3 and 6 across different data sets gathered from the literature (Gillette, 1977; Nickling, 1983; Nickling and Gillies, 1993; Nickling et al., 1999; Gomes et al., 2003a; Rajot et al., 2003; Sow et al., 2009), likely due to differences in soil type and soil-surface conditions.

The saltation flux ranges between about $10^{-1}$ and $10^2 \, g \, m^{-1} \, s^{-1}$. The power law exponent $b$ is slightly higher than that obtained for the diffusive flux, and it is also larger for the upper $u_*$ range compared to the lower one, with $b = 3.66$ for

$u_* > 0.1 \, m \, s^{-1}$ and $b = 4.85$ for $u_* > 0.2 \, m \, s^{-1}$ (Fig. 4b). These values are larger than that reported in Gillette (1977) for most soils ($b \approx 3$). In comparison with Alfaro et al. (2022) (their Fig. 4), where data of two major dust field campaigns (JADE and WIND-O-V) are re-analyzed, we obtain larger saltation fluxes for similar ranges of $u_*$. For $u_* \approx 0.25\text{–}0.45 \, m \, s^{-1}$, our 15-min saltation fluxes vary between $10^0$ and $10^2 \, g \, m^{-1} \, s^{-1}$ while the 1min (16min) measurements from the JADE (WIND-O-V) campaign vary between $10^{-1}$ and $10^1 \, g \, m^{-1} \, s^{-1}$. Using the same instrument (SANTRI) as in our study, Klose et al.

(2019) reported a maximum 1-min saltation flux of almost $10^1 \, g \, m^{-1} \, s^{-1}$ for $u_* > 0.8 \, m \, s^{-1}$, approximately one order of magnitude smaller than our 15-min maximum values occurring during the haboobs for smaller $u_*$. Comparison of the height-dependent saltation flux obtained with SANTRI4 with that from the co-located MWAC sampler (not shown) confirmed that both are largely consistent, with SANTRI4 tending to record slightly higher fluxes. This is in qualitative agreement with the comparison of saltation measurement devices from Goossens et al. (2018).

The intensity of saltation impacts the aerodynamic roughness length due to momentum absorption by the saltating particles (Owen, 1964; Gillette et al., 1998). Figure 5 displays the relationship between aerodynamic roughness length and $u_*$ under saltation conditions, that is 15-min values with a positive saltation flux, in our site. We only use the values in which at the same time $u_* > 0.15 \, m \, s^{-1}$, so there is no doubt of well-developed erosion conditions. The aerodynamic roughness length shows quite a lot of scatter, particularly for $u_*$ below $0.2 \, m \, s^{-1}$, ranges mostly between $10^{-5}$ and $10^{-4} \, m$ and increases with $u_*$. This

increase was also observed in Dupont et al. (2018) and Field and Pelletier (2018), although we obtain roughness lengths about one order of magnitude smaller that are consistent with values obtained in other playas and smooth surfaces (Marticorena et al., 2006). We also observe that the roughness length is sensitive to wind direction. For example roughness lengths can reach about one order of magnitude higher values for wind directions $135\text{–}180\,^\circ$ and $315\text{–}360\,^\circ$, the latter one close to the alignment of our instruments. There are also differences, albeit relatively small, between the two predominant wind directions, $225\text{–}270\,^\circ$ and

$45\text{–}90\,^\circ$ (Fig. 5). Using the relationship $z_0 = C_c \cdot u_*^2 / g$, originally derived by Charnock (1955) for water surfaces, but that can be applied for sand and snow surfaces (Owen, 1964; Chamberlain, 1983), we obtain $C_c = 0.02$ when taking into account all data, although the dispersion is very high and $R^2$ very low. This value coincides with that obtained by Owen (1964) and that derived in Dupont et al. (2018) for some of the wind erosion events during the WIND-O-V 2017 Experiment. Smaller values of $C_c = 0.007$ and $0.004$ and a higher $R^2$ are obtained, when considering separately the predominant wind directions $225\text{–}270\,^\circ$

and $45\text{–}90\,^\circ$, respectively (Fig. 5).



**Figure 4.** (a) Diffusive flux ($\mu$g m$^{-2}$ s$^{-1}$) versus friction velocity $u_*$ (m s$^{-1}$); (b) Saltation flux (g m$^{-1}$ s$^{-1}$) versus $u_*$ (m s$^{-1}$); (c) Sandblasting efficiency (m$^{-1}$) versus $u_*$ (m s$^{-1}$); (d) Sandblasting efficiency (m$^{-1}$) versus saltation flux (g m$^{-1}$ s$^{-1}$). Colours represent wind direction (°). The points shown in all panels correspond to the 15-min values in which there is a simultaneous net positive diffusive flux and saltation flux, and when the diffusive flux is positive in all size bins above 0.4 $\mu$m. Squares (triangles) are used to identify the values corresponding to haboobs on 4th (6th) September. The lines in (a)-(d) represent the regression curves of the form $a \cdot u_*^b$ for $u_* > 0.1$ m s$^{-1}$ (blue) and for $u_* > 0.2$ m s$^{-1}$ (orange). The coefficient of determination of each regression curve is shown in its respective graph and the 95% confidence intervals of $a$ and $b$ are reported in Table G1.

The sandblasting efficiency ranges between about $10^{-6}$ and $10^{-3}$ m$^{-1}$, although most values are concentrated between $10^{-5}$ and $10^{-4}$ m$^{-1}$ (Fig. 4c). These results are similar to those obtained in Gomes et al. (2003a) (corresponding to a soil nominally of silt loam texture in Spain), Gomes et al. (2003b) (for a sandy soil with a very low clay and silt content in Niger), and the results of the soils 4, 5 (classified as sandy) and 9 (clay) reported in Gillette (1977). However, our values are on the lower end of the range reported in Gillette (1977) and Alfaro et al. (2022), where most sandblasting efficiencies are above $10^{-4}$ m$^{-1}$.






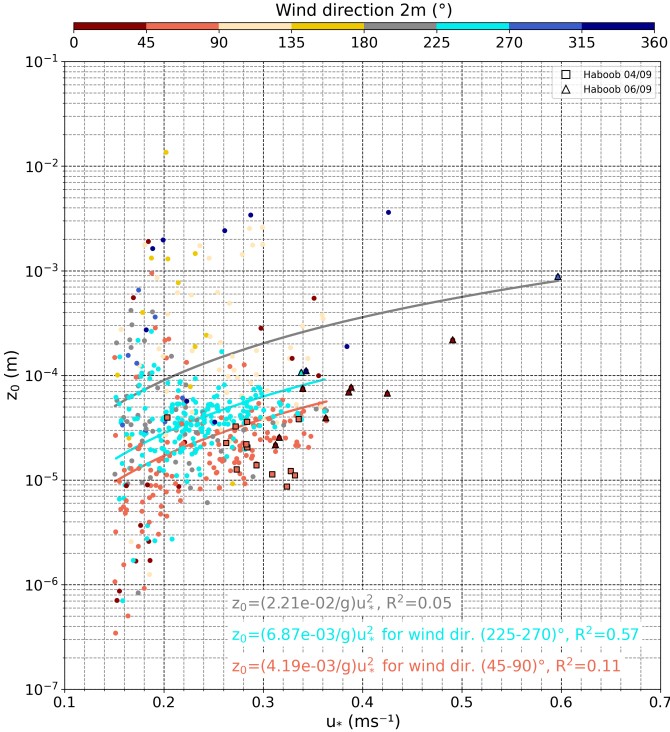

**Figure 5.** Relationship between 15-min averages of surface roughness length ($z_0$) and friction velocity ($u_*$) under wind erosion conditions. Colors indicate wind direction at 2 m height. The lines represent the regression curves of the form $C_c \cdot u_*^2/g$ for all the data (grey) and for wind directions between 45–90 ° (orange) and 225–270 ° (blue). The resulting fit-parameters and coefficients of determination are given in the figure. Squares (triangles) are used to identify the values corresponding to haboobs on 4th (6th) September.

The sandblasting efficiency tends to decrease slightly with increasing $u_*$, but R$^2$ is very small. There is some dependency of the sandblasting efficiency upon wind direction and the $u_*$ range considered. For example, sandblasting efficiencies are higher under south-easterly winds (135–180 °) than under the dominant wind directions (45–90 ° and 225–270 °). The exponent of the power law considering all the wind directions is negative and becomes slightly more negative considering only larger $u_*$

($b = -0.31$ for $u_* > 0.1 \,\mathrm{m\,s^{-1}}$ and $b = -0.81$ for $u_* > 0.2 \,\mathrm{m\,s^{-1}}$). This exponent also changes between predominant wind directions (See Appendix G, Figs. G1c and G2c) but the amount of data is rather small, shows significant scatter, and R$^2$ is small. Interestingly, some of the lowest sandblasting efficiency values (around $10^{-5}$ m) are obtained during the haboob events, at least in part due to an enhanced depletion of coarse and super-coarse particles in the diffusive fluxes during the haboob events as discussed in Sect. 3.3.3.

There is a more robust decrease in sandblasting efficiency with increasing saltation fluxes (Fig. 4d) for all $u_*$ ranges, particularly for $u_* > 0.2 \,\mathrm{m\,s^{-1}}$, which is also evident in each of the two dominant wind directions (See Appendix G, Figs. G1d and G2d). Such decreases of the sandblasting efficiency with increasing $u_*$ and saltation flux are also found in Alfaro et al. (2022) using data from the JADE and WIND-O-V field campaigns. To explain this result, Alfaro et al. (2022) suggests that





the proportion of emitted fine (coarse) particles produced by sandblasting should increase (decrease) with Q due to enhanced
fragmentation of aggregates, which leads to lower sandblasting efficiencies. We discuss in Sect. 3.3.4 a variety of potential
mechanisms to explain the variations in the diffusive flux PSD with $u_*$ that contribute to the decrease in sandblasting efficiency
with increasing $u_*$.

All in all, our results highlight the prominence of saltation in our site, which produces strong diffusive fluxes despite the
relatively low sandblasting efficiencies. These features are consistent with the measured surface sediment properties. On the one
side, L'Bour is surrounded by small dunes with a minimally dispersed volume median diameter of $132.2\,\mu m$ and a considerable
amount of saltators below $100\,\mu m$ (See Appendix A, Fig. A1), which translates into rather optimal saltation conditions. For
instance, saltation can be detected even when $u_* < 0.15\,m\,s^{-1}$ based on 15-min averages (Fig.4b). During such situations,
saltation is typically intermittent during the 15-min period, hence instantaneous $u_*$ threshold values should be higher, and
more consistent with the minimum saltation thresholds ($\sim 0.2\,m\,s^{-1}$) that occur for particle sizes of $\sim$75–100 $\mu m$ (Iversen and
White, 1982; Shao and Lu, 2000). On the other side, the low sandblasting efficiencies are attributed to the hard-crusted paved
sediment that constitutes the surface of the ephemeral lake.

## 3.3 Understanding the dust PSD at emission and its variability

In this section, we analyse variations in the dust PSD and we discuss the potential mechanisms that control such variations,
after identifying and removing any potential anthropogenic aerosol influence. We then compare our PSDs with BFT (Kok,
2011a; Meng et al., 2022). To obtain a comprehensive view of the PSDs, we study the number and mass normalized and
non-normalized concentration (Figs. 6 and 7) and diffusive flux PSDs (Figs. 8 and 9). When we refer to dust concentrations,
we refer to concentrations from FidasL. The results from FidasU are analogous and provided in Appendix H. We consider all
available measurements covering the full range of $u_*$ when it comes to concentration PSDs, but we only consider diffusive
flux PSDs when $u_* > 0.15\,m\,s^{-1}$, i.e. well-developed erosion conditions, and when the flux is positive in all size bins with
$D_i > 0.4\,\mu m$ (this minimum size is taken to avoid any anthropogenic aerosol contamination as discussed in Sect. 3.3.1). To
facilitate the analysis of results, Figs. 6– 9 group the PSDs into $u_*$ intervals, type of event (regular versus haboob events), and
wind direction (for the sake of simplicity we only show two $180\,^\circ$ wind direction sectors to the east and west of the alignment
between the Fidas and the $10\,$-m tower, as shown in Fig. 1d. Our analysis did not show any clear effect of atmospheric stability
independent of $u_*$ upon the PSD in agreement with (Dupont, 2022), and in contrast to some recent studies (Khalfallah et al.,
2020; Shao et al., 2020). Therefore it is not further explored below.

### 3.3.1 Identification and removal of the anthropogenic aerosol influence

The analysis of the number PSDs evidences the influence of non-geogenic (anthropogenic) particles for diameters $< 0.4\,\mu m$.
The number concentration PSDs show a sharp increase of particles with diameters $< 0.4\,\mu m$ during regular events that is
particularly evident for small $u_*$ (Figs. 6a and 6b). This feature tends to diminish and even disappear with increasing $u_*$ in
the number concentration PSD, which demonstrates its little dependence upon wind erosion. It also disappears in the number
flux (Figs. 8a and 8b), which further confirms the transport, and not the emission, of small anthropogenic particles in our



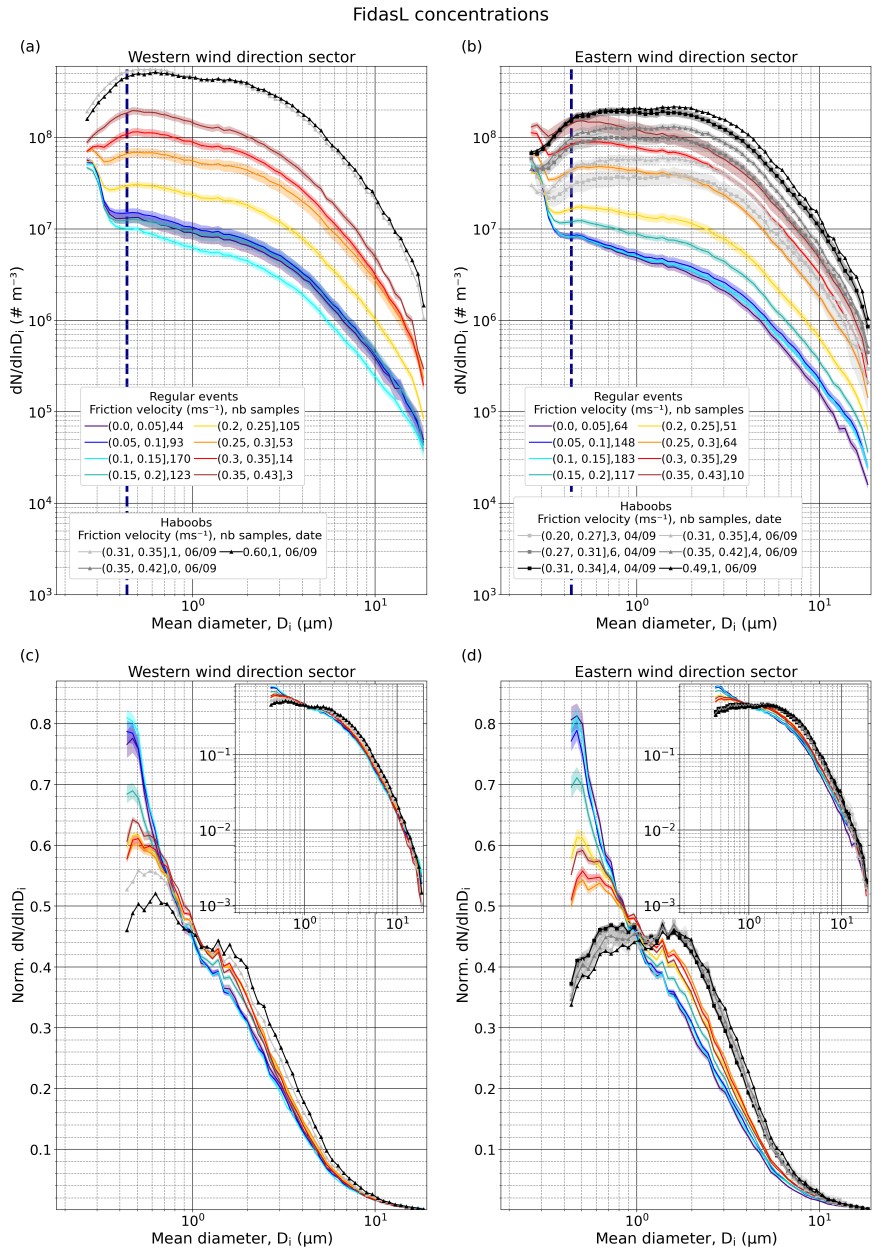

**Figure 6.** Average size-resolved particle number concentration, $dN/dlnD$ ($\#\,\mathrm{m}^{-3}$), for different $u_*$ intervals, type of event (regular or haboob) and wind directions in the range 150–330$^\circ$ (a) and 330–150°(b); The number of available 15-min average PSDs in each $u_*$ interval is indicated in the legend; (c-d) same as (a-b), but normalized ($Norm.\ dN/dlnD$) after removing the anthropogenic mode (normalization from 0.42 to 19.11 μm). Insets show the same data, but with logarithmic ordinate axis-scaling. Shaded areas around the lines depict the standard error. The shown PSDs were obtained from FidasL. In (a) and (b) the dark blue dashed line marks the end of the anthropogenic mode (mean diameter $D_i = 0.44\,\mu m$). Data are shown using original size bin resolution, but first three bins are not represented as Fidas is considered efficient from the fourth one.

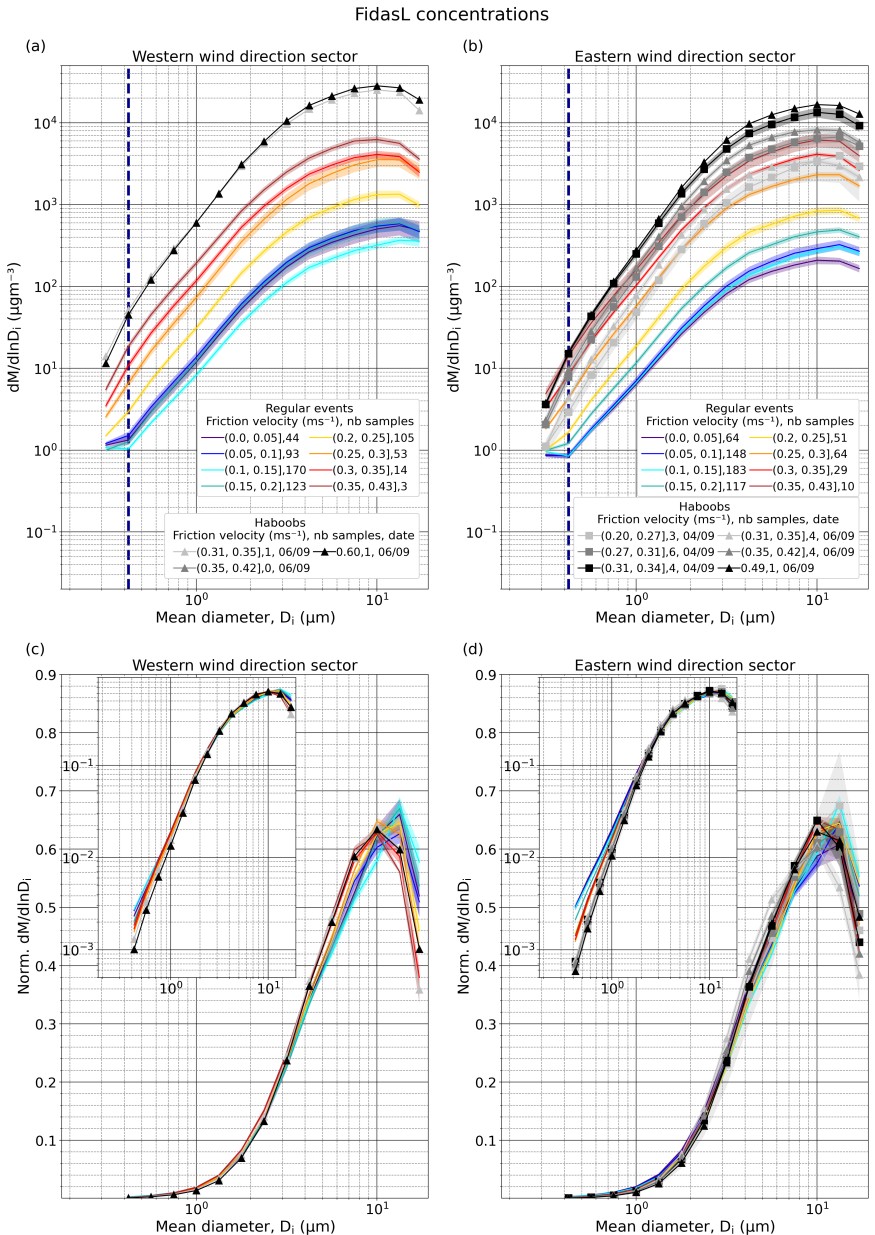

**Figure 7.** Average size-resolved particle mass concentration, $dM/dlnD$ (µg m$^{-3}$), for different $u_*$ intervals, type of event (regular or haboob) and wind directions in the range 150–330 $^\circ$ (a) and 330–150°(b); The number of available 15-min average PSDs in each $u_*$ interval are indicated in the legend; (c-d) same as (a-b), but normalized ($Norm.\ dM/dlnD$) after removing the anthropogenic mode (normalization from 0.37 to 19.11 µm). Insets show the same data, but with logarithmic ordinate axis-scaling. Shaded areas around the lines depict the standard error. The shown PSDs were obtained from FidasL. In (a) and (b) the dark blue dashed line marks the end of the anthropogenic mode (mean diameter $D_i = 0.42$ µm). In this case, the original size resolution of FidasL has been reduced by integrating 4 consecutive bins except for the last one that contains three, resulting in 16 bins. First integrated bin is not represented as Fidas is considered efficient from the second one.





measurement site. This result is further confirmed in companion papers based upon the analysis of airborne samples with electron microscopy (Panta et al., in prep.) and measurements of optical properties (Yus-Díez et al., in prep.); it is also consistent with the anthropogenic sulphate and carbonaceous particle mode detected at Tinfou ($\sim$50 km northeast of L'Bour, beyond the

mountain range and the enclosed desert basin) during the SAMUM field campaign (Kaaden et al., 2009; Kandler et al., 2009).

Compared to regular events, haboob events show markedly less anthropogenic influence (Fig. 6b). We hypothesize this is due to the fresher air masses (carrying less background anthropogenic aerosols) within the cold pool outflows from the convective storms originated in the vicinity of our measurement location.

The analysis of the PSD evolution with $u_*$ shows that the influence of anthropogenic aerosol upon the number concentration

is negligible for diameters $> 0.4\,\mu$m. We note that similar potentially anthropogenic features can be appreciated around 0.3 $\mu$m in PSDs from other wind erosion studies such as in Sow et al. (2009) (their Fig. 8) and Fratini et al. (2007) (their Fig. 5). In this study, in order to avoid any anthropogenic aerosol contamination (particularly for low $u_*$), our normalized PSDs shown in linear and logarithmic scales in Figs. 6c-d, 7c-d, 8c-d and 9c-d consider only diameters $> 0.4\,\mu$m.

### 3.3.2 Differences between concentration and diffusive flux PSDs and their dependencies upon $u_*$ and wind direction

The non-normalized number (Figs. 6a and 6b) and mass concentration PSDs (Figs. 7a and 7b) show the expected strong scaling of concentration with $u_*$ for all size bins, where the number is dominated by fine dust and the mass by coarse and super-coarse dust. For equivalent $u_*$ intervals, concentrations are higher when the wind comes from the western direction sector. The normalized number PSDs (Figs. 6c and 6d) further depict how the shape of the concentration PSD depends upon $u_*$ and wind direction. Overall, there is a relative decrease in sub-micron dust particles and a relative increase in super-micron particles,

especially, around 1.5–2 $\mu$m, with increasing $u_*$, from calm (purplish and blueish lines) to well-developed erosion conditions (yellow, orange and reddish lines). However, it can be subtly observed that for u* intervals above $0.25\,\mathrm{m\,s^{-1}}$ during regular events (orange, red and dark red lines) the fraction of sub-micron (super-micron) particles slightly increases (decreases) with increasing $u_*$, which is even more evident for the eastern wind direction sector. Also for these cases (orange, red and dark red lines), the number fraction of submicron particles is higher when winds come from the western wind direction sector (maxima

at 0.6–0.7) than from the eastern wind direction sector (maxima at 0.5–0.6) .

The normalized mass concentration PSDs (Fig. 7c and 7d) provide further insights into the dependencies of the concentration PSD upon $u_*$. During regular events, the mass fraction of coarse particles with diameters of approximately 4–10 $\mu$m tends to increase and that of super-coarse particles with diameters $> 10\,\mu$m tends to decrease as $u_*$ increases. The peak of the mass PSD, which appears in the super-coarse fraction, tends to shift towards smaller diameters as $u_*$ increases. These features are

broadly similar for both wind direction sectors.

Figs. 8 and 9 depict the diffusive flux PSDs in terms of number and mass, respectively. The PSDs in these figures include the uncertainty (adding both the standard error and the average random uncertainty derived in Appendix D) for each $u_*$ range. For the sake of figure clarity, the uncertainty is shown only for regular events. We provide in Appendix I similar figures including the uncertainties for each $u_*$ range associated to the haboob events (Figs. I1 and I2). We also provide the diffusive flux PSDs

with uncertainties only accounting for standard errors (Figs. I3 and I4).

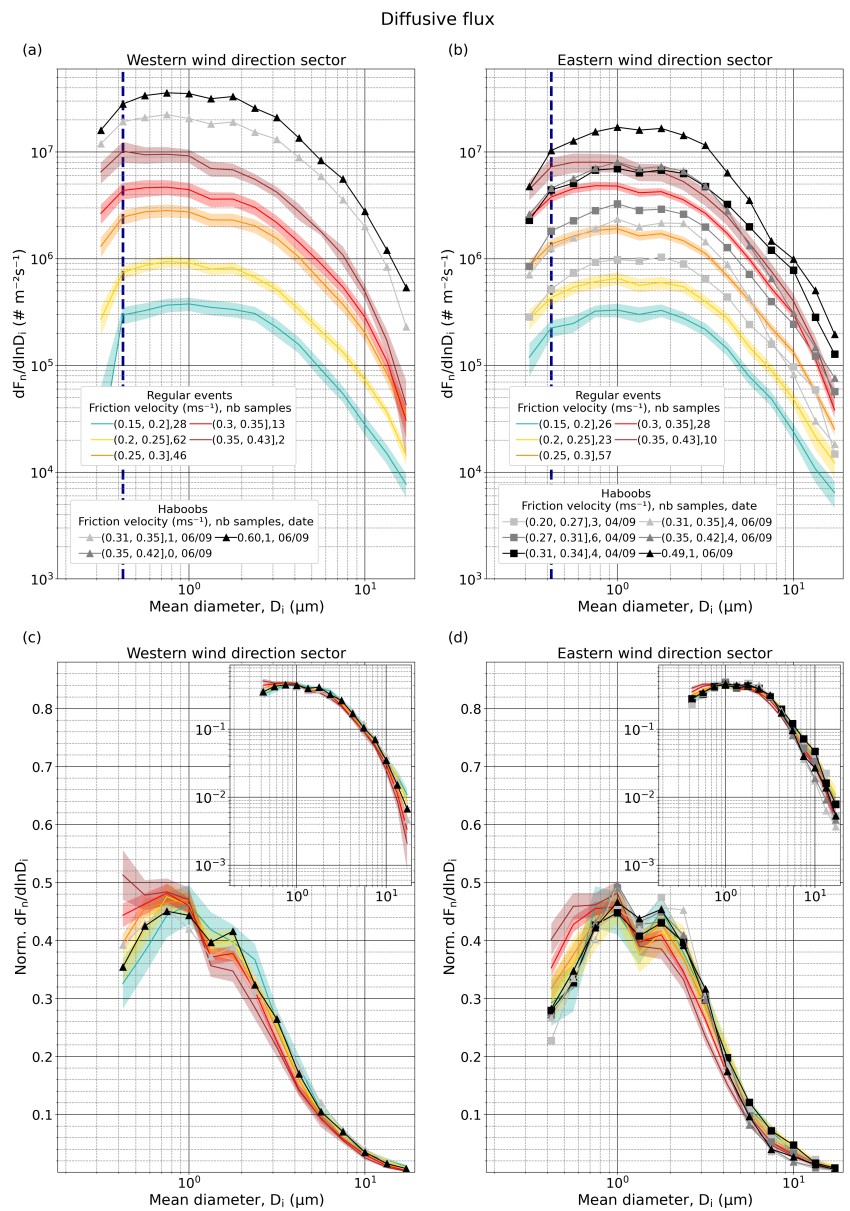

**Figure 8.** Average size-resolved number diffusive flux, $dF_n/dlnD$ ($\#\,\mathrm{m}^{-2}\,\mathrm{s}^{-1}$), for different $u_*$ intervals, type of event (regular or haboob) and wind directions in the range 150–330° (a) and 330–150°(b); The number of available 15-min average PSDs in each $u_*$ interval are indicated in the legend. Only the samples where flux is positive in all the diameter bins above the anthropogenic mode (as discussed in Sect. 3.3.1) have been selected; (c-d) same as (a-b), but normalized (Norm. $dF_n/dlnD$) after removing the anthropogenic mode (normalization from 0.37 to 19.11 µm). Insets show the same data, but with logarithmic ordinate axis-scaling. Shaded areas around the lines of the regular events PSDs depict the combination of random uncertainty and standard error. In (a) and (b) the dark blue dashed line marks the end of the anthropogenic mode (mean diameter of 0.42 µm). In this case, the original size resolution of FidasL has been reduced by integrating 4 consecutive bins except for the last one that contains three, resulting in 16 bins. First integrated bin is not represented as Fidas is considered efficient from the second one. Results are shown only for well-developed erosion conditions ($u_*$>0.15 m s$^{-1}$).

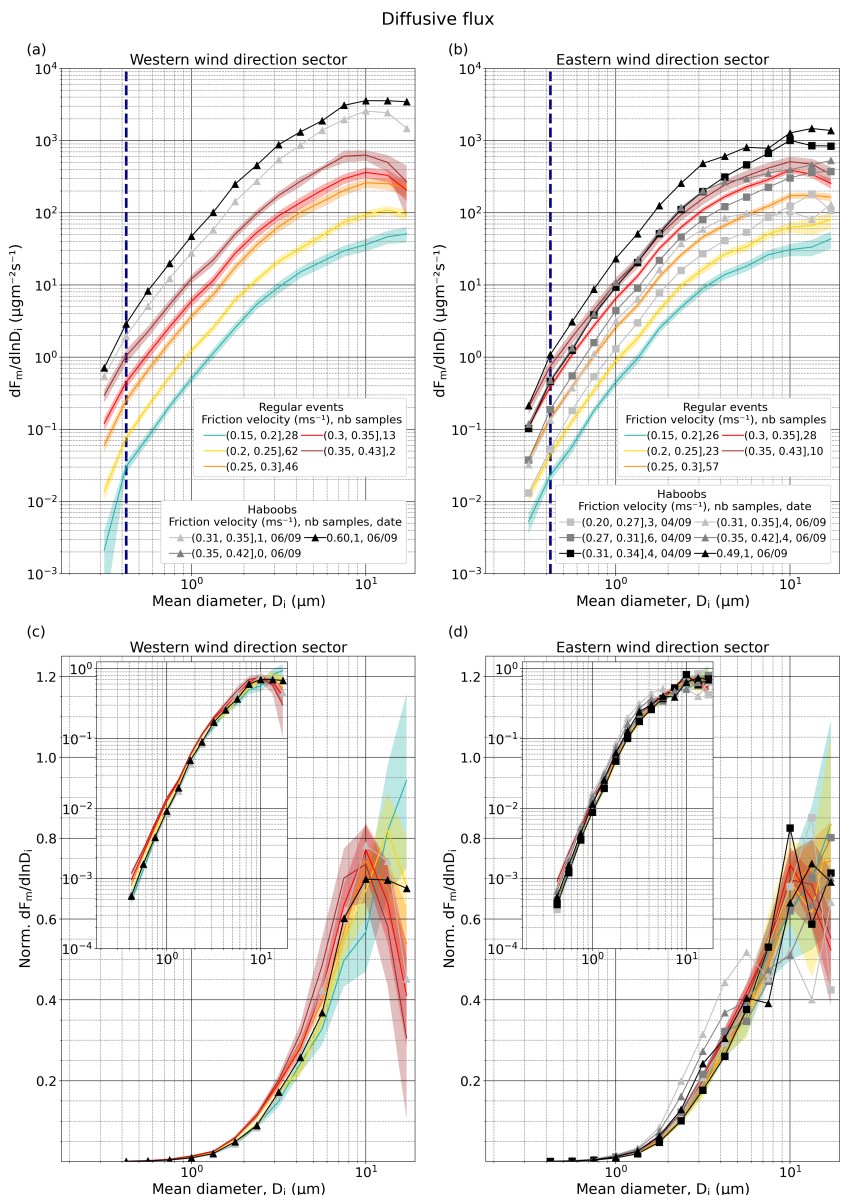

**Figure 9.** Average size-resolved mass diffusive flux, $dF_m/dlnD$ (µg m$^{-2}$ s$^{-1}$), for different $u_*$ intervals, type of event (regular or haboob) and wind directions in the range 150–330 ° (a) and 330–150°(b); The number of available 15-min average PSDs in each $u_*$ class are indicated in the legend. Only the samples where flux is positive in all the diameter bins above the anthropogenic mode (as discussed in Sect. 3.3.1) have been selected; (c-d) same as (a-b), but normalized ($Norm. \; dF_m/dlnD$) after removing the anthropogenic mode (normalization from 0.37 to 19.11 µm). Insets show the same data, but with logarithmic ordinate axis-scaling. Shaded areas around the lines of the regular events PSDs illustrate the combination of random uncertainty and standard error. In (a) and (b) the dark blue dashed line marks the end of the anthropogenic mode (mean diameter of 0.42 µm). In this case, the original size resolution of FidasL has been reduced by integrating 4 consecutive bins except for the last one that contains three, resulting in 16 bins. First integrated bin is not represented as Fidas is considered efficient from the second one. Results are shown only for well-developed erosion conditions ($u_*$>0.15 m s$^{-1}$).





The diffusive flux PSDs show consistent but more marked dependencies upon $u_*$ and wind direction in comparison to the concentration PSDs for well-developed erosion conditions. During regular events, the proportion of submicron (supermicron) particles is lower (higher) and increases (decreases) with $u_*$ more strongly in the flux than in the concentration for both wind direction sectors (Figs. 8c and 8d vs. 6c and 6d). Also, the larger submicron fraction is more enhanced in the flux than in the concentration PSDs when the winds come from the western direction sector. Likewise, the diffusive flux PSDs show more marked variations in coarse and super-coarse particles with increasing $u_*$ compared to the corresponding concentration PSDs, a feature that can be better recognized in terms of mass (Fig. 9). During regular events, there is a depletion of coarse and super-coarse particles with increasing $u_*$ (Figs. 9a and 9b), which translates into a relative decrease (increase) in super-coarse (coarse) particles in the normalized PSDs (Figs. 9c and 9d). As in the case of concentration, there is a shift in the mass diffusive flux PSD towards lower mass median diameters with increasing $u_*$. For the regular events, the uncertainties in the normalized PSDs can partly overlap between contiguous $u_*$ intervals, but the differences among intervals are statistically significant.

In summary, the dependencies of diffusive flux PSDs with $u_*$ and wind direction are consistent with those from concentration for well-developed wind erosion conditions. However, there are relevant differences among them that preclude the use of the near-surface concentration as a proxy for the diffusive flux or the emitted dust PSD.

### 3.3.3   PSD differences between regular and haboob events

The PSDs obtained during the haboob events differ substantially from the PSDs obtained during the regular events even for equivalent $u_*$ intervals and wind direction. When winds come from the eastern direction sector, the number concentration PSDs (Fig. 6b and 6d) show peaks between 1–2 µm (in stark contrast to the 0.5–0.6 µm peak for equivalent $u_*$ during regular events) and the negative slope between 0.4 and 2 µm becomes even positive. There is also a clear relative increase (decrease) in the supermicron (submicron) number dust flux compared to the regular PSDs (Fig. 8d). The coarse and super-coarse dust fractions with diameters $> 5$ µm in the diffusive mass flux PSDs during the haboob events show more variability than during the regular events (Fig. 9d). In some cases we observe a stronger relative decrease (increase) of the super-coarse (coarse) fraction in comparison with the regular events.

Furthermore, coarse and super-coarse dust with diameters $> 4$ µm in the haboob diffusive fluxes show more variability and a higher tendency towards depletion than the regular ones (Figs. 8d and 9d).

When winds come from the western direction sector, the haboob number concentration flux PSDs also tend to show an increase in the supermicron fraction, especially between 1–2 µm (Figs. 6a and 6c), although in this case the maximum fraction of particles still peaks below 1 µm (Fig. 8c). This last feature is consistent with the regular PSDs in that direction showing a more enhanced submicron influence. In contrast to the regular PSDs, we do not detect an increase of submicron particles with increasing $u_*$ in the haboob normalized number flux PSDs in either wind direction (Figs. 8c and 8d). The normalized PSDs associated with the haboob $u_*$ intervals are characterized by larger uncertainties, particularly with increasing particle size than the PSDs associated with the regular events (see in Appendix I Figs. I1 and I2), which is largely due to the smaller number of haboob measurements in each $u_*$ interval.





### 3.3.4 What explains the observed PSD variations? Potential roles of dry deposition and fetch length, aggregate fragmentation, and haboob gust front

In the previous sections we have seen how and to what extent the concentration and diffusive flux PSDs depend upon $u_*$, wind direction and type of event (regular vs haboob). Here, we discuss the potential mechanisms that may explain these PSD variations, which include the effect of dry deposition modulated by the fetch length, the fragmentation of aggregates during wind erosion, and the impact of the haboob gust front.

The proportion of submicron (supermicron) particles decreases (increases) in the concentration PSD between calm (purplish and blueish lines) and well-developed erosion conditions (yellow, orange and red lines) (Figs. 6c and 6d). When $u_*$ is low, i.e., in the absence of local emission, the PSDs represent background conditions and therefore are depleted in supermicron particles due to their shorter lifetime. As $u_*$ increases, the concentration becomes increasingly dominated by freshly emitted dust, reducing the influence of the background dust and hence, the proportion of submicron dust. However, the proportion of submicron (supermicron) particles increases (decreases) in the diffusive flux PSD as $u_*$ increases during regular events (Figs. 8c and 8d). This is also observed, although to a lesser extent, in the concentration PSDs for well-developed erosion conditions when $u_* > 0.25 \, \mathrm{m \, s^{-1}}$ (Figs. 6c and 6d). A priori this could be compatible with two different mechanisms or the combination thereof. On the one side, the relative enhancement of submicron particles may be the result of more aggregate fragmentation as $u_*$ increases (Alfaro et al., 1997; Shao, 2001). On the other side, it could be due to a reduction in supermicron particles by dry deposition, which increases with $u_*$ (Dupont et al., 2015). We examine more thoroughly these two hypotheses below.

The potentially large effect of dry deposition upon the diffusive flux PSDs has been recently suggested based on numerical experiments (Dupont et al., 2015; Fernandes et al., 2019). More specifically, these studies clearly illustrate the key roles of the fetch length and $u_*$ in this process. The fetch is defined as the uninterrupted upwind area generating dust emissions (not to be confused with the flux footprint, which is the upwind area that contributes substantially to the concentration at the measurement location) (Schuepp et al., 1990). For a given surface and uniform $u_*$ along the fetch, the deposition of dust particles slowly increases with the fetch as the concentration of dust is enhanced (Fernandes et al., 2019). Additionally, for a given fetch, an increasing $u_*$ can substantially modify the diffusive flux PSD by enhancing the deposition of supermicron particles through impaction, i.e., the direct collision of particles to a surface resulting from their inertia, and hence, reducing the fraction of these particles.

Our observations evidence the major role of dry deposition in shaping the variations in the concentration and diffusive flux PSDs. For equivalent $u_*$ intervals during regular events, the higher number concentration when the wind comes from the western direction sector is consistent with the longer fetch in that direction. The proportion of submicron (supermicron) particles is higher (lower) when winds come from the western direction sector than when they come from the eastern direction sector both in the concentration and diffusive flux PSDs (Figs. 6c and 8c vs. 6d and 8d). Also, during regular events, the mass fraction of super-coarse particles ($> 10 \, \mu\mathrm{m}$) decreases and that of fine and coarse particles ($< 10 \, \mu\mathrm{m}$) increases as $u_*$ increases, both in the concentration and the diffusive flux PSDs (Figs. 7c, 7d, 9c and 9d), and this effect is more visible when winds come from western wind direction sector, which has a longer fetch.





Our hypothesis is further confirmed when applying the dry deposition (resistance-based) model described in Sect.2.4. The dry deposition velocity increases strongly with particle size due to gravitational settling, and therefore primarily affects coarse and super-coarse dust particles (see Appendix J, Fig.J1). At the same time, the dry deposition velocity scales with $u_*$, in particular, for coarse particles between 2.5 and 10 μm. For example, a value of $\sim 10^{-2}\,\mathrm{m\,s^{-1}}$ is obtained for particles with diameters $\sim$10 μm when $u_*$ is between 0.15 and 0.2 m s$^{-1}$, while when $u_*$ is between 0.35 and 0.45 m s$^{-1}$ (0.55–0.6 m s$^{-1}$) this value is already reached for particles with diameters $\sim$5 μm ($\sim$3 μm). The importance of deposition is clearly depicted in Fig. 10, which displays the size-resolved ratio of the dry deposition flux to the sum of the diffusive and dry deposition fluxes, where this sum basically represents an estimate of the emission flux. (In the Appendix J, we also provide the size-resolved number (Fig. J2) and mass (Fig. J3) dry deposition fluxes). During regular events, we estimate dry deposition to represent up to $\sim$30 % of the emission for super-coarse particles, between 15 and 20 % for 10 μm particles, and up to 10 % for particles as small as 5 μm in diameter. The value of the ratio of the deposition to the diffusive flux is even higher (not shown). While the diffusive flux scales with $u_*$, along with the concentration gradient, the dry deposition flux impacts the coarse and super-coarse fraction increasingly with $u_*$ and concentration, perturbing the diffusive flux PSD when their respective magnitudes are close.

Despite the clear effect of deposition, at least part of the enhancement in submicron particles with $u_*$ could be attributed to an increased aggregate fragmentation. However, while this explanation can hold for regular events, there is no detectable increase in the proportion of submicron particles with increasing u$_*$ in the haboob events in either direction, and the proportion of submicron particles during the haboob events is lower than during regular events even when the former are associated with higher $u_*$ values. This further favors the prevalence of the fetch/deposition mechanism over any potential enhanced fragmentation of aggregates with $u_*$. It is indeed quite remarkable that haboob events tend to show a much higher (lower) proportion of supermicron (submicron) particles than the regular events for equivalent or higher $u_*$ intervals in the concentration PSDs (Figs. 6c and 6d). When it comes to the normalized flux number PSDs (Figs. 8c and 8d), haboob events are similar to the regular events with the lowest $u_*$ (0.15–0.2 m s$^{-1}$), although coarse and super-coarse dust mass fractions with diameters $> 3$ μm during the haboob events show much more variability than during the regular events (Fig. 9d). We hypothesize that these features are explained by the likely smaller horizontal (spatial) extent of the haboob events (smaller than the fetch) compared to the (regional) regular events, due to the proximity of the convective storms originating the initially fresh haboob outflow, and the effect of the moving haboob dust front along with its changing proximity to the measurement site.

A reduced spatial extent during the haboobs is in practice equivalent to a smaller fetch, which can partly explain the enrichment in supermicron particles (dominated by the 1–3 μm size range). Despite the overall increase in the number of supermicron particles, dry deposition visibly affects more strongly the fractions of coarse particles (above 3 μm in diameter) and super-coarse particles in the flux PSDs during the haboob events than during the regular events. As depicted in Fig. 10, the estimated ratios of dry deposition to emission for these fractions are generally higher and more variable during the haboob events than during regular events under similar $u_*$ intervals, reaching up to $\sim$40 % for super-coarse particles, between 15 and 35 % for 10 μm particles, and up to 20 % for particles as small as 5 μm in diameter. This is because the dry deposition flux scales with the concentration, and during the haboobs the concentration of supermicron particles is substantially higher (Figs. 2f and 2g). It is worth noting that the effect of dry deposition with increasing $u_*$ is clearly observed during the haboob events. For example,





when $u_*$ is equal to 0.49 and 0.60 m s$^{-1}$ during the haboob on 6th September (above any other value of $u_*$ during regular events) the dry deposition to emission ratio is enhanced between 2–10 μm, with peaks at 5–6 μm (Fig. 10a) and 7–8 μm (Fig.
10b), respectively, which are clearly detected as reductions in the associated diffusive flux PSDs (Figs. 8a and 8b, respectively). We attribute the higher variability in the diffusive flux PSDs and in the size-resolved deposition to emission ratios during the haboob events to the non-uniformity of the $u_*$ and dust emission across the fetch as the moving gust front (which is where $u_*$ and dust emission are maximized) propagates towards and away from the measurement site. In other words, we hypothesize that the flux PSDs during the haboob events are affected by the distance of the haboob gust front to the measurement site in
each 15-min time interval.

Finally, the increase in air humidity along the haboob outflow and its potential effect upon the soil bonding forces cannot be discarded. During these events, the relative humidity increased substantially, from 15–25 % to ∼50 % (Fig. 2b). Although our near surface soil moisture measurements (2–3 cm deep) (not shown) did not register any associated increase, it has been argued that wet bonding forces in the soil surface, which are dominated by adsorption in arid regions, increase with relative
humidity within approximately the observed variation range (Ravi et al., 2006). This mechanism would be consistent with the smaller proportion of submicron particles due to an increased resistance of soil aggregates to fragmentation with increasing relative humidity as suspected in Dupont (2022).

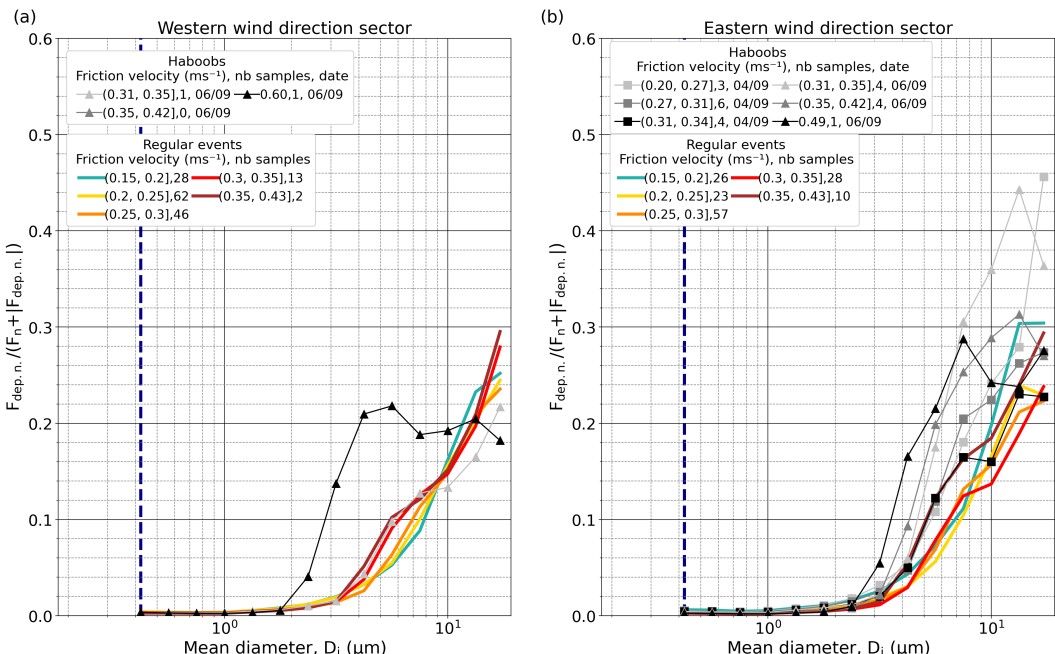

**Figure 10.** Ratio of dry deposition to the sum of diffusive flux and dry deposition flux for different $u_*$ intervals, type of event (regular or haboob) and wind directions in the range 150–330 ° (a) and 330–150°(b)





### 3.3.5 Comparison with Brittle Fragmentation Theory including super-coarse dust

**Figure 11.** Averaged normalized PSDs considering PSL latex spheres with a refractive index of $1.59 + 0i$ removing the anthropogenic mode (normalization from 0.37 to 19.11 µm) for well-developed erosion conditions during regular events and for two PSDs during haboob events for FidasL (a-b) and for diffusive flux (c-d). (a-c) show Norm. dN/dlnD and (b-d) Norm. dM/dlnD. The insets show the same data, but the scale of the ordinate is linear. Pink dashed lines represent the invariant Kok (2011a) size distribution. The original size resolution of FidasL has been reduced by integrating 4 consecutive bins except for the last one that contains three, resulting in 16 bins. First integrated bin is not represented as Fidas is considered efficient from the second one.



**Figure 12.** Averaged normalized PSDs considering tri-axial ellipsoids of $1.49 + 0.0015i$ removing the anthropogenic mode (normalization from 0.37 to 19.11 μm) for well-developed erosion conditions during regular events and for two PSDs during haboob events for FidasL (a-b) and for diffusive flux (c-d). (a-c) show Norm. dN/dlnD and (b-d) Norm. dM/dlnD. The insets show the same data, but the scale of the ordinate is linear. Pink dashed lines represent the invariant Kok (2011a) size distribution. Blue dashed lines represent Meng et al. (2022) data. The original size resolution of FidasL has been reduced by integrating 4 consecutive bins except for the last one that contains three, resulting in 16 bins. First integrated bin is not represented as Fidas is considered efficient from the second one.





Figure 11 sidesteps wind direction differences and focuses on the comparison of the normalized concentration and diffusive
flux PSDs with the emitted PSDs based on the original BFT (Kok, 2011a). (For the sake of clarity in the figure only two haboob
PSDs are represented, corresponding to the two highest values of $u_*$ reached during the haboob events.)

In comparison to the original BFT PSD, we observe a substantially lower (higher) proportion of submicron (supermicron)
particles, particularly in the diffusive flux PSDs (Figs. 11a and 11c). The number concentration PSD is therefore closer to the
BFT PSD than the number flux PSD, in particular during the regular events. In terms of mass, the super-coarse fraction is much
higher in our PSDs (Figs. 11b and 11d), especially in the diffusive flux and during low $u_*$ conditions, which are less affected
by dry deposition. Consequently, the fine and coarse mass fractions are smaller in our measurements than in the BFT PSD.

The measured PSDs shown in Figure 11 assume that dust particles are PSL latex spheres with a refractive index of 1.59+0i.
Fig. 12 is analogous to Fig. 11 but considers a more realistic representation of the shape and composition of the measured dust
particles, i.e. it assumes tri-axial ellipsoids and a refractive index of 1.49+0.0015i, along with the recently updated BFT that
accounts for super-coarse dust (Meng et al., 2022) and is constrained with measured PSDs harmonized to geometric diameters
and also assuming dust is a tri-axial ellipsoid (Huang et al., 2021) (dashed blue line). The proportion of particles in the range
$\sim$0.5–2 µm and above $\sim 14$ µm is higher and that of particles below $\sim$0.5 µm and in the range $\sim$2–14 µm is lower in the
updated BFT number PSD than in the original BFT (blue vs. pink dashed lines in Figs. 12a and 12c). In terms of mass, the
proportion of particles below $\sim$3 µm and above $\sim$12.5 µm is higher and that of particles in the range $\sim$3–12.5 µm is lower in
the updated BFT number PSD than in the original BFT (blue vs. pink dashed lines in Figs. 12b and 12d).

Our converted PSDs show several differences with respect to the updated BFT: 1) both the number concentration (Fig. 12a)
and diffusive flux PSDs (Fig. 12c) show a higher proportion of particles below $\sim$0.8 µm, a lower proportion of particles between
$\sim$0.8 and $\sim$2 µm and a higher proportion of particles above $\sim$2 µm, the latter being even higher in the case of the diffusive
flux; 2) the mass concentration PSDs show from relatively similar to lower fractions below $\sim$2.5 µm that are particularly lower
in the range $\sim$0.8–2.5 µm, relatively similar to higher fractions above $\sim$2.5 µm and below $\sim$12–13 µm, and a higher or lower
fraction of super-coarse dust above $\sim$12–13 µm depending on the type of event and $u_*$; and 3) the mass diffusive flux PSDs
show a similar pattern than the concentration PSDs but feature higher fractions of coarse dust (above $\sim$6–8 µm) and generally
super-coarse dust, and lower fractions of dust below $\sim$6–8 µm, including the strong depletion in the range $\sim$0.8–2.5 µm.

## 4   Conclusions

Soil dust particles created by wind erosion of arid surfaces are a key component of the climate system, and their emitted
PSD partly determines its lifetime and global distribution. In this study, we have contributed towards a better fundamental
understanding of the emitted dust PSD and its variability based on intensive measurements performed during the FRAGMENT
field campaign in the Moroccan Sahara in September 2019. Our measurements were performed in an ephemeral lake located
in the Lower Drâa Valley of Morocco surrounded by small sand dune fields.
Horizontal (saltation) and vertical (diffusive) fluxes occurred regularly, and generally following the diurnal cycles of surface
winds associated to solar heating. In addition to these "regular events", we also identified two "haboob events", on the 4th





and the 6th of September. Two prevailing wind directions were also identified, one centered around $80\,^\circ$ (more aligned with M'hamid El Ghizlane, the closest town) and the other around $240\,^\circ$ (from the Saharan desert).

Our site is characterized by relatively low sandblasting efficiencies in comparison to some previous studies, that we attribute to the hard-crusted paved sediment that constitutes the surface of the ephemeral lake. Despite the low sandblasting efficiencies, diffusive and saltation fluxes are relatively high due to the optimal saltation conditions; the median diameter of the saltators is $132.2\,\mu m$ and a considerable amount is below $100\,\mu m$. The aerodynamic roughness length $z_0$ increases with $u_*$ due to saltation. The sandblasting efficiency decreases with increasing saltation flux and $u_*$, which we partly attribute to the observed increase (decrease) in the proportion of submicron (supermicron) particles in the diffusive flux with increasing $u_*$.

The emitted dust PSD and its variability are still subject to intense debate. In this context, we have thoroughly analyzed the concentration and diffusive flux PSDs in terms of number and mass, observing robust dependencies upon $u_*$, wind direction and type of event (regular vs haboob). We have additionally discussed the mechanisms that may explain the observed PSD variations and compared our PSDs with those predicted by Brittle Fragmentation Theory. During our analysis we identified anthropogenic influence for diameters $< 0.4\,\mu m$, which were removed when evaluating the normalized PSDs.

Our analysis shows differences between the concentration and diffusive flux PSDs, and proves the major role of dry deposition in shaping the PSD variations in both cases, modulated by the wind direction-dependent fetch length, and $u_*$. As far as we know, this is the first time that the effect of dry deposition upon the diffusive fluxes is clearly identified experimentally, supporting results from numerical simulations in recent studies (Dupont et al., 2015; Fernandes et al., 2019). The influence of dry deposition can invalidate the common assumption that the diffusive flux PSD is equivalent to the emitted dust PSD, particularly when including the super-coarse size range, and has consequences on the evaluation of dust emission schemes. In our location, we estimate dry deposition to represent an important portion of dust emission, up to $\sim 40\,\%$ for super-coarse particles, up to $35\,\%$ for $10\,\mu m$ particles, and up to $20\,\%$ for particles as small as $5\,\mu m$ in diameter. This evidences that dry deposition needs to be properly accounted for, even in studies limited to the fine and coarse size ranges, and particularly in measurement locations with long fetches. Our results further imply that at least part of the variability among the diffusive flux PSDs obtained in different locations and that are used to constrain emitted dust PSD theories (e.g. Meng et al., 2022) may be due to the effect of dry deposition modulated by differences in fetch length and $u_*$ regime. While we mainly attribute the enhancement in submicron particles and the reduction in supermicron particles with $u_*$ to the effect of dry deposition, we cannot fully discard that enhanced aggregate fragmentation (Alfaro et al., 1997; Shao, 2001) plays a role, although in the case of haboob events there is no detectable increase in the proportion of submicron particles with increasing $u_*$.

We find clear differences in the haboob PSDs with respect to the regular PSDs, in particular a higher (lower) proportion of supermicron (submicron) particles for equivalent or higher $u_*$ intervals, and more dry deposition and variability in the coarse and super-coarse dust mass fractions with diameters $> 3\,\mu m$. We suggest that these features are due to a smaller horizontal (spatial) extent of the haboob events compared to the (regional) regular events, due to the proximity of the convective storms originating the initially fresh haboob outflow (which is equivalent to the effect of a smaller fetch along with a cleaner background air), and to the effect of the moving haboob dust front, where $u_*$ and dust emission are maximized, along with its changing proximity to the measurement site (which is equivalent to a variable fetch). Our explanation is largely hypothetical



and its validity remains to be verified with, for example, properly designed numerical experiments. We suggest that another mechanism consistent with the smaller proportion of submicron particles would be an increased resistance of soil aggregates to fragmentation with the observed increase in relative humidity along the haboob outflow.

We finally compared our PSDs with the invariant PSDs derived with the original BFT Kok (2011a) and the recently updated BFT that accounts for super-coarse dust emission and uses measurements harmonized in terms of geometric diameter (Meng et al., 2022). We obtain a substantially lower (higher) proportion of submicron (supermicron) particles in the diffusive flux PSDs in comparison with the original BFT PSDs. The super-coarse fraction is substantially higher in our PSDs, especially during low $u_*$ conditions that are less affected by dry deposition. Our comparison with the updated BFT is performed after

transforming the standard optical diameter PSDs into geometric diameter PSDs, where we account for a more realistic index of refraction and shape of the dust particles. Despite the inclusion of super-coarse dust in the updated BFT, our PSDs show a higher proportion of particles above $\sim 2\,\mu m$ and a higher mass fraction of super-coarse particles. It is important to emphasize that this diameter transformation can be very sensitive to shape, refractive index and wavelength (or spectrum) of the light beam. While detailed analysis of this sensitivity was beyond the scope of this study, we plan to assess it in forthcoming studies.

Our study represents an important step towards better understanding the emitted dust PSD in the context of the FRAGMENT project. In companion studies we have tackled the size-resolved composition, shape and mixing state of emitted dust minerals (Panta et al., in prep.), their optical properties (Yus-Díez et al., in prep.), and the mineralogy and size distribution of the parent soil González-Romero et al. (in prep.). Our future studies will combine the obtained results to provide a comprehensive understanding of the emitted dust PSD and its size-resolved composition along with its relationship with the parent soil

properties.

*Data availability.* Data will be available in a public repository upon acceptance of the manuscript.

*Video supplement.* We provide a 1-minute frequency time-lapse video recorded from the Fidas location during September 6th, which clearly shows the arrival of a haboob in the afternoon.





## Appendix A: Surface particle-size distributions at the L'Bour measurement site

The surface of our measurement site at L'Bour consists mainly of a smooth hard-crust paved sediment surrounded by small sand dunes. Fig.A1 shows the PSDs of samples taken for both surface types analyzed in dry (minimally dispersed) and wet dispersion (fully dispersed) along with pictures of the corresponding surfaces. Details on the sampling and analysis methods are provided in González-Romero et al. (in prep.).

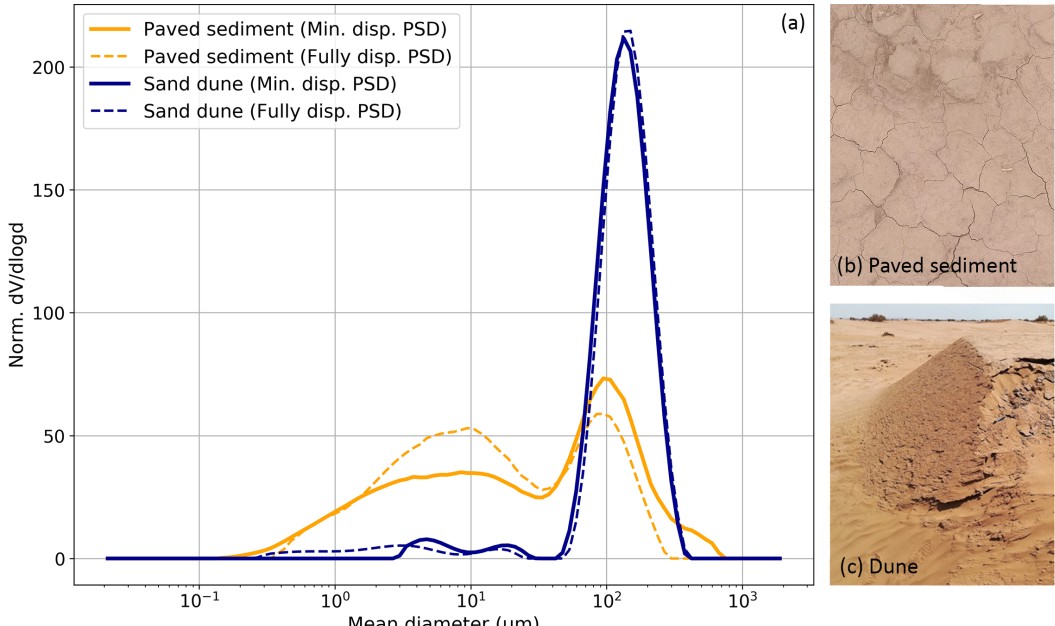

**Figure A1.** (a) Minimally and fully dispersed normalized mean PSDs of a sand dune (blue) and the hard crust paved sediments (orange) in L'Bour. (b) Picture of the paved sediment. (c) Picture of a small sand dune in L'Bour.





**Appendix B:  Wind rose at L'Bour measurement site**

Winds were generally channelled through the valley, broadly parallel to the Drâa river bed, alternating between two opposite and preferential wind directions, centered around $80\,^\circ$ and $240\,^\circ$ as shown in Fig. B1, where colours represent different $u_*$ intervals.

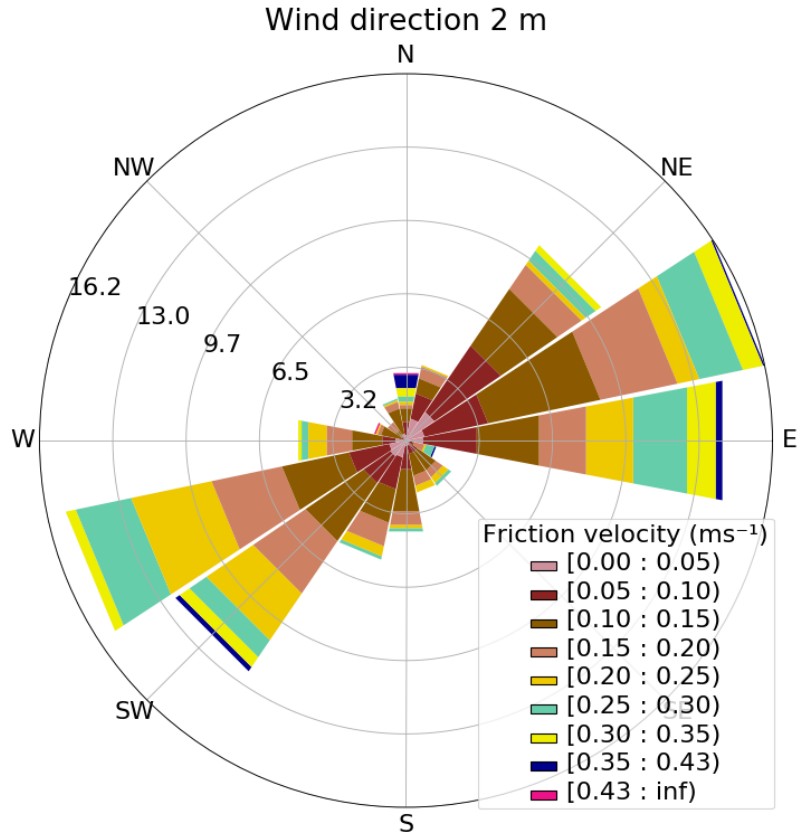

**Figure B1.** Wind rose at 2 m height for different $u_*$ intervals $(\mathrm{m\,s^{-1}})$. The length of each bar represents the fraction of time the wind blows from that direction.





## Appendix C: Comparison between optical and geometric diameters

Fig. C1 displays in both linear and logarithmic scales the default optical diameters of the Fidas OPC versus the associated
geometric diameters whose calculation is described in Sect. 2.2.2.

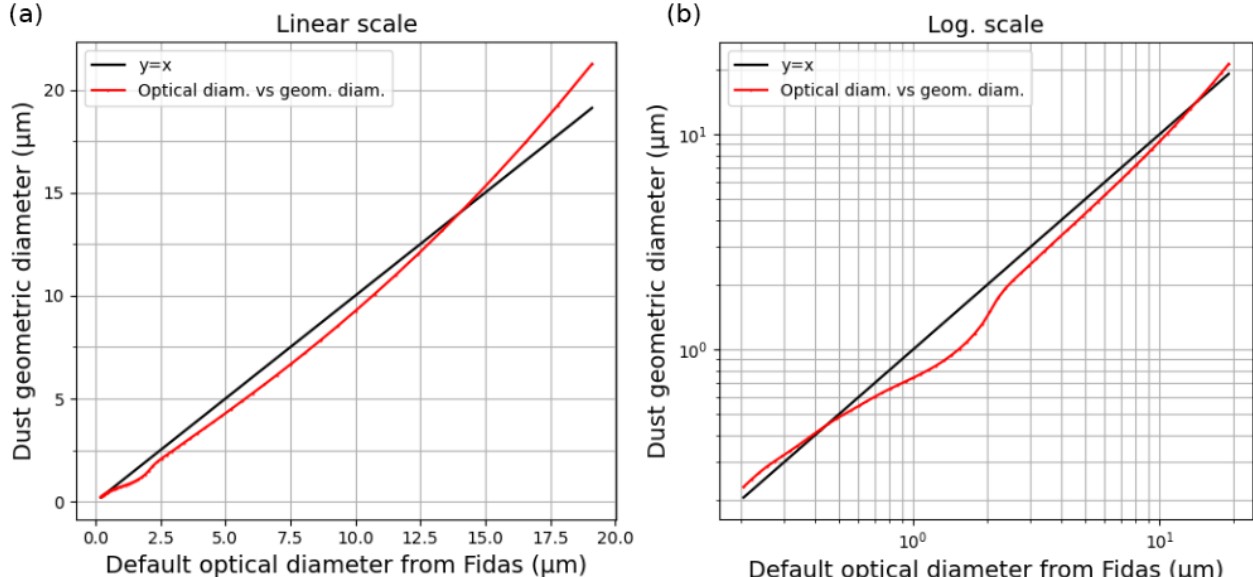

**Figure C1.** Default optical diameters (μm) of the Fidas versus geometric diameters calculated assuming that dust particles are tri-axial ellipsoids with a refractive index of 1.49 + 0.0015 i. (a) Representation in linear scale. (b) Representation in logarithmic scale.



## Appendix D: Fidas systematic correction

By the end of the campaign, the two Fidas were intercompared bin by bin (in the original size bin resolution) at the same height (1.8 m) from 1st October at 10:15 UTC to 2nd October at 08:00 UTC. The goal of the intercomparison was to 1) obtain a correction factor per bin that removes the systematic differences between sensors, and 2) estimate the (random) uncertainty in the size-resolved diffusive flux (see Appendix E). The intercomparison period was affected by a regular event from $\sim$14 to 17 UTC reaching maximum 15-min number and mass concentrations of $\sim 9 \cdot 10^7 \, \# \, \mathrm{m}^{-3}$ and $\sim$2700 $\mu$g m$^{-3}$, respectively, which are very far from the maximum 15-min dust number and mass concentrations of $\sim 1 \cdot 10^9 \, \# \, \mathrm{m}^{-3}$ and $\sim$44700 $\mu$g m$^{-3}$, respectively, measured during the campaign.

We consider the FidasL as the reference device and therefore we correct the systematic deviation of the FidasU. The systematic correction parameter $\lambda_i$ for each bin $i$ shown in Fig. D1a is calculated as the slope of the regression between the concentration of the two Fidas during the intercomparison period:

$$c_{l_0}(D_i) = \lambda_i c_{u_0}(D_i) \tag{D1}$$

where $c_{l_0}$ is the concentration from FidasL and $c_{u_0}$ is the uncorrected concentration from FidasU with diameter $D_i$ during the intercomparison period. If $\lambda_i > 1$ ($< 1$) the concentration of FidasU (FidasL) is lower (higher). Fig. D1a shows $\lambda_i$ in the integrated size bin resolution both in terms of number (green line) and mass (black line) concentrations. Note that number concentrations were transformed to mass concentrations in the original size bin resolution before obtaining the integrated size bin concentrations used to calculate these $\lambda_i$. As shown in Fig. D1b the Pearson correlation coefficient $r$ was above 0.95 for all bins, except for the two coarsest ones where it decays to $\sim$0.88 and $\sim$0.75, respectively.

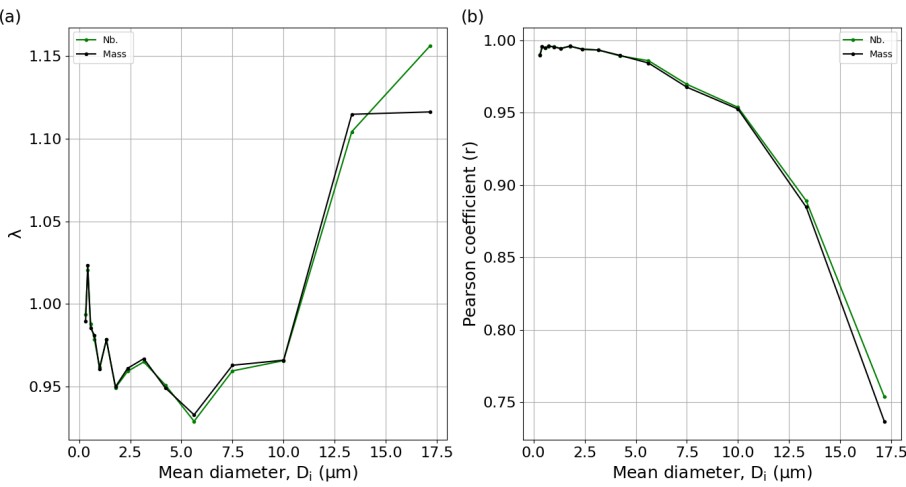

**Figure D1.** (a) Systematic correction parameter $\lambda_i$ and (b) Pearson coefficient $r$ for each integrated size bin $i$. Green (black) lines depict these variables in terms of number (mass) of particles.





The corrected FidasU concentration ($c_u$) during the campaign was then obtained by simply scaling the uncorrected concentration over the whole campaign $c_{u_{uncorr.}}$ with $\lambda_i$:

$$c_u(D_i) = \lambda_i c_{u_{uncorr.}}(D_i) \tag{D2}$$

Similarly, the corrected FidasU concentration ($c_{u_{0_{corr.}}}$) during the intercomparison period is:

$$c_{u_{0_{corr.}}}(D_i) = \lambda_i c_{u_0}(D_i) \tag{D3}$$





## Appendix E: Uncertainty in the size-resolved diffusive flux

There are mainly three sources of uncertainty in the size-resolved diffusive flux calculated with the flux-gradient method (Eq. 9) (Dupont et al., 2021): 1) $u_*$, 2) the difference between FidasU and FidasL concentrations and 3) the difference of stability between the two levels. We neglect the uncertainties on $u_*$ and stability because they are size-independent and small compared to the size-resolved concentration uncertainties (Dupont et al., 2018), and our main interest is the PSD.

We take the FidasL as the reference device, thus the uncertainty in the diffusive flux $\sigma_{F(D_i)}$ only depends on the uncertainty
of the FidasU concentration with respect to the FidasL concentration $\sigma_{c_u(D_i)}$, where $\sigma$ represents the standard deviation:

$$\sigma_{F(D_i)} = u_* \kappa \frac{\sigma_{c_u(D_i)}}{\ln\left(\frac{z_u}{z_l}\right) - \Psi_m\left(\frac{z_u}{L}\right) + \Psi_m\left(\frac{z_l}{L}\right)} \tag{E1}$$

Fig. E1a displays the number concentrations measured by the FidasU after the systematic correction (see Appendix D) versus the FidasL concentrations in each bin during the intercomparison period. We observe a clear relative increase in the scatter as the number concentration decreases both for each bin and across bins. In other words, the relative uncertainty of the number
concentration is strongly dependent upon the number concentration, which is orders of magnitude smaller for large particles than for fine particles. Based on this, we can express the relative uncertainty $\sigma_r$ as:

$$\sigma_r = a(c_u^n)^b \tag{E2}$$

where $c_u^n$ is the FidasU number concentration in any size bin and $a$ and $b$ are constants that can be obtained by fitting the data as described below. Being able to express the uncertainty as a function of the number concentration independent of size
is key to avoid overestimating the uncertainty of the diffusive flux because the concentrations measured during the campaign were generally much higher than the ones measured during the intercomparison period (see Appendix D).

In order to fit Eq. E2, we first calculate the ratio $\lambda_{ij}^n$ of the FidasL to the corrected FidasU number concentrations for each bin $i$ and time step $j$ (every 15-min) during the intercomparison period:

$$\lambda_{ij}^n = c_{l_0}^n(D_i)_j / c_{u_{0_{corr.}}}^n(D_i)_j \tag{E3}$$

where $c_{l_0}^n$ and $c_{u_{0_{corr.}}}^n$ are the FidasL and corrected FidasU number concentrations. Then, we calculate the standard deviation of these ratios $\sigma_{rk}$ within $k$ number concentration intervals as:

$$\sigma_{rk} = \sqrt{\frac{\sum(\lambda_{ij}^{nk} - \overline{\lambda^{nk}})^2}{N-1}} \tag{E4}$$

where $\lambda_{ij}^{nk}$ are the ratios $\lambda_{ij}^n$ within each $k$ interval, $\overline{\lambda^{nk}} \cong 1$ is the average ratio within each interval $k$, and $N$ is the number of samples in each interval $k$. We select four $k$ intervals with the following number concentration ranges: $10^3$–$10^4$, $10^4$–$10^5$,
$10^5$–$10^6$ and $10^6$–$10^7$ $\#\,\mathrm{m}^{-3}$, covering the range of most of the points during the intercomparison period (Fig. E1a).

The $\sigma_{rk}$ values associated to each of the four intervals are displayed in Fig. E1b as a function of $c_u^n$, which is taken as the geometric mean $c_u^n$ within each interval. Using these values we fit $\sigma_r$ and we obtain $a = 51.3$ and $b = -0.45$ with $R^2 = 0.98$ (Fig. E1b).



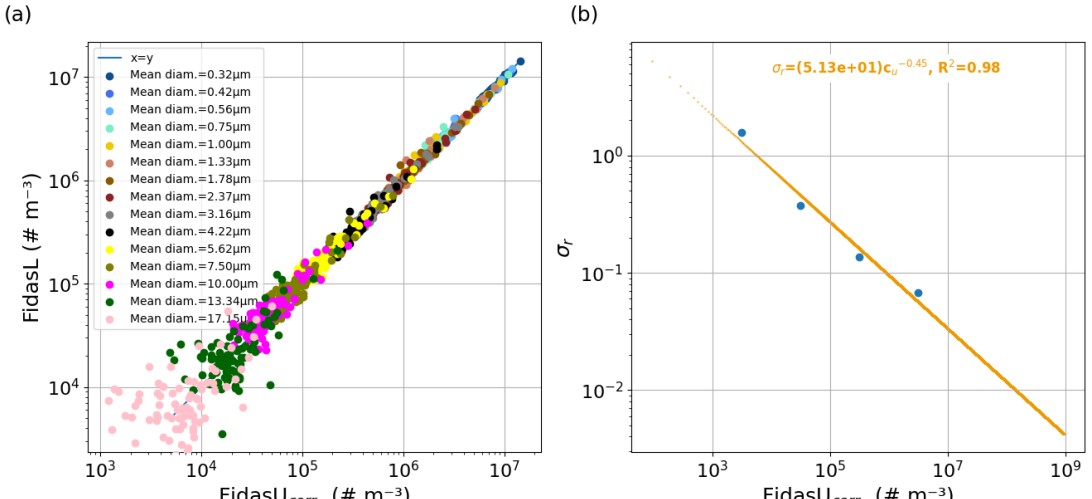

**Figure E1.** (a) FidasL versus FidasU (after systematic correction) number concentrations ($\#\,\mathrm{m}^{-3}$) during the intercomparison period. Concentrations in each bin are represented with different colours. (b) $\sigma_r$ versus corrected FidasU number concentrations ($\#\,\mathrm{m}^{-3}$) during the intercomparison period. The line in (b) represents the regression curve of the form $a \cdot c_u^b$.

Finally, the uncertainty of the FidasU number concentration for each bin $i$ and time step $j$ during the campaign is calculated as:

$$\sigma_{c_u^n(D_i)_j} = \sigma_r c_u^n(D_i)_j = 51.3(c_u^n)^{0.55}, \tag{E5}$$

and the uncertainty of the FidasU mass concentration is then calculated as:

$$\sigma_{c_u^m(D_i)_j} = \sigma_{c_u^n(D_i)_j} \frac{1}{6}\rho_d \pi D_i^3 \tag{E6}$$

where $c_u^m(D_i)_j$ is the corrected mass concentration of FidasU in each bin $i$ and time step $j$ during the campaign, $D_i = \sqrt{d_{max} * d_{min}}$ is the mean logarithmic diameter in bin number $i$, $\mathrm{d}_{max}$ and $\mathrm{d}_{min}$ are the minimum and maximum particle diameters of bin $i$ and $\rho_d$ is the dust particle density, which we assume to be $2500\,\mathrm{kg\,m}^3$.





## Appendix F: Time series of dust concentrations and size-resolved mass fractions

The presence of particles with diameters below ∼0.4 μm, that have an anthropogenic origin as explained in Sect. 3.3.1, is better appreciated in Fig. F1, where size-resolved concentrations (colour contours in right y-axis) are represented as mass fractions

805 (%).

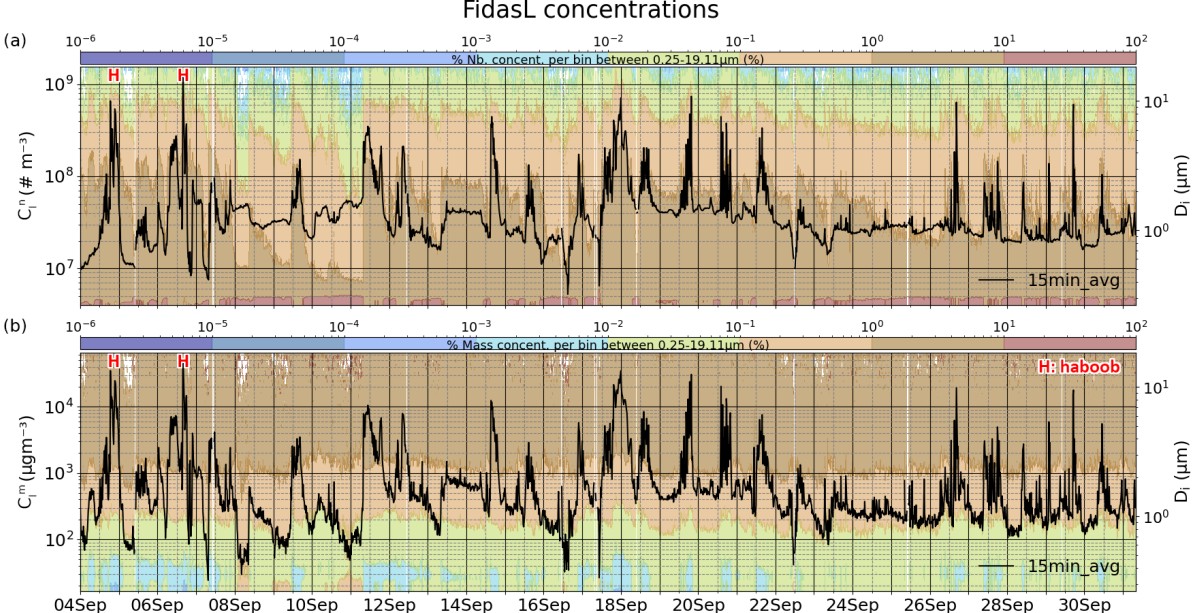

**Figure F1.** Solid lines represent the time evolution of the 15-min average total particle concentrations between 0.25 to 19.11 μm in number ($\# \, m^{-3}$) (a) and mass ($\mu g m^{-3}$) (b). Contour plots on the background show the size-resolved particle number (a) and mass (b) concentration fractions (%) for each time step.





## Appendix G: Additional information about saltation and sandblasting efficiency at L'Bour

Figs. G1 and G2 are similar to Fig. 4 but are done selecting only the 15-min values corresponding to the two predominant wind directions (45–90 ° and 225–270 °, respectively).

Tables G1, G2 and G3 report the parameters $a$ and $b$ derived from each regression curve in Figs. 4, G1 and G2, respectively,
and their 95% confidence intervals.

**Table G1.** Obtained parameters $a$ and $b$ from each regression curve in Fig. 4 along with their 95% confidence intervals.

| | a | a [95% C.I.] | b | b [95% C.I.] |
|---|---|---|---|---|
| $F = a \cdot u_*^b$ | | | | |
| $u_* > 0.1\,\mathrm{m\,s^{-1}}$ | $1.75 \cdot 10^4$ | $[1.16, 2.63] \cdot 10^4$ | $3.35$ | $[3.07, 3.64]$ |
| $u_* > 0.2\,\mathrm{m\,s^{-1}}$ | $4.21 \cdot 10^4$ | $[2.29, 7.75] \cdot 10^4$ | $4.04$ | $[3.58, 4.51]$ |
| $Q = a \cdot u_*^b$ | | | | |
| $u_* > 0.1\,\mathrm{m\,s^{-1}}$ | $7.18 \cdot 10^2$ | $[4.18, 12.33] \cdot 10^2$ | $3.66$ | $[3.28, 4.04]$ |
| $u_* > 0.2\,\mathrm{m\,s^{-1}}$ | $32.46 \cdot 10^2$ | $[14.31, 73.61] \cdot 10^2$ | $4.85$ | $[4.23, 5.47]$ |
| $F/Q = a \cdot u_*^b$ | | | | |
| $u_* > 0.1\,\mathrm{m\,s^{-1}}$ | $2.44 \cdot 10^{-5}$ | $[1.53, 3.87] \cdot 10^{-5}$ | $-0.31$ | $[-0.63, 0.02]$ |
| $u_* > 0.2\,\mathrm{m\,s^{-1}}$ | $1.30 \cdot 10^{-5}$ | $[0.66, 2.57] \cdot 10^{-5}$ | $-0.81$ | $[-1.33, -0.29]$ |
| $F/Q = a \cdot Q^b$ | | | | |
| $u_* > 0.1\,\mathrm{m\,s^{-1}}$ | $5.97 \cdot 10^{-5}$ | $[5.37, 6.64] \cdot 10^{-5}$ | $-0.32$ | $[-0.38, -0.27]$ |
| $u_* > 0.2\,\mathrm{m\,s^{-1}}$ | $7.28 \cdot 10^{-5}$ | $[6.41, 8.27] \cdot 10^{-5}$ | $-0.38$ | $[-0.44, -0.32]$ |

**Figure G1.** (a) Diffusive flux ($\mu g\, m^{-2}\, s^{-1}$) versus friction velocity $u_*$ ($m\, s^{-1}$); (b) Saltation flux ($g\, m^{-1}\, s^{-1}$) versus $u_*$ ($m\, s^{-1}$); (c) Sandblasting efficiency ($m^{-1}$) versus $u_*$ ($m\, s^{-1}$); (d) Sandblasting efficiency ($m^{-1}$) versus saltation flux ($g\, m^{-1}\, s^{-1}$). The points shown in all panels correspond to the 15-min values in which 1) there is a simultaneous net positive diffusive flux and saltation flux, 2) the diffusive flux is positive in all size bins above $0.4\,\mu m$ and 3) wind direction is between $45$–$90\,°$. Squares (triangles) are used to identify the values corresponding to haboobs on 4th (6th) September. The lines in (a)-(d) represent the regression curves of the form $a \cdot u_*^b$ for $u_* > 0.1\,m\, s^{-1}$ (blue) and for $u_* > 0.2\,m\, s^{-1}$ (orange). Parameters $a$ and $b$ of each regression curve and their respective 95% confidence intervals are reported in Table G2.





**Figure G2.** (a) Diffusive flux ($\mu g\,m^{-2}\,s^{-1}$) versus friction velocity $u_*$ ($m\,s^{-1}$); (b) Saltation flux ($g\,m^{-1}\,s^{-1}$) versus $u_*$ ($m\,s^{-1}$); (c) Sandblasting efficiency ($m^{-1}$) versus $u_*$ ($m\,s^{-1}$); (d) Sandblasting efficiency ($m^{-1}$) versus saltation flux ($g\,m^{-1}\,s^{-1}$). The points shown in all panels correspond to the 15-min values in which 1) there is a simultaneous net positive diffusive flux and saltation flux, 2) the diffusive flux is positive in all size bins above $0.4\,\mu m$ and 3) wind direction is between 225–270°. Squares (triangles) are used to identify the values corresponding to haboobs on 4th (6th) September. The lines in (a)-(d) represent the regression curves of the form $a \cdot u_*^b$ for $u_* > 0.1\,m\,s^{-1}$ (blue) and for $u_* > 0.2\,m\,s^{-1}$ (orange). Parameters $a$ and $b$ of each regression curve and their respective 95% confidence intervals are reported in Table G3.





**Table G2.** Obtained parameters $a$ and $b$ from each regression curve in Fig. G1 (wind directions between 45–90 $^\circ$) along with their 95% confidence intervals.

| | a | a [95% C.I.] | b | b [95% C.I.] |
|---|---|---|---|---|
| | | $F = a \cdot u_*^b$ | | |
| $u_* > 0.1\,\mathrm{m\,s^{-1}}$ | $6.62 \cdot 10^4$ | $[3.54, 12.37] \cdot 10^4$ | $4.25$ | $[3.80, 4.70]$ |
| $u_* > 0.2\,\mathrm{m\,s^{-1}}$ | $1.98 \cdot 10^5$ | $[0.72, 5.49] \cdot 10^5$ | $5.13$ | $[4.34, 5.92]$ |
| | | $Q = a \cdot u_*^b$ | | |
| $u_* > 0.1\,\mathrm{m\,s^{-1}}$ | $2.96 \cdot 10^3$ | $[1.13, 7.75] \cdot 10^3$ | $4.38$ | $[3.69, 5.08]$ |
| $u_* > 0.2\,\mathrm{m\,s^{-1}}$ | $8.10 \cdot 10^3$ | $[1.63, 40.37] \cdot 10^3$ | $5.18$ | $[3.94, 6.42]$ |
| | | $F/Q = a \cdot u_*^b$ | | |
| $u_* > 0.1\,\mathrm{m\,s^{-1}}$ | $2.23 \cdot 10^{-5}$ | $[1.04, 4.83] \cdot 10^{-5}$ | $-0.13$ | $[-0.69, 0.42]$ |
| $u_* > 0.2\,\mathrm{m\,s^{-1}}$ | $2.45 \cdot 10^{-5}$ | $[0.73, 8.21] \cdot 10^{-5}$ | $0.05$ | $[-0.99, 0.89]$ |
| | | $F/Q = a \cdot Q^b$ | | |
| $u_* > 0.1\,\mathrm{m\,s^{-1}}$ | $4.53 \cdot 10^{-5}$ | $[3.69, 5.55] \cdot 10^{-5}$ | $-0.26$ | $[-0.35, -0.18]$ |
| $u_* > 0.2\,\mathrm{m\,s^{-1}}$ | $5.71 \cdot 10^{-5}$ | $[4.42, 7.35] \cdot 10^{-5}$ | $-0.33$ | $[-0.43, -0.24]$ |



**Table G3.** Obtained parameters $a$ and $b$ from each regression curve in Fig. G2 (wind directions between 225–270 $^\circ$) along with their 95% confidence intervals.

| | a | a [95% C.I.] | b | b [95% C.I.] |
|---|---|---|---|---|
| | | $F = a \cdot u_*^b$ | | |
| $u_* > 0.1\,\mathrm{m\,s^{-1}}$ | $1.37 \cdot 10^4$ | $[0.61, 3.07] \cdot 10^4$ | $3.25$ | $[2.70, 3.81]$ |
| $u_* > 0.2\,\mathrm{m\,s^{-1}}$ | $8.37 \cdot 10^4$ | $[2.61, 26.89] \cdot 10^4$ | $4.61$ | $[3.76, 5.46]$ |
| | | $Q = a \cdot u_*^b$ | | |
| $u_* > 0.1\,\mathrm{m\,s^{-1}}$ | $3.14 \cdot 10^2$ | $[1.34, 7.33] \cdot 10^2$ | $3.25$ | $[2.67, 3.83]$ |
| $u_* > 0.2\,\mathrm{m\,s^{-1}}$ | $18.30 \cdot 10^2$ | $[5.22, 64.12] \cdot 10^2$ | $4.57$ | $[3.66, 5.48]$ |
| | | $F/Q = a \cdot u_*^b$ | | |
| $u_* > 0.1\,\mathrm{m\,s^{-1}}$ | $4.36 \cdot 10^{-5}$ | $[2.05, 9.29] \cdot 10^{-5}$ | $-0.00$ | $[-0.52, 0.53]$ |
| $u_* > 0.2\,\mathrm{m\,s^{-1}}$ | $4.58 \cdot 10^{-5}$ | $[1.44, 14.57] \cdot 10^{-5}$ | $0.04$ | $[-0.81, 0.88]$ |
| | | $F/Q = a \cdot Q^b$ | | |
| $u_* > 0.1\,\mathrm{m\,s^{-1}}$ | $5.49 \cdot 10^{-5}$ | $[4.71, 6.41] \cdot 10^{-5}$ | $-0.22$ | $[-0.33, -0.11]$ |
| $u_* > 0.2\,\mathrm{m\,s^{-1}}$ | $5.93 \cdot 10^{-5}$ | $[4.91, 7.16] \cdot 10^{-5}$ | $-0.24$ | $[-0.37, -0.12]$ |





## Appendix H: PSDs obtained with FidasU

Figs. H1 and H2 are equivalent to Figs. 6 and 7, but using data from FidasU after correcting the systematic deviation (see Appendix D). As it is usual, as the height increases dust concentration decreases. However, we find the same features (explained in Sect. 3.3) than for FidasL.



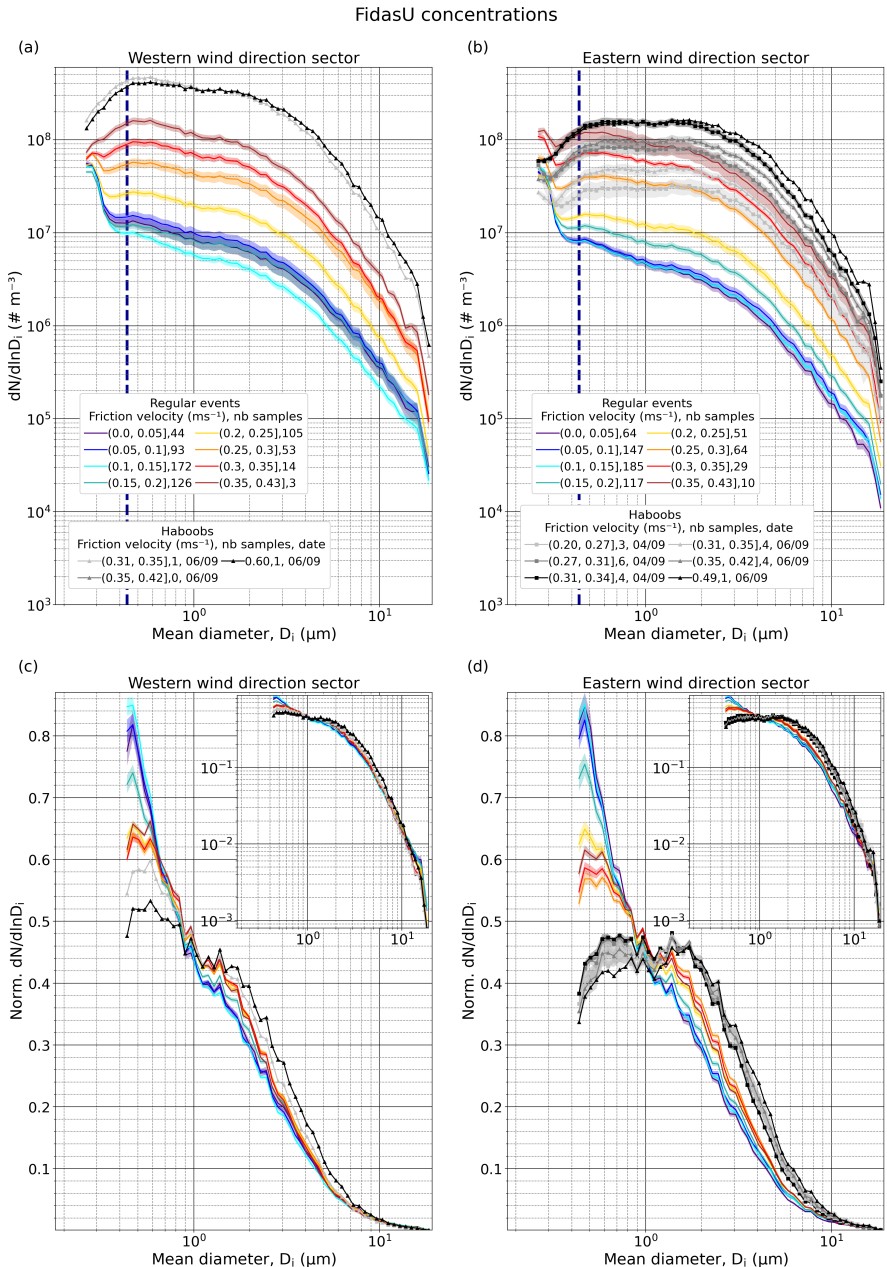

**Figure H1.** Average size-resolved particle number concentration, $dN/dlnD$ (# m$^{-3}$), for different $u_*$ intervals, type of event (regular or haboob) and wind directions in the range 150–330 ° (a) and 330–150°(b); The number of available 15-min average PSDs in each $u_*$ interval is indicated in the legend; (c-d) same as (a-b), but normalized ($Norm.\ dN/dlnD$) after removing the anthropogenic mode (normalization from 0.42 to 19.11 µm). Insets show the same data, but with logarithmic ordinate axis-scaling. Shaded areas around the lines depict the standard error. The shown PSDs were obtained from FidasU. In (a) and (b) the dark blue dashed line marks the end of the anthropogenic mode (mean diameter $D_i = 0.44$ µm). Data are shown using original size bin resolution, but first three bins are not represented as Fidas is considered efficient from the fourth one.



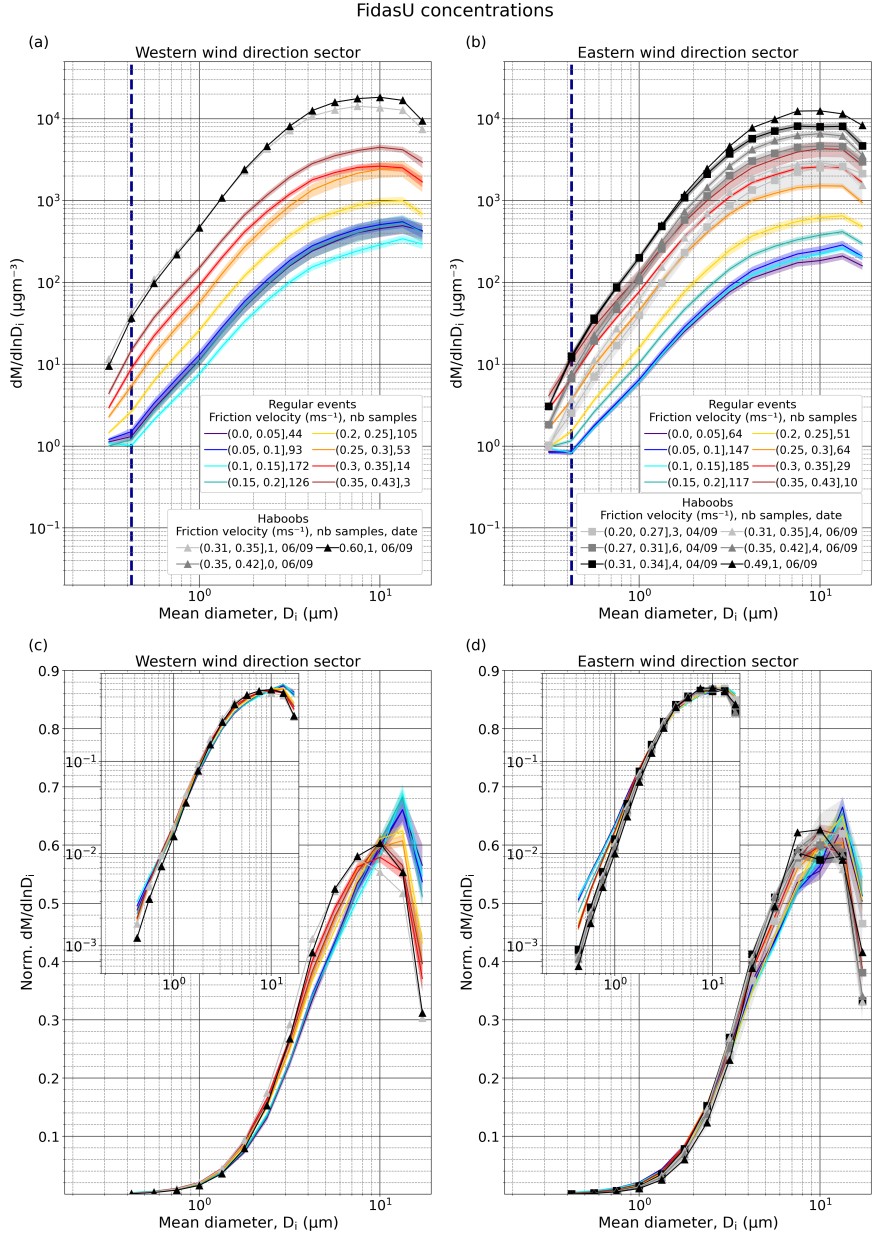

**Figure H2.** Average size-resolved particle mass concentration, $dM/dlnD$ (µg m$^{-3}$), for different $u_*$ intervals, type of event (regular or haboob) and wind directions in the range 150–330$^\circ$ (a) and 330–150°(b); The number of available 15-min average PSDs in each $u_*$ interval are indicated in the legend; (c-d) same as (a-b), but normalized ($Norm.\ dM/dlnD$) after removing the anthropogenic mode (normalization from 0.37 to 19.11 µm). Insets show the same data, but with logarithmic ordinate axis-scaling. Shaded areas around the lines depict the standard error. The shown PSDs were obtained from FidasU. In (a) and (b) the dark blue dashed line marks the end of the anthropogenic mode (mean diameter $D_i = 0.42$ µm). In this case, the original size resolution of FidasU has been reduced by integrating 4 consecutive bins except for the last one that contains three, resulting in 16 bins. First integrated bin is not represented as Fidas is considered efficient from the second one.





## Appendix I: Additional plots of the diffusive flux PSDs


Figs. I1 and I2 show the same plots as Figs. 8 and 9 but including the uncertainties for each $u_*$ range only for the haboob events. We also provide the diffusive flux PSDs with uncertainties only accounting for standard errors (Figs. I3 and I4). As the standard error depends inversely on the number of samples, those cases in which there is only a sample do not show any shaded areas.



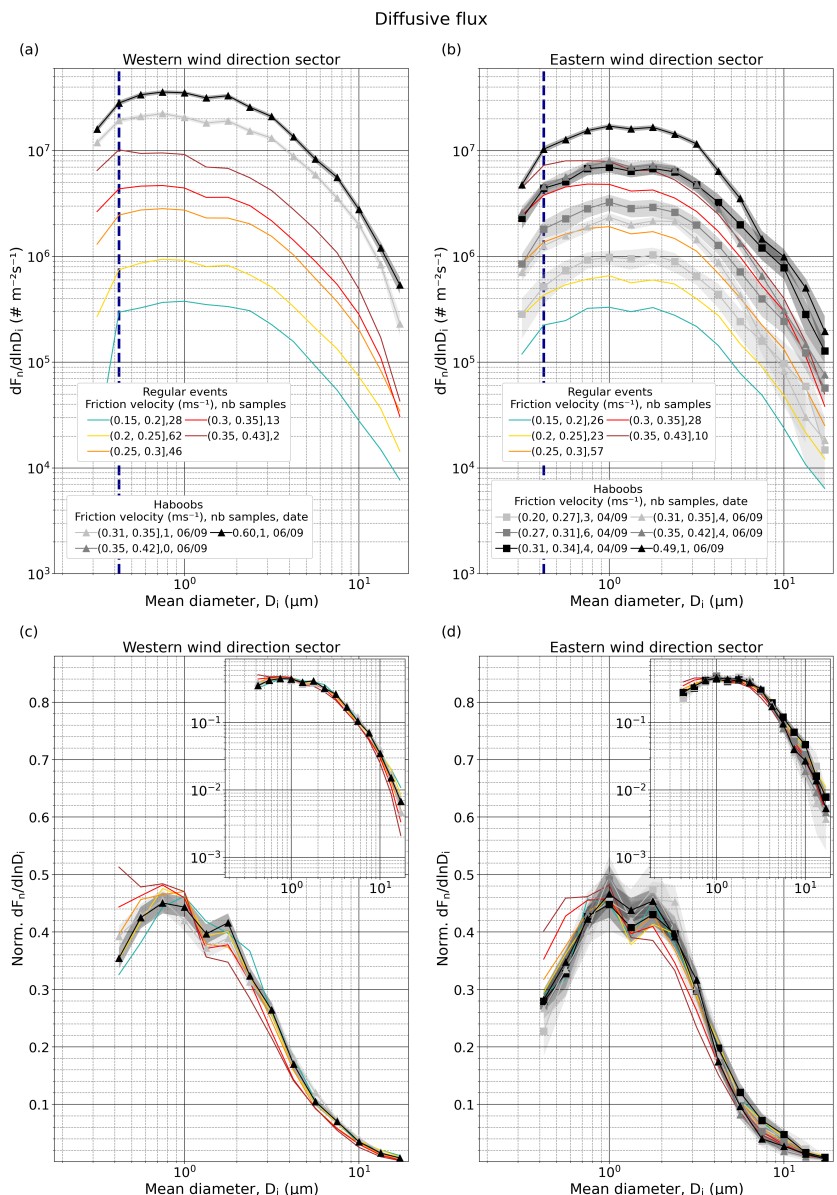

**Figure I1.** Average size-resolved number diffusive flux, $dF_n/dlnD$ ($\# \, \mathrm{m^{-2} \, s^{-1}}$), for different $u_*$ intervals, type of event (regular or haboob) and wind directions in the range 150–330° (a) and 330–150°(b); The number of available 15-min average PSDs in each $u_*$ interval are indicated in the legend. Only the samples where flux is positive in all the diameter bins above the anthropogenic mode (as discussed in Sect. 3.3.1) have been selected; (c-d) same as (a-b), but normalized (Norm. $dF_n/dlnD$) after removing the anthropogenic mode (normalization from 0.37 to 19.11 µm). Insets show the same data, but with logarithmic ordinate axis-scaling. Shaded areas around the lines of the haboob events PSDs depict the combination of random uncertainty and standard error. In (a) and (b) the dark blue dashed line marks the end of the anthropogenic mode (mean diameter of 0.42 µm). In this case, the original size resolution of FidasL has been reduced by integrating 4 consecutive bins except for the last one that contains three, resulting in 16 bins. First integrated bin is not represented as Fidas is considered efficient from the second one. Results are shown only for well-developed erosion conditions ($u_*>0.15 \, \mathrm{m \, s^{-1}}$).



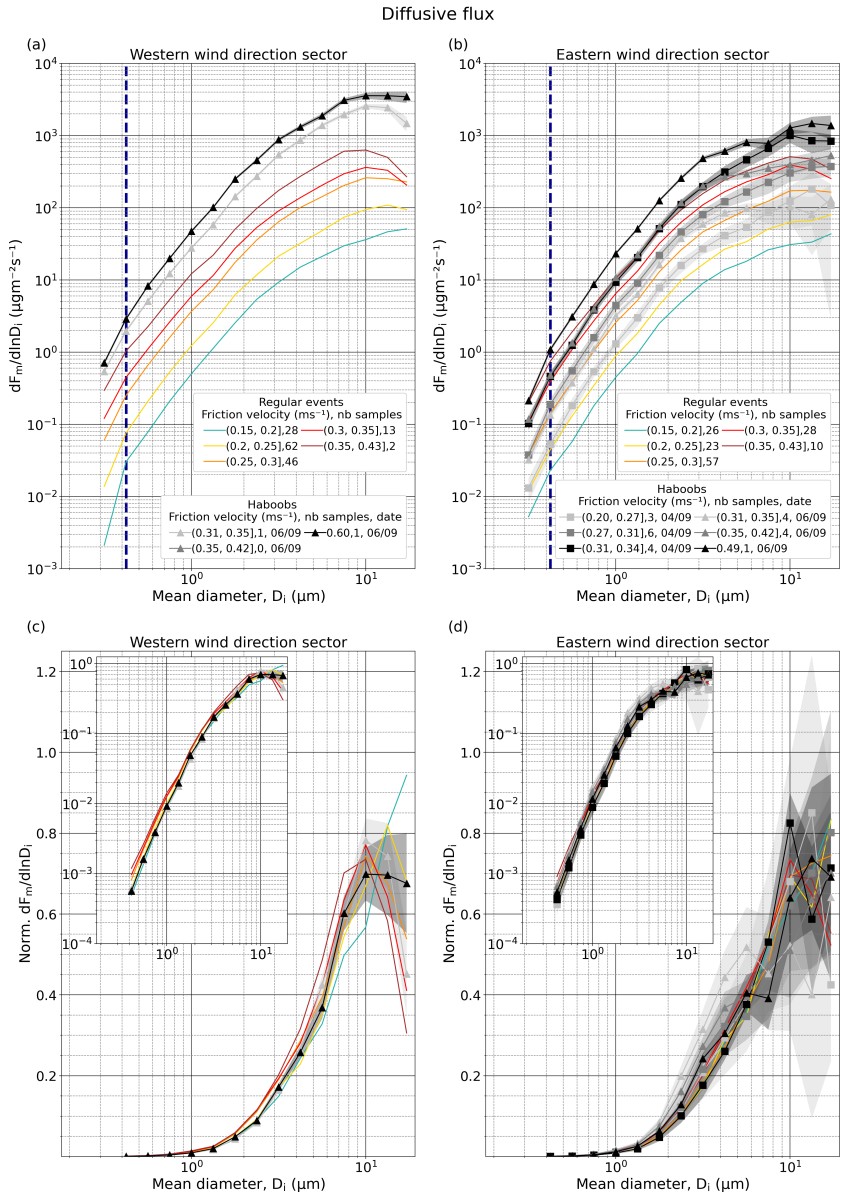

**Figure I2.** Average size-resolved mass diffusive flux, $dF_m/dlnD$ ($\mu g\,m^{-2}\,s^{-1}$), for different $u_*$ intervals, type of event (regular or haboob) and wind directions in the range 150–330 ° (a) and 330–150°(b); The number of available 15-min average PSDs in each $u_*$ class are indicated in the legend. Only the samples where flux is positive in all the diameter bins above the anthropogenic mode (as discussed in Sect. 3.3.1) have been selected; (c-d) same as (a-b), but normalized ($Norm.\ dF_m/dlnD$) after removing the anthropogenic mode (normalization from 0.37 to 19.11 µm). Insets show the same data, but with logarithmic ordinate axis-scaling. Shaded areas around the lines of the haboob events PSDs depict the combination of random uncertainty and standard error. In (a) and (b) the dark blue dashed line marks the end of the anthropogenic mode (mean diameter of 0.42 µm). In this case, the original size resolution of FidasL has been reduced by integrating 4 consecutive bins except for the last one that contains three, resulting in 16 bins. First integrated bin is not represented as Fidas is considered efficient from the second one. Results are shown only for well-developed erosion conditions ($u_*$>0.15 m s$^{-1}$).



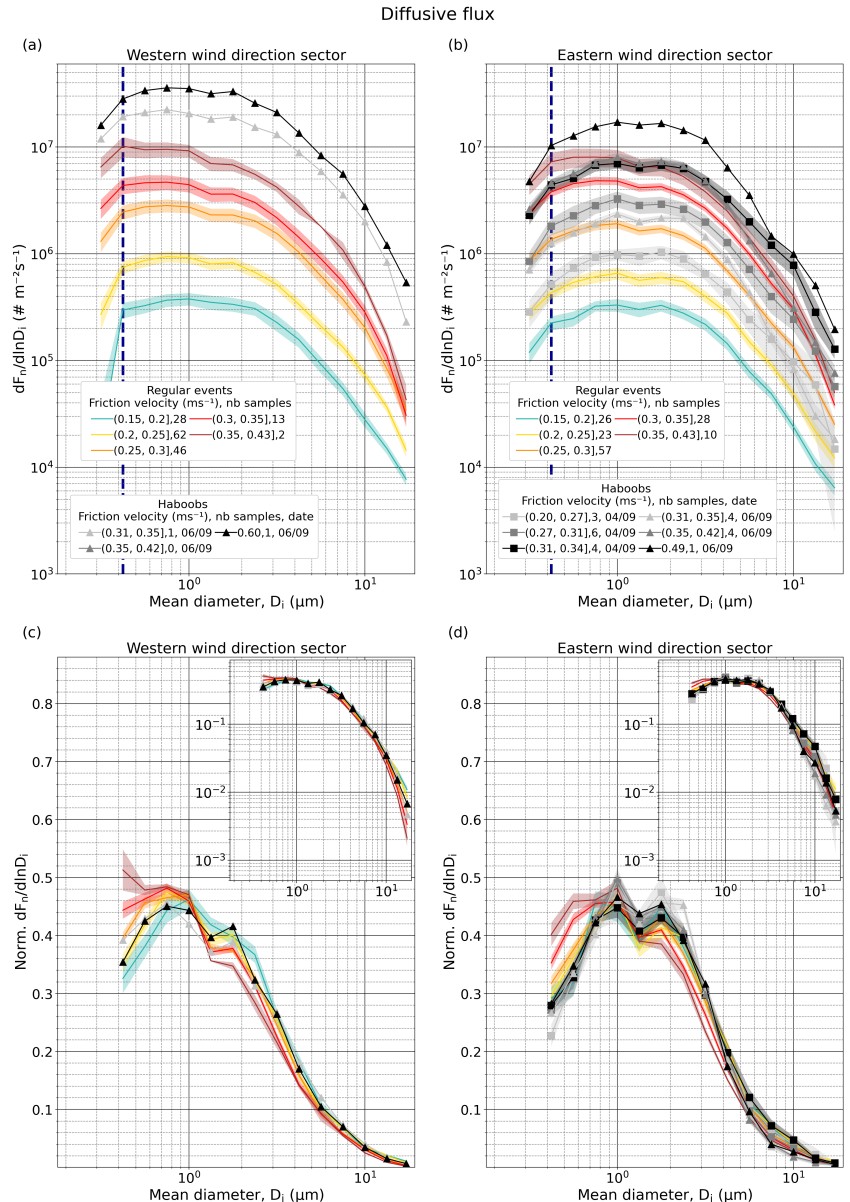

**Figure I3.** Average size-resolved number diffusive flux, $dF_n/dlnD$ ($\#\,\mathrm{m^{-2}\,s^{-1}}$), for different $u_*$ intervals, type of event (regular or haboob) and wind directions in the range 150–330 ° (a) and 330–150°(b); The number of available 15-min average PSDs in each $u_*$ interval are indicated in the legend. Only the samples where flux is positive in all the diameter bins above the anthropogenic mode (as discussed in Sect. 3.3.1) have been selected; (c-d) same as (a-b), but normalized (Norm. $dF_n/dlnD$) after removing the anthropogenic mode (normalization from 0.37 to 19.11 μm). Insets show the same data, but with logarithmic ordinate axis-scaling. Shaded areas around the lines depict the standard error. In (a) and (b) the dark blue dashed line marks the end of the anthropogenic mode (mean diameter of 0.42 μm). In this case, the original size resolution of FidasL has been reduced by integrating 4 consecutive bins except for the last one that contains three, resulting in 16 bins. First integrated bin is not represented as Fidas is considered efficient from the second one. Results are shown only for well-developed erosion conditions ($u_*$>0.15 m s$^{-1}$).

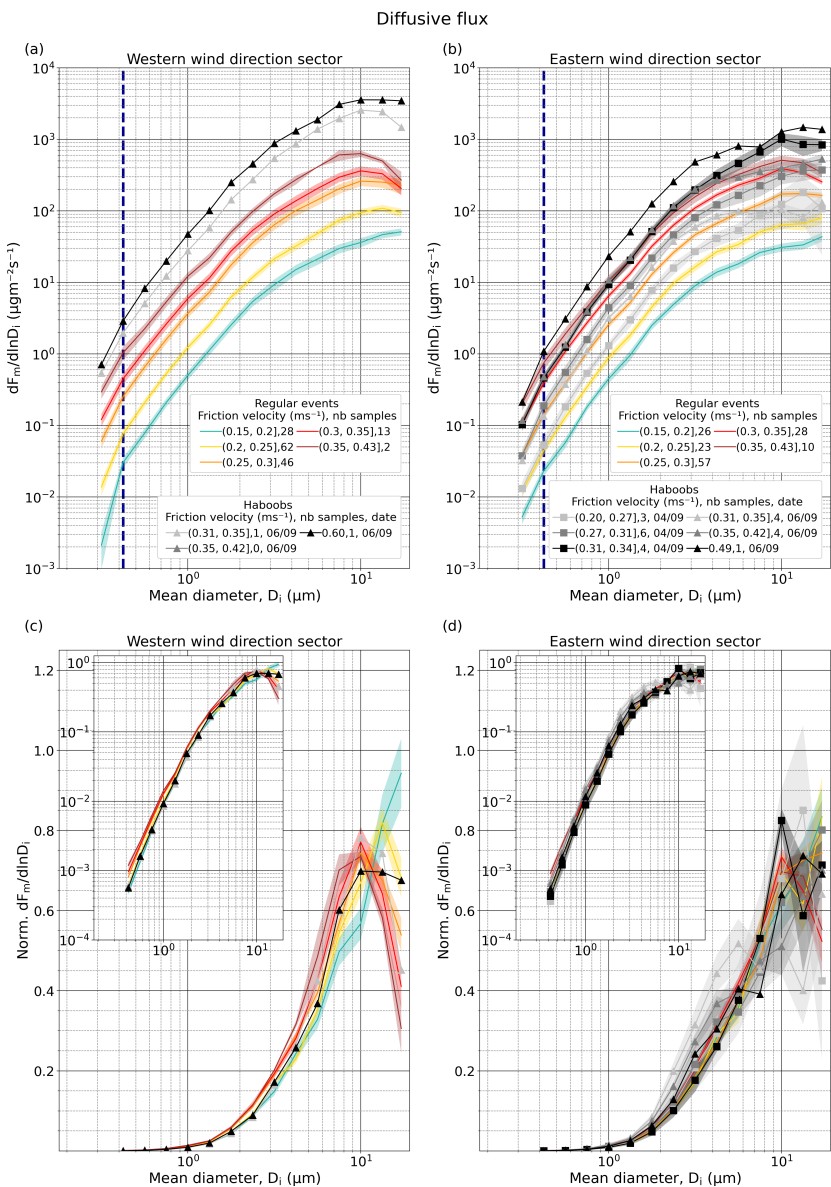

**Figure I4.** Average size-resolved mass diffusive flux, $dF_m/dlnD$ (µg m$^{-2}$ s$^{-1}$), for different $u_*$ intervals, type of event (regular or haboob) and wind directions in the range 150–330 ° (a) and 330–150°(b); The number of available 15-min average PSDs in each $u_*$ class are indicated in the legend. Only the samples where flux is positive in all the diameter bins above the anthropogenic mode (as discussed in Sect. 3.3.1) have been selected; (c-d) same as (a-b), but normalized ($Norm. \ dF_m/dlnD$) after removing the anthropogenic mode (normalization from 0.37 to 19.11 µm). Insets show the same data, but with logarithmic ordinate axis-scaling. Shaded areas around the lines depict the standard error. In (a) and (b) the dark blue dashed line marks the end of the anthropogenic mode (mean diameter of 0.42 µm). In this case, the original size resolution of FidasL has been reduced by integrating 4 consecutive bins except for the last one that contains three, resulting in 16 bins. First integrated bin is not represented as Fidas is considered efficient from the second one. Results are shown only for well-developed erosion conditions ($u_*$>0.15 m s$^{-1}$).





**Appendix J: Dry deposition flux calculations**

Fig. J1 shows the dry deposition velocity $v_{dep}$ calculated from Eq. 13 for different $u_*$ intervals using the measurements.

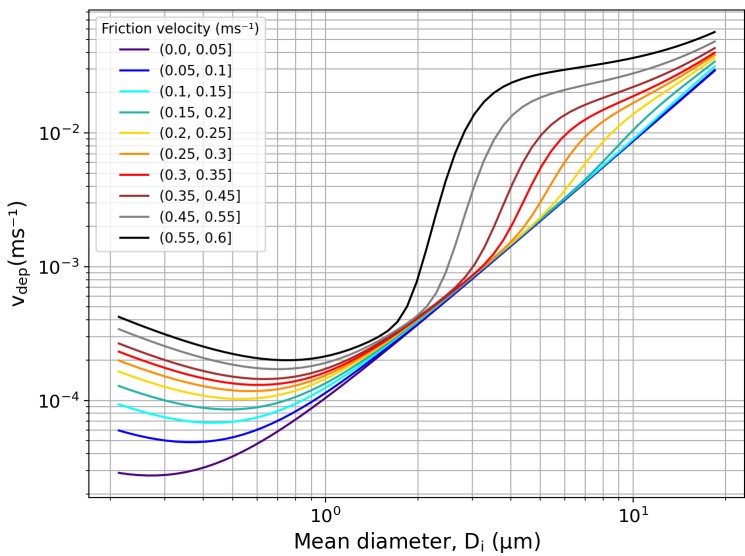

**Figure J1.** Dry deposition velocity $v_{dep}(\mathrm{m\,s^{-1}})$ versus mean diameter $D_i(\mu m)$ for different $u_*$ intervals $(\mathrm{m\,s^{-1}})$.

Figs. J2 and J2 represent, respectively, the number and mass dry deposition flux calculated from Eq. 12 for different $u_*$ intervals, types of dust event (regular and haboob events) and wind direction (Eastern and Western wind directions sectors).

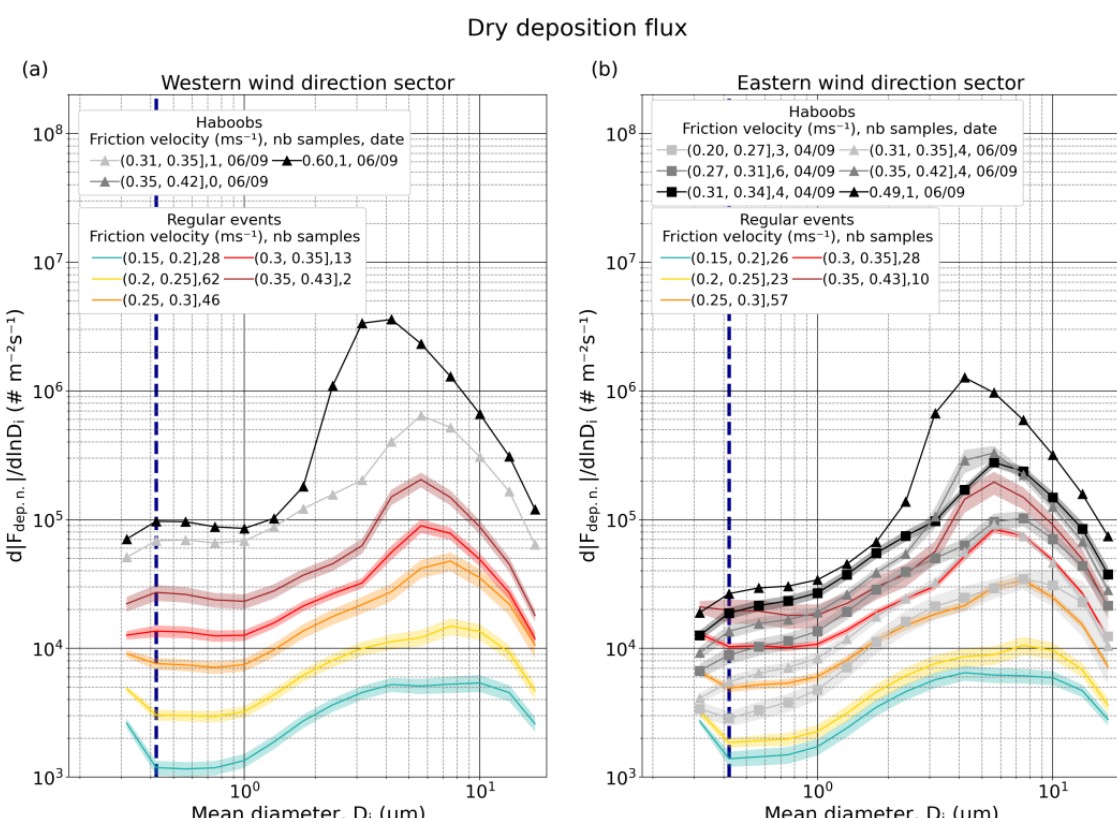

**Figure J2.** Average size-resolved number dry deposition flux, $d|F_{dep.n}|/dlnD$ ($\# \, \mathrm{m}^{-2} \, \mathrm{s}^{-1}$), for different $u_*$ intervals, type of event (regular or haboob) and wind directions in the range 150–330 ° (a) and 330–150°(b). The number of available 15-min average PSDs in each $u_*$ interval are indicated in the legend. Shaded areas around the lines depict the standard error. In (a) and (b) the dark blue dashed line marks the end of the anthropogenic mode (mean diameter of $0.42 \, \mu\mathrm{m}$). In this case, the original size resolution of FidasL has been reduced by integrating 4 consecutive bins except for the last one that contains three, resulting in 16 bins. The first integrated bin is not represented as the Fidas is considered efficient from the second one.

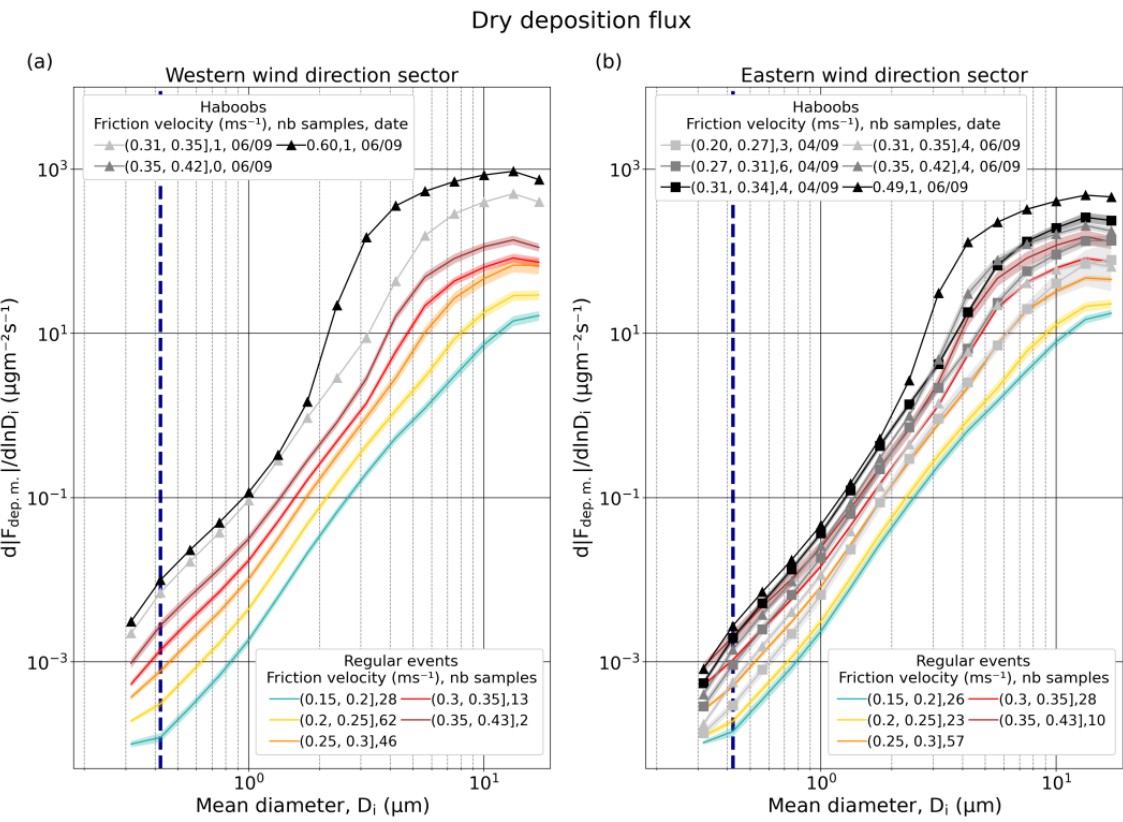

**Figure J3.** Average size-resolved mass dry deposition flux, $d|F_{dep.m}|/dlnD$ (µg m$^{-2}$ s$^{-1}$), for different $u_*$ intervals, type of event (regular or haboob) and wind directions in the range 150–330 $^\circ$ (a) and 330–150°(b). The number of available 15-min average PSDs in each $u_*$ interval are indicated in the legend. Shaded areas around the lines depict the standard error. In (a) and (b) the dark blue dashed line marks the end of the anthropogenic mode (mean diameter of 0.42 µm). In this case, the original size resolution of FidasL has been reduced by integrating 4 consecutive bins except for the last one that contains three, resulting in 16 bins. The first integrated bin is not represented as the Fidas is considered efficient from the second one.



*Author contributions.* CGF processed the meteorological and OPCs datasets, analyzed the results, created all the figures and drafted the manuscript. CPG-P and MK supervised the work. CPG-P proposed and designed the measurement campaign with contributions from XQ, MK, AA, KK, SD and VE. CGF, AGR, MK, KK, AP, XQ, CR, JYD, AA and CPG-P implemented the field campaign. VE and GN provided the SANTRIs as well as corresponding scientific and technical support. MK calculated the saltation flux. AGR and XQ performed the soil analysis. JE provided the conversions between optical and geometric diameters in collaboration with YH. All authors provided feedback on the structure and/or the content of the final manuscript. CPG-P re-edited the manuscript.

*Competing interests.* The authors declare that they have no competing interests.

*Acknowledgements.* The field campaign and its associated research, including this work, was primarily funded by the European Research Council under the Horizon 2020 research and innovation programme through the ERC Consolidator Grant FRAGMENT (grant agreement No. 773051) and the AXA Research Fund through the AXA Chair on Sand and Dust Storms at BSC. CGF was supported by a PhD fellowship from the Agència de Gestió d'Ajuts Universitaris i de Recerca (AGAUR) grant 2020-FI-B 00678. MK received funding through the Helmholtz Association's Initiative and Networking Fund (grant agreement no. VH-NG-1533). KK was funded by the Deutsche Forschungsgemeinschaft (DFG, German Research Foundation) – 264907654; 416816480. YH acknowledges the financial support by the Columbia University Earth Institute Postdoctoral Research Fellowship. The SANTRI instruments used in this study were constructed under a grant (No. EAR-1124609) from the US National Science Foundation.

We acknowledge the EMIT project, which is supported by the NASA Earth Venture Instrument program, under the Earth Science Division of the Science Mission Directorate. We thank Joaquim Cebolla-Alemany for his help in editing some of the figures. We thank Dr. Santiago Beguería from the National Scientific Council of Spain for facilitating a field site in Zaragoza, Spain, to test our instrumentation and field procedures prior to the campaign in Morocco. We thank Prof. Kamal Taj Eddine from Cady Ayyad University, Marrakesh, Morocco for his invaluable support and suggestions in the preparation of the field campaign. We thank Houssine Dakhamat and the crew of Hotel Chez le Pacha in M'hamid el Ghizlane for their local support during the campaign. We thank Andrés Carrillo and Agnès Sauleda for their support with the Fidas before, during and after the campaign. We thank Campbell Scientific in Barcelona for their support with the meteorological instruments. We thank PALAS for their help with some technical issues with the fidas.



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
