# Peer review of "Insights into the size-resolved dust emission from field measurements in the Moroccan Sahara"

_Atmospheric Chemistry and Physics, 2022_

## Referee Comment (RC2)

Review report on

Insights into the size-resolved dust emission from field measurements in the Moroccan Sahara

By Cristina González-Flórez, Martina Klose, Andrés Alastuey, Sylvain Dupont, Jerónimo Escribano, Vicken Etyemezian, Adolfo Gonzalez-Romero, Yue Huang, Konrad Kandler, George Nikolich, Agnesh Panta, Xavier Querol, Cristina Reche, Jesús Yus-Díez, and Carlos Pérez García-Pando1

Gonzalez-Florez et al. use field measurements to investigate size-resolved dust emission. This is a comprehensive study, covering a number of issues important to dust research. The authors have made a major effort to push forward the dust research boundaries. This effort is significant as indeed we still do not know enough about dust emission processes, and our capacity of quantifying dust emission is still unsatisfactory, in particular size-resolved dust emission. In recent years, there has been a number of studies on this topic, with contradictory outcomes.

It is probably useful beforehand to clarify that there is never any doubt that dust emission, including the particle size characteristics of emitted dust, depends on the balance between the forces lifting dust and the forces resisting the lifting. In some studies, the emphasis is placed on the former, while in others on the latter. For example, Khalfallah et al. (2020) and Shao et al. (2021) emphasized the importance of the forces to entrain the dust, implicitly assume the soil conditions are the same. These authors never said that the resistance forces, such as soil binding due to soil moisture, are not important. Of course, they are important, as Dupont (2022) suggests. In fact, the use of minimally and fully-dispersed PSD (Shao, 2001) is an attempt to represent the soil binding strength, while the BFT (Kok, 2011a, b) assumes the dependency of binding force on particle size is universal (which is, in my view, extremely unlikely). When we discuss these earlier papers, I believe, we should bare in mind the explicit and implicit assumptions made and present the discussions in a sound framework.

The main conclusion of this paper seems to be that dry dust deposition is important to the size-resolved dust emission. This seems to be a sensible conclusion to make, but the line of argument seems to me both interesting and confusing. In large scale models (e.g. Klose et al. 2021, MONARCH), dust emission and deposition are treated separately. We must therefore clearly define which dust flux is being studied, the part of entrained by wind, or the net dust flux. Both dust emission and deposition are fluxes which serve as boundary conditions for the diffusion process in the atmosphere. At this stage of parameterization, dust emission is parameterized without considering ambient dust concentration, while deposition is parameterized by considering ambient dust concentration profile (following this study, deposition flux equals deposition velocity x dust concentration at a reference level). We know dust concentration in the atmospheric boundary-layer (ABL) is stability dependent, because stability affects both dust concentration profile and deposition velocity (Yin et al. 2022, ACP, Large-eddy-simulation …). It thus seems to be contradictory to state that "size-resolved dust emission" is not ABL stability dependent, while dust deposition is important. But deposition depends on ABL stability (or not?).

The method of Gillette et al. (1972) for computing diffusive dust flux is widely used. Normally, it is not a big problem if we only want to provide an estimate for the total dust emission and assume diffusive dust flux is the same as the emission flux. But Gillette et al. (1972) method is problematic to use for the purpose of this study, as it concerns size-resolved dust emission. There is a discussion in Section 7.1 of Shao (2008, Physics and Modelling of Wind Erosion) on the different definitions of the fluxes, and why the Monin-Obukhov similarity relationship may not apply here. I believe, Line 285 of the paper, $\Phi_d = \Phi_m$, is problematic, even if the Csanady (1963) approximation holds, due to the gravitational settling. As this

paper emphasizes on the impact of dry deposition on diffusive flux, a correction to Eq. (9) seems to be warranted. The correction (e.g., Shao et al. 2011a) may result in stronger corrections (with respect to Gillette et al. 1972) to larger dust particles and hence lead to somewhat different size-resolved dust emission.

On several occasions, the impact of fetch (and haboob and wind direction) is mentioned. The effect of fetch is to generate a horizontal advection which influences the diffusive flux, while the Obukhov similarity (and hence the Gillette et al. 1972 method, i.e., Eq. (9)) assumes horizontal homogeneity. Again, the dust concentration equation is

$$\frac{\partial c}{\partial t} + u\frac{\partial c}{\partial x} + \frac{(w - w_t)\partial c}{\partial z} = K_p \partial^2 c/\partial z^2$$

Under the assumption of steady state and homogeneity, the above equation reads

$$w_t\frac{\partial c}{\partial z} + K_p\frac{\partial^2 c}{\partial z^2} = 0$$

(by the way the Obukhov similarity assumes $K_p\frac{\partial^2 c}{\partial z^2} = 0$). If there is a fetch effect, then the horizontal homogeneity assumption of Obukhov (and hence Gillette et al. 1972) is no-longer valid and the above equation reads for steady state

$$u\frac{\partial c}{\partial x} + \frac{(w - w_t)\partial c}{\partial z} = K_p \partial^2 c/\partial z^2$$

It seems contradictory to me to apply the Obukhov similarity to analyse the data and then conclude that the fetch effect is important to the size-resolved dust emission, as some sort of physics-based interpretation, rather than to say it may be the uncertainty related to the use of the Obukhov theory. The statement related to wind direction and haboob is probably also attributed to advection.

In specifying the similarity functions, z0 is sometime considered and sometimes not, e.g., Eq. (2) and (5). How important is z0?

In general, I find the work very well done, and the authors have thoroughly studied the literature. But it is (for me at least) a very heavy paper with lengthy descriptions. I find the abstract very long. It may be trying to solve too many problems at once (size-resolved dust emission, deposition, fetch, haboob, BFT etc.). I believe the paper would have a larger impact, if it were more concentrated on the core issues.

---

## Author Comment (AC1)

**Referee #1 Yaping Shao**

We wish to thank Yaping Shao for his very valuable comments. Below we provide detailed responses to each comment separately. Our responses to reviewer comments are in blue normal text.

**General comments**

González-Flórez et al. use field measurements to investigate size-resolved dust emission. This is a comprehensive study, covering a number of issues important to dust research. The authors have made a major effort to push forward the dust research boundaries. This effort is significant as indeed we still do not know enough about dust emission processes, and our capacity of quantifying dust emission is still unsatisfactory, in particular size-resolved dust emission. In recent years, there has been a number of studies on this topic, with contradictory outcomes.

We really appreciate the positive perspective of Yaping Shao on our work.

It is probably useful beforehand to clarify that there is never any doubt that dust emission, including the particle size characteristics of emitted dust, depends on the balance between the forces lifting dust and the forces resisting the lifting. In some studies, the emphasis is placed on the former, while in others on the latter. For example, Khalfallah et al. (2020) and Shao et al. (2021) emphasized the importance of the forces to entrain the dust, implicitly assume the soil conditions are the same. These authors never said that the resistance forces, such as soil binding due to soil moisture, are not important. Of course, they are important, as Dupont (2022) suggests. In fact, the use of minimally and fully-dispersed PSD (Shao, 2001) is an attempt to represent the soil binding strength, while the BFT Kok (2011a, b) assumes the dependency of binding force on particle size is universal (which is, in my view, extremely unlikely). When we discuss these earlier papers, I believe, we should bare in mind the explicit and implicit assumptions made and present the discussions in a sound framework.

We thank Yaping Shao for this comment and overview, with which we agree. In fact, in the introduction we briefly describe some distinct aspects of current theories/approaches in relation to the variability of the PSD at emission that broadly reflects your comment : *"In the particle size range up to $\sim 10\,\mu m$ in diameter, some theoretical frameworks predict a higher proportion of emitted fine particles with increasing wind speed during saltation along with dependencies of the PSD on soil properties (Shao et al., 1993; Alfaro et al., 1997; Shao, 2001). In contrast, the emitted PSD is posited to be relatively independent of wind speed and soil properties in another theoretical framework (Kok, 2011a), based on Brittle Fragmentation Theory (BFT)."*

The main conclusion of this paper seems to be that dry dust deposition is important to the size-resolved dust emission. This seems to be a sensible conclusion to make, but the line of argument seems to me both interesting and confusing. In large scale models (e.g. Klose et al., 2021), dust emission and deposition are treated separately. We must therefore clearly define which dust flux is being studied, the part of entrained by wind, or the net dust flux. Both dust emission and deposition are fluxes which serve as boundary conditions for the diffusion process in the atmosphere. At this stage of parameterization, dust emission is parameterized without considering ambient dust concentration, while deposition is parameterized by considering ambient dust concentration profile (following this study, deposition flux equals deposition velocity x dust concentration at a reference level). We know dust concentration in the atmospheric boundary-layer (ABL) is stability dependent, because stability affects both dust concentration profile and deposition velocity (Yin et al., 2022). It thus seems to be contradictory to state that "size-resolved dust emission" is not ABL stability dependent, while dust deposition is important. But deposition depends on ABL stability (or not?).

As discussed in the introduction of the paper, most studies relate the diffusive flux PSD obtained at a few meters above the surface to the emitted dust flux at the surface, assuming a constant dust flux layer and neglecting the gravitational settling and turbulent dry deposition. The gravitational settling term is assumed to be small for dust smaller than $\sim 10\,\mu m$. The diffusive flux PSD is afterward used directly to constrain or evaluate dust emission schemes, or even to assess to what extent the emitted dust PSD may be affected by atmospheric forcing and soil properties, neglecting the deposition component of the net dust flux at the surface. In previous studies, using modeling, Dupont et al. (2015) and Fernandes et al. (2019) have shown the potentially large effect of dry deposition (including losses by turbulent and Brownian motion, and inertial impaction) upon the diffusive flux PSD.

45    In our paper, we analyzed this effect based on field experimental data. Specifically, we discussed the potential role of dry deposition in shaping at least part of the variability observed in the diffusive flux. In the revised version of the paper, we also provide estimates of the emitted flux by accounting for dry deposition. For that purpose, we estimated the dry deposition velocity by tuning a dry deposition velocity parameterization to fit our measurement data (see Sects. 2.4, S10 and Appendix D).

50    Regarding the dependence of deposition fluxes on ABL stability, we revisited Yin et al. (2022) and we found that in their case a large range of stability conditions (from unstable to stable) were covered, while our dust events in Morocco (driven by saltation) comprised a narrower range of stability conditions (mostly near-neutral, forced convection regimes) where very likely the stochasticity of $u_*$ remained stable and poorly impacted by convective motions as in Dupont (2022). This means that for our range of stability during dust events, the deposition velocities are unlikely stability dependent. Therefore, it seems not

55    contradictory to observe no stability effect on the diffusive dust flux PSD and for the diffusive flux to include deposition. We have added a corresponding remark in the text. In addition, Yin et al. (2022) argued that $u_*$ is stability dependent because of the increasing stochasticity of $u_*$ in unstable conditions, but in Dupont (2022) (WIND-O-V data) that $u_*$ stochasticity remained quite stable for dust events driven by saltation.

    The method of Gillette et al. (1972) for computing diffusive dust flux is widely used. Normally, it is not a big problem if we

60    only want to provide an estimate for the total dust emission and assume diffusive dust flux is the same as the emission flux. But Gillette et al. (1972) method is problematic to use for the purpose of this study, as it concerns size-resolved dust emission. There is a discussion in Section 7.1 of Shao (2008) on the different definitions of the fluxes, and why the Monin Obukhov similarity relationship may not apply here. I believe, line 285 of the paper, $\phi_d = \phi_m$, is problematic, even if the Csanady (1963) approximation holds, due to the gravitational settling. As this paper emphasizes on the impact of dry deposition on

65    diffusive flux, a correction to Eq. (9) seems to be warranted. The correction (e.g. Shao et al., 2011) may result in stronger corrections (with respect to Gillette et al. (1972)) to larger dust particles and hence lead to somewhat different size-resolved dust emission.

    We agree that for coarse particles, our assumption $\phi_d = \phi_m$ may not be well justified. However, the correction for heavy particles used in Shao et al. (2011), which represents the change in the turbulent diffusivity due to the trajectory crossing effect,

70    appears small here (see Fig. 1). We therefore decided to not account for it.
    The correction term for heavy particles $C$ that would multiply our Eq. 9 is given by Eq. 1.

$$C = \left(1 + \frac{\beta^2 v_g^2}{\sigma_w^2}\right)^{-1/2} \tag{1}$$

where $\beta$ is a dimensionless coefficient relating the fluid Lagrangian integral time scale, the integral length scale of the Eulerian fluid velocity field and the standard deviation of the turbulent velocity. In Csanady (1963), it is said that $\beta$ is very close to 1,

75    so we assume $\beta = 1$. The settling velocity $v_g(D_i)$ is calculated for each size bin as $v_g(D_i) = C_c \sigma_{pa} g D_i^2/(18\nu)$ where $C_c$ is the Cunningham slip correction factor, $\nu = 1.45 \cdot 10^{-5}\,\mathrm{m^2\,s^{-1}}$ is the air kinematic viscosity, $g = 9.81\,\mathrm{m\,s^{-2}}$ is the gravitational acceleration, $D_i$ is the mean logarithmic diameter in bin number $i$, and $\sigma_{pa} = (\rho_d - \rho_{air})/\rho_{air}$ is the particle-to-air density ratio. The unbiased variance of turbulent velocity $\sigma_w$ is calculated from the $w$ component of the 3-D sonic anemometer placed at $1\,\mathrm{m}$. Figure 1 shows the mean correction term for heavy particles per size bin only considering the periods where diffusive

80    flux was positive in all the diameter bins above the anthropogenic mode (as discussed in Sect. 3.3.1).

    On several occasions, the impact of fetch (and haboob and wind direction) is mentioned. The effect of fetch is to generate a horizontal advection which influences the diffusive flux, while the Obukhov similarity (and hence the Gillette et al. (1972) method, i.e., Eq. (9)) assumes horizontal homogeneity. Again, the dust concentration equation is:

$$\frac{\partial c}{\partial t} + u\frac{\partial c}{\partial x} + \frac{(w - w_t)\partial c}{\partial z} = \kappa_p \frac{\partial^2 c}{\partial z^2} \tag{2}$$

85 Under the assumption of steady state and homogeneity, the above equation reads

$$w_t \frac{\partial c}{\partial z} + \kappa_p \frac{\partial^2 c}{\partial z^2} = 0 \tag{3}$$

[Figure]

**Figure 1.** Mean correction factor for heavy particles per size bin. Only the samples where diffusive flux is positive in all the diameter bins above the anthropogenic mode (as discussed in Sect. 3.3.1) have been used. Error bars represent the standard deviation.

(by the way the Obukhov similarity assumes $\kappa_p \frac{\partial^2 c}{\partial z^2}$=0). If there is a fetch effect, then the horizontal homogeneity assumption of Obukhov (and hence Gillette et al. (1972)) is no-longer valid and the above equation reads for steady state

$$u\frac{\partial c}{\partial x} + \frac{(w - w_t)\partial c}{\partial z} = \kappa_p \frac{\partial^2 c}{\partial z^2} \qquad (4)$$

90    It seems contradictory to me to apply the Obukhov similarity to analyse the data and then conclude that the fetch effect is important to the size-resolved dust emission, as some sort of physics-based interpretation, rather than to say it may be the uncertainty related to the use of the Obukhov theory. The statement related to wind direction and haboob is probably also attributed to advection.

        The fetch effect mentioned in the paper corresponds to a long distance dust source fetch effect of the order of 10 km
95    in the Eastern direction and 60 km in the Western direction. This fetch effect leads to a small-continuous increase of dust concentration along the fetch, and thus an increase of dust deposition while the dust emission remains unaffected. Importantly, this dust fetch effect does not impact the wind dynamics. Consequently, the conditions of application of the Monin-Obukhov similarity theory are valid for the wind velocity fields. In the formulation of the dust flux-gradient method from Gillette et al. (1972) (Eq. 9), we can therefore still relate the local momentum flux to the surface friction velocity ($\langle u'w'\rangle = -u_*^2$) and replace
100   the mean wind velocity profile using the pseudo logarithmic form with the usual stability functions (Sect. 2.2 in Dupont et al. (2021)).

        We agree that a long distance variation of the dust concentration implies horizontal dust advection. In fact, in presence of a non-zero diffusive flux (emission or deposition or both), there should always be advection as particles should accumulate or diminish within the boundary layer, assuming (1) that the boundary-layer depth does not change after (let's say) more than
105   1 km from the upwind border of the dust source and (2) that the dust flux at the top of the boundary layer is negligible. If the

horizontal dust advection term is significant near the surface, then the estimated diffusive dust flux at several meters height cannot be considered equivalent to the surface flux, and the applicability of the flux-gradient method between a 2–3 m depth layer could be questionable. Hence, the question is: is this horizontal advection of dust large enough to induce a significant vertical gradient of the vertical diffusive flux near the surface, and thus to invalidate the constant dust flux layer hypothesis used to relate the diffusive flux to the surface flux?

We estimated the advection term as follows. Assuming that dust emission, surface roughness and thermal stability are similar between western and eastern erosion events for equivalent $u_*$ (and therefore equivalent mean $$), then $udC/dx$ can be approximated as $(C_w-C_e)/(F_w-F_e)$, where $C_w$ and $C_e$ are the mean dust concentrations at the intermediate height between the two Fidas of the western and eastern direction events of intensity $u_*$, respectively, and $F_w$ and $F_e$ are the dust-source fetch lengths in the western and eastern directions, respectively, being $F_w = 60\,\text{km}$ and $F_e = 10\,\text{km}$. As an estimation, we considered the mean wind speed $$ measured by the 2-D sonic sensor at 2 m high during erosion events, i.e. $=8.34\,\text{m\,s}^{-1}$, corresponding to $u_*$ between 0.35 and 0.43 m s$^{-1}$. For this $u_*$ interval and particles of 0.6 μm diameter, the advection term is $udC/dx = 5.1 \times 10^{-4}\,\text{μg\,m}^{-3}\,\text{s}^{-1}$ and the diffusive flux is about $F = 2.5\,\text{μg\,m}^{-2}\,\text{s}^{-1}$. If we neglect the gravitational settling term for these fine dust particles, then it means that the vertical gradient of the diffusive flux is $dF/dz = udC/dx = 5.1 \times 10^{-4}\,\text{μg\,m}^{-3}\,\text{s}^{-1}$. Between the two concentration measurement heights (1.7 m difference), this corresponds to a variation of the diffusive flux of about $8.7 \times 10^{-4}\,\text{μg\,m}^{-2}\,\text{s}^{-1}$, i.e. 0.035% of the flux. Between the intermediate height between the fidas and the surface ($z_{int} = 2.65\,\text{m}$), it corresponds to about 0.054% of the flux. Hence, for fine particles we find it reasonable to conclude that the dust flux layer is constant, and thus that the flux-gradient method is applicable between 1.8 and 3.5 m, and that the 2.65 m high diffusive flux can be related to the surface flux. We can conclude the same for larger particles regarding the small horizontal advection but we have to account for the gravitational settling term for the constant dust flux layer.

**Specific comments**

In specifying the similarity functions, z0 is sometimes considered and sometimes not, e.g., Eq. (2) and (5). How important is z0?

We thank Yaping Shao for making us aware of this slight inconsistency. We have revised our use of the flux-profile relationships and are now consistently including $z_0$ as the lower integration limit. Comparison with our earlier calculation confirmed that the impact of this change to our results is negligible. We have also revised our formulation of Eq. (1), which now reads:

$$\overline{U}(z) = \frac{u_*}{\kappa}\left[\ln\left(\frac{z}{z_0}\right) - \Psi_m\right] \tag{5}$$

as in Kaimal and Finnigan (1994). Also for consistency, we have removed "neglecting $\psi_m(\zeta_0)$" in line 222.

In general, I find the work very well done, and the authors have thoroughly studied the literature. But it is (for me at least) a very heavy paper with lengthy descriptions. I find the abstract very long. It may be trying to solve too many problems at once (size-resolved dust emission, deposition, fetch, haboob, BFT etc.). I believe the paper would have a larger impact, if it were more concentrated on the core issues.

We thank Yaping Shao for acknowledging the quality of our work. We agree with him that the paper is long. Indeed, Jasper Kok (the other reviewer) suggested to split the original paper into two separate papers (one focused on bulk dust emissions and one focused on size-resolved dust emissions). However, after careful consideration we have ultimately decided against splitting the paper. We believe that our comprehensive analysis of the dust PSD and its variability (the core and novel part of the paper), benefits from the exploratory analysis of the time series of bulk dust concentration, dust and saltation fluxes and meteorology at our site, along with a first view of the bulk saltation and sandblasting conditions. In order for this first part to become a separate paper, additional substantial developments and more exhaustive/detailed analyses would have been required, which are beyond our current scope. Our analysis of the PSD and its variability is comprehensive by design. It attempts to elucidate the potential causes of the PSD variability and therefore involves discussion on the deposition, fetch, haboob, and BFT. However this does not preclude future more specific studies and papers on the different aspects. We note that the revised version of the paper

includes some structural changes and additions to accommodate suggestions by both reviewers and to improve the quality of certain parts of the text. We have also reduced some parts of the text, moved material from the main paper to the Appendix and from the Appendix to the new Supplement document.

Here is the list of all relevant changes made in the manuscript:

– Abstract is now more concise

– Most of the appendices and associated figures along with the updated Fig. 10 (ratio of dry deposition to the estimated emitted flux), now Fig. S32, have been moved to the new Supplement material document.

– Part of the specifications needed to calculate the scattered intensities of the PSLs and the aspherical dust, originally described in Sect. 2.2.2, have been moved to the Appendix A to reduce the length of the paper.

– Original lines 420-435 along with the Fig. 5, both associated to the relationship between $z_0$ and $u_*$, have been moved to Sect. S7 in the new Supplement material document to reduce the length of the paper. Additionally, as suggested by Jasper Kok, in this new revised version, our $z_0$ measurements have been fitted not only to the relationship derived by Charnock (1955) but also to the modified Charnock's model proposed by Sherman (1992), which uses a more physical relation and accounts for the presence of a threshold (Sect. S7 in the Supplement).

– Subsection 2.4 in the Data and methods section is now called "Estimation of the size-resolved dry deposition and emitted fluxes" to include the methodology associated to the dry deposition velocity, dry deposition fluxes and estimated emitted flux. This was motivated by comments and suggestions from Jasper Kok. Appendix D contains details about the parameterizations used for dry deposition velocity.

– Subsection 3.3 in the Results and discussion section has been renamed as "Variability of the dust PSD at emission" and now only includes the first three subsections.

– The content in original Subsect. 3.3.4 has been split into the new Subsects. 3.4 "What explains the observed PSD variations? Potential roles of dry deposition and fetch length, aggregate disintegration, and haboob gust front" and 3.5 "Evaluation of the estimated dry deposition and emitted fluxes". The latter containing mostly the new results related to the estimated emitted flux, including the new Figs. 10 (which is based on the original Fig. J1 placed in the Appendix) and 11. Additional figures are shown in Sects. S11 and S12.

– The original Subsect. 3.3.5 "Comparison with Brittle Fragmentation Theory including super-coarse dust" is now the Sect. 3.6 and its two associated figures have been updated and now include also the normalized estimated emitted flux PSDs.

– As suggested by Jasper Kok a small discussion of the changes in the diffusive PSD with $u_*$ and wind direction in quantitative terms has been included in Sect. 3.3.2 along with some additional plots in Sect. S13 of the Supplement. Also a quantitative comparison between diffusive and estimated emitted fluxes considering dust particles as polystyrene latex spheres and assuming tri-axial ellipsoids has been added in Sects. 3.5 and 3.6, respectively.

– All the sections have been updated according to the comments of the reviewers

Additional changes

– As suggested by Jasper Kok, we determined $u_{*th}$ using both a linear and a 3/2 fitting of the saltation flux as a function of wind shear stress (Martin and Kok, 2017). Results are depicted in Sect. S3.

– Figs. 2 and 3 have been updated. Their new grey areas highlight times with $u_* > u_{*th}$. Also, we realized that when plotting Figs. 2f and 2g we had forgotten to remove the bins with diameters below $0.25\,\mu m$, considered unrealistic due to border measurement limitations. This has been corrected in the revised version.

– As recommended by Jasper Kok, Figs. 5, S6 and S7 have been updated to show the fits for $u_* > u_{*th}$. Also, Tables S1, S2 and S3 containing the obtained parameters from each regression curve along with their 95% confidence intervals have been updated accordingly.

– Figs. 7, 9, S12, S14 and S16 have been updated to adjust the y-axis limits and facilitate the comparison between figures.

**References**

Alfaro, S. C., Gaudichet, A., Gomes, L., and Maillé, M.: Modeling the size distribution of a soil aerosol produced by sandblasting, J. Geophys. Res., 102, 11 239–11 249, https://doi.org/10.1029/97JD00403, 1997.

Charnock, H.: Wind stress on a water surface, Q. J. Roy. Meteor. Soc., 81, 639–640, https://doi.org/10.1002/qj.49708135027, 1955.

Csanady, G.: Turbulent diffusion of heavy particles in the atmosphere, J. Atmos. Sci., 20, 201–208, https://doi.org/10.1175/1520-0469(1963)020<0201:TDOHPI>2.0.CO;2, 1963.

Dupont, S.: On the influence of thermal stratification on emitted dust flux, J. Geophys. Res. Atmos., p. e2022JD037364, https://doi.org/10.1029/2022JD037364, 2022.

Dupont, S., Alfaro, S., Bergametti, G., and Marticorena, B.: Near-surface dust flux enrichment in small particles during erosion events, Geophys. Res. Lett., 42, 1992–2000, https://doi.org/10.1002/2015GL063116, 2015.

Dupont, S., Rajot, J.-L., Lamaud, E., Bergametti, G., Labiadh, M., Khalfallah, B., Bouet, C., Marticorena, B., and Fernandes, R.: Comparison between eddy-covariance and flux-gradient size-resolved dust fluxes during wind erosion events, J. Geophys. Res. Atmos., 126, e2021JD034 735, https://doi.org/10.1029/2021JD034735, 2021.

Fernandes, R., Dupont, S., and Lamaud, E.: Investigating the role of deposition on the size distribution of near-surface dust flux during erosion events, Aeolian Res., 37, 32–43, https://doi.org/10.1016/j.aeolia.2019.02.002, 2019.

Gillette, D. A., Blifford Jr., I. H., and Fenster, C. R.: Measurements of aerosol size distributions and vertical fluxes of aerosols on land subject to wind erosion, J. Appl. Meteorol., pp. 977–987, https://doi.org/10.1175/1520-0450(1972)011<0977:MOASDA>2.0.CO;2, 1972.

Kaimal, J. C. and Finnigan, J. J.: Atmospheric boundary layer flows: their structure and measurement, Oxford university press, 1994.

Khalfallah, B., Bouet, C., Labiadh, M. T., Alfaro, S. C., Bergametti, G., Marticorena, B., Lafon, S., Chevaillier, S., Féron, A., Hease, P., Henry des Tureaux, T., Sekrafi, S., Zapf, P., and Rajot, J. L.: Influence of Atmospheric Stability on the Size Distribution of the Vertical Dust Flux Measured in Eroding Conditions Over a Flat Bare Sandy Field, J. Geophys. Res. Atmos., 125, e2019JD031 185, https://doi.org/10.1029/2019JD031185, 2020.

Klose, M., Jorba, O., Gonçalves Ageitos, M., Escribano, J., Dawson, M. L., Obiso, V., Di Tomaso, E., Basart, S., Montané Pinto, G., Macchia, F., et al.: Mineral dust cycle in the Multiscale Online Nonhydrostatic AtmospheRe CHemistry model (MONARCH) version 2.0, Geosci. Model Dev., 14, 6403–6444, 2021.

Kok, J. F.: A scaling theory for the size distribution of emitted dust aerosols suggests climate models underestimate the size of the global dust cycle, Earth, Atmospheric, and Planetary Sciences, 108, 1016–1021, https://doi.org/10.1073/pnas.1014798108, 2011a.

Kok, J. F.: Does the size distribution of mineral dust aerosols depend on the wind speed at emission?, Atmos. Chem. Phys., 11, 10 149–10 156, https://doi.org/10.5194/acp-11-10149-2011, 2011b.

Martin, R. L. and Kok, J. F.: Wind-invariant saltation heights imply linear scaling of aeolian saltation flux with shear stress, Science advances, 3, e1602 569, https://doi.org/10.1126/sciadv.1602569, 2017.

Shao, Y.: A model for mineral dust emission, J. Geophys. Res. Atmos., 106, 20 239–20 254, https://doi.org/10.1029/2001JD900171, 2001.

Shao, Y.: Physics and Modelling of Wind Erosion, Springer–Verlag, Berlin, 2 edn., https://doi.org/10.1007/978-1-4020-8895-7, 2008.

Shao, Y., Raupach, M., and Findlater, P.: Effect of saltation bombardment on the entrainment of dust by wind, J. Geophys. Res. Atmos., 98, 12 719–12 726, https://doi.org/10.1029/93JD00396, 1993.

Shao, Y., Ishizuka, M., Mikami, M., and Leys, J. F.: Parameterization of size-resolved dust emission and validation with measurements, J. Geophys. Res. Atmos., 116, https://doi.org/https://doi.org/10.1029/2010JD014527, 2011.

Sherman, D. J.: An equilibrium relationship for shear velocity and apparent roughness lenght in aeolian saltation, Geomorphology, 5, 419–431, https://doi.org/10.1016/0169-555X(92)90016-H, 1992.

Yin, X., Jiang, C., Shao, Y., Huang, N., and Zhang, J.: Large-eddy-simulation study on turbulent particle deposition and its dependence on atmospheric-boundary-layer stability, Atmospheric Chemistry and Physics, 22, 4509–4522, https://doi.org/https://doi.org/10.5194/acp-22-4509-2022, 2022.

---

## Author Comment (AC2)

**Referee #2 Jasper Kok**

We wish to thank Jasper Kok for his very valuable comments. Below we provide detailed responses to each comment separately. Our responses to reviewer comments are in blue normal text.

**Overview**

5     This paper reports on field measurements of dust emissions and studies how the diffusive dust flux depends on particle size, wind speed, and fetch. The reported measurements are probably the most extensive measurements of dust emission ever made and, in my view, represent a critical contribution to the field. This study makes an additional important contribution to the literature by showing the effect of dust deposition on the emitted dust PSD. I do think that parts of the paper can be improved substantially in both style and content, which could help the results of this paper be more fully appreciated by the community.

10     We thank Jasper Kok for his assessment of our paper and the helpful suggestions for improvement.

**Major Comments**

    1. Similar to the review by Yaping Shao, I think the paper tries to do too much, resulting in a very long paper that risks diluting some of the important findings. I would suggest that the authors split the paper into two separate papers: one focused on bulk dust emissions and one focused on size-resolved dust emissions. I think that will enhance the value of this exhaustive 15  work to the community and will make it easier on the finite attention span of most readers.

    We thank Jasper Kok for this suggestion. However, after careful consideration we decided not to split the paper into two separate papers for the following reasons. We believe that our comprehensive analysis of the dust PSD and its variability (the core and novel part of the paper), benefits from the exploratory analysis of the time series of bulk dust concentration, dust and saltation fluxes and meteorology, along with a first view of the bulk saltation and sandblasting conditions. In order for this first 20  part to become a separate paper, additional substantial developments and more exhaustive/detailed analyses, which are planned for future papers but are beyond our current scope, would have been required. Therefore, we decided to keep both parts. As discussed below, the revised version of the paper includes some structural changes and additions to accommodate Jasper Kok's comments and suggestions as much as possible and to improve the quality of certain parts of the text. We have also reduced some parts of the text and we have moved some material from the main paper to the Appendix and from the Appendix to the 25  Supplement.

    2. One of the key contributions of this paper is the finding that the shift towards a finer dust PSD with $u_*$ is likely due to an increase in the deposition flux of coarse dust with $u_*$. This is overall convincing but I noticed a few issues with the presentation that should be addressed to make this part of the paper stronger: First, the paper does not actually show explicitly that accounting for deposition explains the observed shift to a finer dust PSD. Please show the emitted flux by subtracting the 30  calculated dry deposition flux and discuss whether this indeed shows that accounting for the deposition flux eliminates the shift towards finer dust with $u_*$ (within the uncertainties). For this, it would also be needed to propagate uncertainty in the calculation of the deposition flux, which might be tricky. You could for instance use several different deposition flux models and use the range of their predictions as an estimate of uncertainty.

    We really thank the reviewer for this suggestion. As proposed, in the revised version of the paper, we estimate and show the 35  emitted flux after accounting for dry deposition. This has involved a substantial amount of work and modifications in the paper.
    In the original version of the paper we only implemented the dry deposition scheme used in Fernandes et al. (2019). However, deposition schemes are affected by strong uncertainties. Therefore we decided to estimate the deposition velocities also using another parameterization Zhang et al. (2001) (see Sect. 2.4 and the new Appendix D). Both parameterizations were compared with observation-based deposition velocities for $u_*$ smaller than the erosion threshold value in our site (new Fig. 10) and 40  from those observed by Bergametti et al. (2018) (Sect. S10). The parameterizations appeared to significantly underestimate the deposition velocities compared to our measurements. We, therefore, tuned the Zhang et al. (2001) dry deposition velocity parameterization to fit our measurement data (see Appendix D and Sect. S10) and extend it to $u_*$ values above the threshold of wind erosion. The emitted dust flux (new Fig. 11 and additional plots in Sect. S12) was then estimated as the diffusive flux plus the gravitational settling at the intermediate level between the two Fidas minus the dry deposition flux at the surface. Sects. S11 45  and S12 show the estimated dry deposition and emitted fluxes obtained using the tuned $v_{dep}$ parameterization as well as those

obtained from the Fernandes et al. (2019) and Zhang et al. (2001) $v_{dep}$ parameterizations. A quantitative comparison between diffusive flux and estimated emitted flux, calculated from the tuned $v_{dep}$ parameterization, has been added in the new Sect. 3.5. We show that the normalized emitted flux PSDs exhibit less dependencies upon $u_*$, a considerably lower shift towards finer dust and a lower decrease of super-coarse particles with diameters > 10 µm with increasing $u_*$ compared to the normalized net diffusive flux PSDs (see new Table I).

As explained in Sect. S12, to estimate the uncertainty of the estimated emitted flux, we neglected the uncertainty of dry deposition velocity $v_{dep}$ because 1) we did not have observation-based $v_{dep}$ estimates for $u_*$ values above the threshold of wind erosion and 2) the parameterizations from Fernandes et al. (2019) and Zhang et al. (2001) are likely afflicted by large structural uncertainties as evidenced from the comparison with measured data. Future work may further explore this uncertainty along with the causes for systematic underestimation. Also other deposition models could be used, but is out of the scope of this paper.

To accommodate these new results in the paper, we changed the structure of certain sections as follows: 1) Subsection 2.4 in the Data and methods section is now called "Estimation of the size-resolved dry deposition and emitted fluxes" to include the methodology associated to the dry deposition velocity, dry deposition fluxes and estimated emitted flux; 2) Subsection 3.3 in the Results and discussion section has been renamed as "Variability of the dust PSD at emission" and now only includes the first three subsections; 3) the content in original Subsect. 3.3.4 has been splitted into the new Subsects. 3.4 "What explains the observed PSD variations? Potential roles of dry deposition and fetch length, aggregate disintegration, and haboob gust front" and 3.5 "Evaluation of the estimated dry deposition and emitted fluxes", the latter containing mostly the new results about the estimated emitted flux; and 4) the original Subsect. 3.3.5 "Comparison with Brittle Fragmentation Theory including super-coarse dust" is now the Sect. 3.6 and its two associated figures have been updated and now include also the normalized estimated emitted flux PSDs.

Second, the changes in the PSD with $u_*$ seem to be quite small, which is a point that's easily missed because the changes are discussed mostly in qualitative terms. But while it is of substantial interest whether or not there is indeed a shift in PSD with $u_*$, the size of this effect might be of even greater interest because it for instance determines whether this is worth parameterizing in climate models. Therefore, could you include a plot that shows the shift in PSD quantitatively and with uncertainty? For instance by plotting the contributions of fine ($D_i < 2.5$ µm), coarse ($2.5 < D_i < 10$ µm) and super coarse ($D_i > 10$ µm) dust (per the size terminology in Adebiyi et al. (2023) as a function of $u_*$ for the two different wind directions and for the haboob events? That would be great to include both for the diffusive flux and for the estimated emitted flux (i.e., after subtracting the calculated dry deposition flux). And please also add some discussion of the changes in the PSD in quantitative terms and with uncertainty, especially for the estimated emitted flux, which will help the reader appreciate whether or not these changes are important from a broader Earth system perspective.

Probably the most controversial dependency found in the concentration and diffusive flux PSDs is the shift towards a finer dust PSD with increasing $u_*$. It is true that in the original version of the paper we discussed this feature in Sect. 3.3.2 only in qualitative terms. As suggested, in the revised version we have included a couple of sentences in this section discussing this dependency of diffusive flux PSD in quantitative terms using Fig. S30 as a support (which corresponds to the plot suggested by Jasper Kok for the diffusive flux with the only change that we have divided the contribution of fine dust in $\sim 0.37 < D_i < 1$ µm and $\sim 1 < D_i < 2.5$ µm). From a one-tailed test of significance we evaluated that this increase in the sub-micron fraction of diffusive flux with increasing $u_*$ was not statistically significant at a significance level of 0.05. However, on the contrary, we checked that this increase was statistically significant if we considered individually the two size bins between 0.37 and 0.49 µm and 0.49 and 0.65 µm. Similar results about the statistical significance were obtained when evaluating the larger sub-micron fraction found when winds came from the western sector compared to the eastern sector. While it was not statistically significant when considering the whole sub-micron fraction, it was when we considered only the two size bins between 0.37 and 0.49 µm and 0.49 and 0.65 µm. We also evaluated if for the mass fraction > 10 µm the differences among $u_*$ intervals were statistically significant at a significance level of 0.05 for both wind sectors and in this case these differences were statistically significant both considering only the last bin and the whole mass fraction > 10 µm. As previously said, results about the estimated emitted flux have been introduced in the new Sect. 3.5, where we refer to Fig. S31, which is analogous to Fig. S30 but for the estimated emitted flux. One-tailed tests of significance proved that despite the lower shift towards finer dust with

increasing $u_*$ for the estimated emitted flux, this increase was still statistically significant at a significance level of 0.05 when considering individually the two size bins between 0.37 and 0.49 µm and 0.49 and 0.65 µm, as also occurred for the diffusive flux. Also, despite the lower decrease of super-coarse particles with diameters > 10 µm with increasing $u_*$ for the estimated emitted flux, similarly to the diffusive flux the differences among $u_*$ intervals were statistically significant at a significance level of 0.05 for both wind sectors and considering both the whole mass fraction > 10 µm (Figs. S31c and S31d) and considering individually only the last integrated size bin (Figs. 11c and 11d).

3. The authors fit power laws to the bulk saltation and diffusive ($\approx$ dust emission) fluxes in Fig. 4. However, these fluxes are well known to depend on both $u_*$ and the threshold friction velocity $u_{*th}$ so these fits are not particularly useful or insightful. Can the authors obtain the $u_{*th}$ (for instance, from fitting the flux versus wind shear stress; (e.g. Martin and Kok, 2017) and compare their measurements of both saltation and dust emission fluxes against current parameterizations? This would need to be done for several periods if events (e.g., rainfall) changed the threshold during the campaign.

As suggested, we have determined the threshold friction velocity $u_{*th}$ by using both a linear and a 3/2 fitting of the saltation flux as a function of wind shear stress. The corresponding fits are presented in Fig. S3, with the standard error of the regression being slightly lower for the 3/2 fit. As a result, we have established $u_{*th} = 0.16 \, \mathrm{m \, s^{-1}}$, which is very close to our prior estimation. As later suggested by Jasper Kok, now the fits in Fig. 4 are reported only above the threshold friction velocity. During the campaign, the surface in our main site remained essentially unchanged. As mentioned in the paper, during the night of September 6th there was water flowing downriver, which caused flooding of large areas in the vicinity of our site on the next day. However, this did not affect our site itself. The comparison of our measured saltation and dust emission fluxes against current parameterizations is planned but falls beyond the scope of the paper.

4. The authors report measurements for dust with diameter up to 20 µm. This is quite valuable, as very few measurements of diffusive dust fluxes for D > 10 µm have been made. However, one of the reasons that there are so few measurements is that the transmission efficiency of inlets normally decreases sharply with particle size and furthermore that the diameter at which 50% is transmitted decreases with wind speed (e.g. Von der Weiden et al., 2009). The authors are well aware of this problem and on lines 161-4 they cite several previous studies that have used their sigma-2 sampling head and concluded that this is accurate for coarse and super coarse dust. However, this issue seems critical for the papers conclusions - for instance, measured changes in dust PSD could be due to a decrease in sampling efficiency with $u_*$ for super coarse dust. Therefore, please elaborate on the evidence that this inlet is in fact suitable for super coarse dust. For instance, has the sampling efficiency actually been measured as a function of particle size and wind speed?

This is a very good point and a remaining uncertainty. As we mention in the methods section, the sigma-2 inlet has been designed to be efficient for coarse particles, and it is expected to be largely insensitive to wind intensity as it ensures a wind-sheltered, low-turbulence air volume inside the sampler. However, we do not really know to what extent the transmission efficiency may decrease with friction velocity and particle size. This is clearly a potential problem for most inlets and most previous field campaigns. It is not an easy task to quantify the efficiency of this inlet to friction velocity. Experimentally it is difficult, probably only possible in the laboratory under controlled conditions, and theoretically, we cannot infer it due to its relatively complex geometry. We have more recently performed a dust field campaign in Iceland using this inlet in parallel with another directional inlet (which is easier to model) and their comparison may provide some insights into this issue in the future. However, the proper analysis of this dataset will require some time and the potentially interesting results will be considered in a future publication. In our revised version of the paper we have tried to reflect this uncertainty (lines 148-150 in Sect. 2.2.2 and 704-709 in the conclusions). In fact, in the conclusion section we argue that given the large uncertainties associated to resistance-based deposition parameterizations it cannot be discarded that our tuned parameterization partly overestimates dry deposition velocity, thereby indirectly accounting for sampling inefficiencies of the inlet, which may affect coarse and super coarse particles for high wind velocities. Although the Sigma-2 inlet has been designed to be efficient for coarse particles, we currently ignore its sensitivity upon $u_*$.

5. The paper is made very long by the inclusion of no fewer than 10 appendices. Most of these appendices are in my view of interest only to a few readers and are not needed to appreciate the paper main conclusions. I therefore recommend moving most of the appendices to the supplement (this will also cut down on publication costs!). One exception is Figure J1, which

shows the deposition velocity as a function of $D$ and $u_*$. This figure is in fact critical to understanding the paper's results and conclusions and I think it should be moved be moved to the main text.

As suggested most of the appendices and associated figures along with the updated Fig. 10 (ratio of dry deposition to the estimated emitted flux calculated from the tuned parameterization) have been moved to the new Supplement material document. In particular the updated Fig. 10 which now corresponds to the new Fig. S32 is included in Sect. S14. Ratios obtained using the Fernandes et al. (2019) and Zhang et al. (2001) parameterizations for $v_{dep}$ have been also included there.

Also, as recommended, the updated figure about dry deposition velocity as a function of diameter and friction velocity (originally Fig. J1) has been included now in the main text as Fig. 10. This updated figure shows the original dry deposition velocity parameterization used in our original paper (previously used in Dupont et al. (2015) and Fernandes et al. (2019)), the Zhang et al. (2001) parameterization and a tuned parameterization that fits our experimentally-based deposition velocity estimates. Two additional figures closely related to Fig. 10 have been added in Sect. S10, one compares our experimental $v_{dep}$ data with the measurements conducted by Bergametti et al. (2018) (Fig. S17) and the other one shows the sensitivity of the tuned parameterization used to estimate $v_{dep}$ (described in Appendix D) to three parameters of the parameterization (Fig. S18).

Furthermore, in order to reduce the length of the paper: 1) part of the specifications needed to calculate the scattered intensities of the PSLs and the aspherical dust, originally described in Sect. 2.2.2, have been moved to the Appendix A and 2) Original lines 420-435 along with the Fig. 5, both associated to the relationship between $z_0$ and $u_*$, have been moved to Sect. S7.

6. The authors compared their results to the predicted PSD from both the original and the updated parameterizations based on brittle fragmentation theory (BFT), which is my own past work. This parameterization has some dependence on the fully dispersed size distribution of the parent soil (e.g., Eq. 3 in Kok (2011)). But because the fully-dispersed PSD of the soil of a GCM grid box is unknown, the standard version of this parameterization uses an "average" size distribution of desert soils. However, the authors actually measured the fully-dispersed soil PSD (Fig. A1)! Therefore, please perform a more test against BFT by inserting the cumulative soil PSD into Eq. 3 and redoing the comparisons.

We thank Jasper Kok for highlighting the difference between the theory based on brittle fragmentation and the parameterization derived from it that is used in models. The parameterization assumes an average fully-dispersed soil PSD due to the lack of reliable gridded fully-dispersed PSDs of the soil. Our goal in this paper is to compare the obtained PSDs with the average emitted dust PSD based on Brittle Fragmentation that is currently proposed and used in many models. As in the case of the bulk saltation and diffusive fluxes, a detailed comparison of our measurements with theory is beyond the scope of this paper. We plan to compare our PSDs with theories of dust emission in other papers. We also note that to test the updated BFT parameterization we also have the minimally-dispersed PSD. To avoid confusion, in the paper, we have highlighted the distinction between the theory and the parameterization when introducing the comparison with BFT. Also note that we have updated current Figs. 12 and 13 to incorporate also the estimated emitted dust flux PSD in addition to the concentration and diffusive flux PSDs.

**Specific Comments**

Do the authors know whether their measurements were of transport-limited or supply-limited saltation / dust emission? I assume the former, and it would be good to state that somewhere because it affects the physics of dust emission and thus the interpretation of the results.

The large observed saltation fluxes and sand dunes in the vicinity of our site suggest that sand supply was substantial, even though a characteristic of our site was the hard surface crusting. We therefore expect that atmospheric momentum was the main driver for sediment transport and not particle availability. We have added a corresponding sentence in lines 380-382 (Sect.3.2)

The authors used the law-of-the-wall to calculate $u_*$. However, the law-of-the-wall is technically applicable only for idealized conditions that, as Yaping Shao also pointed out, might not apply. It'd therefore be more accurate to quantify $u_*$ from the Reynolds stress obtained from the two 3-D anemometers at 1 and 3 m (lines 140-1). Depending on how close the experimental conditions were to homogeneous to isotropic turbulence, there might be substantial differences between the Reynolds-stress based $u_*$ and the law-of-the-wall-based $u_*$.

When we started this work we calculated $u_*$ both from the law-of-the-wall method and from eddy covariance at $1\,\mathrm{m}$, $3\,\mathrm{m}$ and extrapolated to the surface. The following figure illustrates that our preliminary results were largely in agreement. Ultimately, we opted to use the $u_*$ calculated from the law-of-the-wall method, due to gaps in our 3D sonic measurements during the haboob events. As we plan to make these data available after publication, other groups may examine the sensitivity of our results to the calculation method of $u_*$.

[Figure]

**Figure 1.** Time series of $u_*$ calculated from the law of the wall method (blue line) and eddy covariance method at $1\,\mathrm{m}$ (magenta line), $3\,\mathrm{m}$ (grey line) and extrapolated to the surface (orange line).

Line 218: confront –> convert

Changed by "compare"

Line 242: The von Karman constant actually depends a bit on flow properties like the Reynolds number and for atmospheric boundary layer flow it was measured to be 0.387 (Andreas et al., 2006). I recommend you use that.

We decided to keep the value of 0.4 to be consistent with the expressions of the similarity functions for momentum and sensible heat, derived from the Kansas-experiment of 1968 by Businger et al. (1971) and later modified by Högström (1988). The expressions from Businger et al. (1971) assume the von-Kármán constant $\kappa = 0.35$. Högström (1988) considered important criticisms of the Kansas experiment and recalculated these functions for $\kappa = 0.40$.

Line 282: please define the Schmidt number

Added as requested.

The relation used to link $z_0$ and $u_*$ was a good first estimate at the time it was proposed (50s and 60s) but is outdated for several reasons, but primarily because it does not account for the presence of a threshold. The authors should thus use a more physical relation, such as that proposed in Sherman (1992), which does account for the threshold.

Due to the paper extension, we have opted to eliminate the original Fig. 5 from the main text, which depicted the relationship between $z_0$ and $u_*$ under wind erosion conditions. It is now included in Sect. S7 as Fig. S8. Furthermore, as asked we have included a new figure depicting not only the adjustment to the expression derived by Charnock (1955) but also to the expression proposed in Sherman (1992), that accounts for threshold friction velocity $u_{*th}$ (Fig. S9).

Where appropriate, please specify whether $R^2$ values are calculated in linear space or log space (e.g., Fig. 4, where $R^2$ should be calculated in log space because the measurements span almost 3 orders of magnitude).

Added as requested.

The authors have several fits that extend down to $u_* = 0.1\,\mathrm{m\,s^{-1}}$. However, presumably they found many negative (for dust)
210 or zero (for saltation) fluxes in the range between 0.1 and 0.15 (or 0.20) $\mathrm{m\,s^{-1}}$. So how did you treat those zeroes in the fit?
Just omitting seems incorrect. I recommend only reporting fits for $u_*$ above the threshold.

As recommended, Figs. 5, S6 and S7 have been updated to show the fits for $u_* > u_{*th}$. Also, Tables S1, S2 and S3 containing
the obtained parameters from each regression curve along with their 95% confidence intervals have been updated accordingly.

Line 463: there's no such thing as "instantaneous $u_*$" because $u_*$ is by definition a time-averaged quantity over at least
215 several minutes. Please correct.

We have corrected it.

Lines 478-80: The authors note that they do not find "any clear effect of atmospheric stability independent of $u_*$ upon the
PSD" but they do not actually show this. Since this is an ongoing debate in the literature, as the authors note (Khalfallah et al.,
2020; Shao et al., 2020; Dupont, 2022), could you include a graph supporting your conclusion here?

220 A thorough investigation is necessary to determine whether atmospheric stability has a discernible effect on the PSD beyond
the friction velocity range present in our measurements. Up to now, we only possess preliminary results to address this question.
Therefore, we have reformulated original lines 478-480 highlighting the ongoing debate and emphasizing the requirement for
further analysis in the future (current lines 428-431).

Line 531: statistically significant at what level? Please include a p-value.

225 As mentioned in the answer to the second major comment, we performed two types of one-tailed test of significance (see
Sect. S13 for more details) one considering the sub-micron fraction and the other one considering individually the size bins
between 0.37–0.49 μm and 0.49–0.65 μm to evaluate the statistical significance of the shift towards a finer diffusive flux PSD
with increasing $u_*$. We obtained that while it was not statistically significant when considering the whole sub-micron fraction,
it was statistically significant at a significance level of 0.05 when the two bins were considered individually. It is important to
230 add that the comparison was made between the $u_*$ intervals of 0.15–0.20 $\mathrm{m\,s^{-1}}$ and 0.30–0.35 $\mathrm{m\,s^{-1}}$. The $u_*$ interval between
0.35–0.43 $\mathrm{m\,s^{-1}}$ was not used due to the small number of samples, specially in the western sector.

Lines 581-3: by how much is the sub-micron proportion higher for western winds? Is this difference statistically significant?
And what's the p-value?

Similar one-tailed tests of significance as before were applied here, but in this case to evaluate the statistical significance
235 of the change in the sub-micron fraction of diffusive flux for different wind sectors, selecting the $u_*$ interval between 0.25–
0.30 $\mathrm{m\,s^{-1}}$. This $u_*$ interval was chosen as we had similar number of samples in both wind sectors. As before, we obtained
that when considering the whole sub-micron fraction the differences between wind sectors were not statistically significant.
However, when considering individually the size bins between 0.37–0.49 μm and 0.49–0.65 μm the difference was statistically
significant at a significance level of 0.05.

240 Line 602: here and elsewhere in the paper, I think the use of the term "aggregate fragmentation" is confusing in the context
of the Alfaro '97 and Shao '01 models because those papers did not hypothesize that saltation bombardment causes aggregates
fragmentation, which is a term that has a specific meaning in material science (i.e., the size of the largest fragment is small
compared to the size of the original object). So this could cause the reader to think this is referring to brittle material fragmen-
tation, which does hypothesize that soil aggregates are fragmented by saltation bombardment. Please use a term that is more
245 consistent with these theories, such as "aggregate disintegration".

As suggested we have changed the term "aggregate fragmentation" to "aggregate disintegration".

Line 609-613: I personally think that the explanation of the smaller fetch length during haboobs is not clear here. That's a shame because it's a very elegant natural experiment. So I recommend rewriting this more clearly, perhaps with a schematic illustration.

250 We have rephrased those lines.

Line 685: I think "proves" is too strong a word here. Maybe "provides evidence for" or "indicates"?

Changed as requested.

At many places in the manuscript, the authors use parentheses to indicate the opposite of a statement or to include multiple values in a sentence, presumably to save space. This practice obscures the writing and makes the paper more difficult to read.
255 I thus recommend the authors eliminate this from the paper. In most cases, the opposite statement is obvious anyways so it's really not needed (e.g., "the proportion of submicron (supermicron) particles decreases (increases) in the concentration PSD between calm (purplish and blueish lines) and well-developed conditions (yellow, orange, and red lines)"). See also https://agupubs.onlinelibrary.wiley.com/doi/abs/10.1029/2010EO450004

Changed as requested.

260 Additional changes

– Figs. 2 and 3 have been updated. Their new grey areas highlight times with $u_* > u_{*th}$. Also, we realized that when plotting Figs. 2f and 2g we had forgotten to remove the bins with diameters below $0.25\,\mu m$, considered unrealistic due to border measurement limitations. This has been corrected in the revised version.

– Figs. 7, 9, S12, S14 and S16 have been updated to adjust the y-axis limits and facilitate the comparison between figures.

**265 References**

Adebiyi, A., Kok, J., Murray, B., Ryder, C., Stuut, J.-B., Kahn, R., Knippertz, P., Formenti, P., Mahowald, N., Pérez García-Pando, C., Klose, M., Ansmann, A., Samset, B., Ito, A., Balkanski, Y., Di Biagio, C., Romanias, M., Huang, Y., and Meng, J.: A review of coarse mineral dust in the Earth system, Aeolian Research, 60, 100 849, https://doi.org/https://doi.org/10.1016/j.aeolia.2022.100849, 2023.

Andreas, E., Claey, K., Jordan, R., Fairall, C., Guest, P., Perrson, P., and Grachev, A.: Evaluations of the von K [notdef] rm [notdef] n constant
in the atmospheric surface layer, J. Fluid. Mech, 559, 117 149, https://doi.org/10.1017/S0022112006000164, 2006.

Bergametti, G., Marticorena, B., Rajot, J.-L., Foret, G., Alfaro, S., and Laurent, B.: Size-resolved dry deposition velocities of dust particles: in situ measurements and parameterizations testing, Journal of Geophysical Research: Atmospheres, 123, 11–080, https://doi.org/10.1029/2018JD028964, 2018.

Businger, J. A., Wyngaard, J. C., Izumi, Y., and Bradley, E. F.: Flux-profile relationships in the atmospheric surface layer, J. Atmos. Sci., 28, 181–189, https://doi.org/10.1175/1520-0469(1971)028<0181:FPRITA>2.0.CO;2, 1971.

Charnock, H.: Wind stress on a water surface, Q. J. Roy. Meteor. Soc., 81, 639–640, https://doi.org/10.1002/qj.49708135027, 1955.

Dupont, S.: On the influence of thermal stratification on emitted dust flux, J. Geophys. Res. Atmos., p. e2022JD037364, https://doi.org/10.1029/2022JD037364, 2022.

Dupont, S., Alfaro, S., Bergametti, G., and Marticorena, B.: Near-surface dust flux enrichment in small particles during erosion events, Geophys. Res. Lett., 42, 1992–2000, https://doi.org/10.1002/2015GL063116, 2015.

Fernandes, R., Dupont, S., and Lamaud, E.: Investigating the role of deposition on the size distribution of near-surface dust flux during erosion events, Aeolian Res., 37, 32–43, https://doi.org/10.1016/j.aeolia.2019.02.002, 2019.

Högström, U.: Non-dimensional wind and temperature profiles in the atmospheric surface layer: A re-evaluation, Bound. Lay. Meteorol., 42, 55–78, https://doi.org/10.1007/BF00119875, 1988.

Khalfallah, B., Bouet, C., Labiadh, M. T., Alfaro, S. C., Bergametti, G., Marticorena, B., Lafon, S., Chevaillier, S., Féron, A., Hease, P., Henry des Tureaux, T., Sekrafi, S., Zapf, P., and Rajot, J. L.: Influence of Atmospheric Stability on the Size Distribution of the Vertical Dust Flux Measured in Eroding Conditions Over a Flat Bare Sandy Field, J. Geophys. Res. Atmos., 125, e2019JD031 185, https://doi.org/10.1029/2019JD031185, 2020.

Kok, J. F.: A scaling theory for the size distribution of emitted dust aerosols suggests climate models underestimate the size of the global dust cycle, Earth, Atmospheric, and Planetary Sciences, 108, 1016–1021, https://doi.org/10.1073/pnas.1014798108, 2011.

Martin, R. L. and Kok, J. F.: Wind-invariant saltation heights imply linear scaling of aeolian saltation flux with shear stress, Science advances, 3, e1602 569, https://doi.org/10.1126/sciadv.1602569, 2017.

Shao, Y., Zhang, J., Ishizuka, M., Mikami, M., Leys, J., and Huang, N.: Dependency of particle size distribution at dust emission on friction velocity and atmospheric boundary-layer stability, Atmos. Chem. Phys., 20, 12 939–12 953, https://doi.org/10.5194/acp-20-12939-2020, 2020.

Sherman, D. J.: An equilibrium relationship for shear velocity and apparent roughness lenght in aeolian saltation, Geomorphology, 5, 419–431, https://doi.org/10.1016/0169-555X(92)90016-H, 1992.

Von der Weiden, S.-L., Drewnick, F., and Borrmann, S.: Particle Loss Calculator–a new software tool for the assessment of the performance of aerosol inlet systems, Atmospheric Measurement Techniques, 2, 479–494, https://doi.org/https://doi.org/10.5194/amt-2-479-2009, 2009.

Zhang, L., Gong, S., Padro, J., and Barrie, L.: A size-segregated particle dry deposition scheme for an atmospheric aerosol module, Atmospheric environment, 35, 549–560, 2001.